# ADAPTING TO BOTH FINITE-SAMPLE AND ASYMPTOTIC REGIMES

## ABSTRACT

This paper introduces an empirical risk minimization based approach with concomitant scaling, which eliminates the need for tuning a robustification parameter in the presence of heavy-tailed data. This method leverages a new loss function that concurrently optimizes both the mean and robustification parameters. Through this dual-parameter optimization, the robustification parameter automatically adjusts to the unknown data variance, rendering the method self-tuning. Our approach surpasses previous models in both computational and asymptotic efficiency. Notably, it avoids the reliance on cross-validation or Lepski's method for tuning the robustification parameter, and the variance of our estimator attains the Cramér-Rao lower bound, demonstrating optimal efficiency. In essence, our approach demonstrates optimal performance across both finite-sample and large-sample scenarios, a feature we describe as *algorithmic adaptivity to both asymptotic and finite-sample regimes*. Numerical studies lend strong support to our methodology.

## 1 INTRODUCTION

The success of many statistical and learning methods heavily relies on the assumption of sub-Gaussian errors (Wainwright, 2019). A random variable $Z$ is considered to have sub-Gaussian tails if there exist constants $c_1$ and $c_2$ such that $\mathbb{P}(|Z - \mathbb{E}Z| > t) \leq c_1 \exp(-c_2 t^2)$ for any $t \geq 0$. However, in many practical applications, data are often collected with a high degree of noise. For instance, in the context of gene expression data analysis, it has been observed that certain gene expression levels exhibit kurtoses much larger than 3, regardless of the normalization method used (Wang et al., 2015). Furthermore, a recent study on functional magnetic resonance imaging (Eklund et al., 2016) demonstrates that the principal cause of invalid functional magnetic resonance imaging inferences is that the data do not follow the assumed Gaussian shape. It is therefore important to develop robust and efficient statistical methods with desirable statistical performance in the presence of heavy-tailed data, which refer to data with only finite variances.

This paper focuses on mean estimation problems with potentially heavy-tailed data, which serves as the foundation for tackling more general problems. Specifically, we consider a generative model for data $\{y_i, 1 \leq i \leq n\}$:

$$y_i = \mu^* + \varepsilon_i, \ 1 \leq i \leq n, \tag{1.1}$$

where $\varepsilon_i \in \mathbb{R}$ are independent and identically distributed (i.i.d.) copies of $\varepsilon$, following the law $F_0$ with zero mean and only finite variance. Specifically, $\mathbb{E}_{\varepsilon \sim F_0} \varepsilon = 0$ and $\mathbb{E}_{\varepsilon \sim F_0} \varepsilon^2 = \sigma^2$.

When estimating the mean, the sample mean estimator $\sum_{i=1}^{n} y_i/n$ generally achieves only a polynomial-type nonasymptotic confidence width (Catoni, 2012) under the conditions where the errors have only finite variances. Specifically, there exists a distribution $F = F_{n,\delta}$ for $\varepsilon$ with a zero mean and a variance of $\sigma^2$, such that the followings hold simultaneously:

$$\mathbb{P}\left(\left|\sum_{i=1}^{n} \frac{y_i}{n} - \mu^*\right| \leq \sigma\sqrt{\frac{1}{2n} \cdot \frac{1}{\delta}}\right) \geq 1 - 2\delta, \ \ \forall\, \delta \in \left(0, 1/2\right); \tag{1.2}$$

$$\mathbb{P}\left(\left|\sum_{i=1}^{n} \frac{y_i}{n} - \mu^*\right| \leq \sigma\sqrt{\frac{1}{2n} \cdot \frac{1}{\delta}}\left(1 - \frac{2e\delta}{n}\right)^{(n-1)/2}\right) \leq 1 - 2\delta, \ \ \forall\, \delta \in \left(0, (2e)^{-1}\right).$$

In essence, the above indicates that the convergence of the sample mean to the true mean is notably slow when the error terms are characterized by only finite variances.

Catoni (2012) made an important step towards mean estimation by introducing a robust estimator $\widehat{\mu}(\tau)$, which depends on a tuning parameter $\tau$ and achieves logarithmic deviation from the true mean $\mu^*$ with respect to $1/\delta$. For a sufficiently large sample size $n$ and optimal tuning of $\tau$, this estimator satisfies the following concentration inequality:

$$\mathbb{P}\left(|\widehat{\mu}(\tau) - \mu^*| \leq c\sigma\sqrt{\frac{1}{n} \cdot \log\left(\frac{1}{\delta}\right)}\right) \geq 1 - 2\delta, \quad \forall\, \delta \in \left(0, 1/2\right), \tag{1.3}$$

where $c$ is some constant. Such estimators are referred to as sub-Gaussian mean estimators due to their performance equivalence to scenarios assuming sub-Gaussian data. Catoni's estimator is based on the empirical risk minimization (ERM) framework and thus can be generalized to various contexts (Brownlees et al., 2015; Hsu & Sabato, 2016; Fan et al., 2017; Avella-Medina et al., 2018; Lugosi & Mendelson, 2019b; Lecué & Lerasle, 2020; Wang et al., 2021; Sun et al., 2020). For a recent comprehensive review, see Ke et al. (2019).

However, implementing Catoni's estimator (Catoni, 2012) requires careful tuning the parameter $\tau = \tau(\sigma)$, which is dependent on the unknown variance $\sigma^2$. This process often involves computationally intensive techniques such as cross-validation or Lepski's method (Catoni, 2012). For instance, when using the adaptive Huber estimator (Sun et al., 2020; Avella-Medina et al., 2018) to estimate each entry of a $d \times d$ covariance matrix, up to $O(d^2)$ tuning parameters can be involved. Utilizing cross-validation or Lepski's method in such scenarios significantly escalates the computational burden as $d$ increases. To mitigate these computational barriers, median-of-means (MoM) techniques (Devroye et al., 2016; Lugosi & Mendelson, 2019b;a; Lecué & Lerasle, 2020) can be used to construct robust and tuning-free estimators; see Section 3.2. However, based on our experience, MoM typically underperforms numerically compared to ERM-based estimators. An asymptotic analysis reveals that the relative efficiency of the MoM estimator, compared to a fully efficient estimator, is only $2/\pi \approx 0.64$. These observations prompt a pertinent question:

*Is it possible to develop computationally and statistically effcient robust estimators for data with finite and unknown variances?*

In response, this paper introduces a robust empirical risk minimization approach with concomitant scaling. We utilize a new loss function that is smooth with respect to both the mean and robustification parameters. By joint optimizing these parameters, we prove that the resulting robustification parameter can automatically adapt to the unknown variance, enabling the resulting mean estimator to achieve sub-Gaussian accuracy up to logarithmic terms. Thus, our approach eliminates the need for cross-validation or Lepski's method for tuning, significantly enhancing the computational efficiency of robust data analysis in practical settings. Moreover, from an asymptotic viewpoint, we establish that our proposed estimator is asymptotically efficient, achieving the Cramér-Rao lower bound (Van der Vaart, 2000). In essence, our approach demonstrates optimal performance across both finite-sample and large-sample scenarios, a feature we describe as *algorithmic adaptivity to both asymptotic and finite-sample regimes*.

**Paper overview**   Section 2 introduces a novel loss function and presents the empirical risk minimization (ERM) approach. The theoretical properties are presented in Section 3, where we also compare our estimator with the MoM mean estimator in terms of asymptotic performance. Section 4 provides numerical experiments. We conclude in Section 5. The supplementary material collects additional results and proofs of the main results.

**Notation**   We summarize here the notation that will be used throughout the paper. We use $c$ and $C$ to denote generic constants which may change from line to line. For two sequences of real numbers $\{a_n, n \geq 1\}$ and $\{b_n, n \geq 1\}$, $a_n \lesssim b_n$ or $a_n = O(b_0)$ denotes $a_n \leq Cb_n$ for some constant $C > 0$, and $a_n \gtrsim b_n$ if $b_n \lesssim a_n$. We use $a_n \propto b_n$ to denote that $a_n \gtrsim b_n$ and $a_n \lesssim b_n$. The $\log$ operator is understood with respect to the base $e$. For a function $f(x, y)$, we use $\nabla_x f(x, y)$ or $\frac{\partial}{\partial x} f(x, y)$ to denote its partial derivative of $f(x, y)$ with respect to $x$. Let $\nabla f(x, y)$ denote the gradient of $f(x, y)$. For a vector $x \in \mathbb{R}^d$, $\|x\|_2$ denotes its Euclidean norm. For a symmetric positive semi-definite matrix $\Sigma$, $\lambda_{\max}(\Sigma)$ denotes its largest eigenvalue. For any set $\mathcal{A}$, $1(\mathcal{A})$ is the indicator function of the set $\mathcal{A}$.

## 2 A LOSS FUNCTION WITH CONCOMITANT SCALING

This section introduces a new loss function designed to robustly estimate the mean of distributions with only finite variances, while also facilitating automatic tuning of the robustification parameter. We start with the pseudo-Huber loss (Hastie et al., 2009):

$$\ell_\tau(x) = \tau\sqrt{\tau^2 + x^2} - \tau^2 = \tau^2\sqrt{1 + x^2/\tau^2} - \tau^2, \tag{2.1}$$

where $\tau$ acts as a tuning parameter. This loss function mirrors the behavior of the Huber loss (Huber, 1964), approximating $x^2/2$ when $x^2 \lesssim \tau^2$ and transitioning to a linear form with slope $\tau$ when $x^2 \gtrsim \tau^2$. We refer to $\tau$ as the robustification parameter because it mediates the balance between quadratic loss and least absolute deviations loss, with the latter inducing robustness. In practice, tuning $\tau$ often requires computationally intensive methods such as Lepski's method (Catoni, 2012) or cross-validation (Sun et al., 2020).

To bypass these computationally demanding methods, our objective is to develop a novel loss function that depends on both the mean parameter $\mu$ and the robustification parameter $\tau$ (or its equivalent). By jointly optimizing these parameters, we can achieve an automatically tuned robustification parameter $\widehat{\tau}$, which in turn leads to the corresponding self-tuned mean estimator $\widehat{\mu}(\widehat{\tau})$. In contrast to the Huber loss (Sun et al., 2020), the pseudo-Huber loss is a smooth function of $\tau$, making optimization with respect to $\tau$ possible. To motivate the new loss function, let us first consider the estimator $\widehat{\mu}(\tau)$ with $\tau$ fixed *a priori*:

$$\widehat{\mu}(\tau) = \arg\min_\mu \left\{ \frac{1}{n} \sum_{i=1}^n \ell_\tau(y_i - \mu) \right\}. \tag{2.2}$$

Below, we provide an informal result, with its rigorous version available in the appendix.

**Theorem 2.1** (An informal result)**.** Take $\tau = \sigma\sqrt{n}/z$ with $z = \sqrt{\log(1/\delta)}$, and assume $n$ is sufficiently large. Then, for any $0 < \delta < 1$, with probability at least $1 - \delta$, we have

$$|\widehat{\mu}(\tau) - \mu^*| \lesssim \sigma\sqrt{\frac{\log(2/\delta)}{n}}.$$

The result above demonstrates that when $\tau = \sigma\sqrt{n}/z$ with $z = \sqrt{\log(1/\delta)}$, the estimator $\widehat{\mu}(\tau)$ achieves the desired sub-Gaussian performance. Here, the sole unknown in $\tau$ is the standard deviation $\sigma$. In view of this, we treat $\sigma$ as an unknown parameter $v$, and substitute $\tau = \sqrt{n}v/z$ into the loss function (2.1), obtaining

$$\ell(x, v) := \ell_\tau(x) = \frac{nv^2}{z^2}\left(\sqrt{1 + \frac{x^2z^2}{nv^2}} - 1\right), \tag{2.3}$$

where $z$ acts as a confidence parameter due to its dependence on $\delta$ as specified in the preceding theorem.

Instead of determining the optimal $\tau$, we will identify the optimal $v$, which is intuitively expected to approximate the underlying standard deviation $\sigma$. We will use the term "robustification parameter interchangeably for both $\tau$ and $v$, as they differ only by a scaling factor. However, directly minimizing $\ell(x, v)$ with respect to $v$ leads to meaningless solutions, specifically $v = 0$ and $v = +\infty$. To circumvent these trivial outcomes, we we modify the loss function by dividing $\ell(x, v)$ by $v$ and then adding a linear penalty term $av$. This will be referred to as the penalized pseudo-Huber loss, which is formally defined as follows.

**Definition 2.2** (Penalized pseudo-Huber loss)**.** The penalized pseudo-Huber loss $\ell^{\mathrm{P}}(x, v)$ is defined as:

$$\ell^{\mathrm{P}}(x, v) := \frac{\ell(x, v) + av^2}{v} = \frac{nv}{z^2}\left(\sqrt{1 + \frac{x^2z^2}{nv^2}} - 1\right) + av, \tag{2.4}$$

where $n$ is the sample size, $z$ is a confidence parameter, and $a$ is an adjustment factor.

We thus propose to jointly optimize over $\mu$ and $v$ by solving the following ERM problem:

$$\{\widehat{\mu}, \widehat{v}\} = \underset{\mu, v}{\operatorname{argmin}} \left\{ L_n(\mu, v) := \frac{1}{n} \sum_{i=1}^{n} \ell^{\mathrm{P}}(y_i - \mu, v) \right\}. \tag{2.5}$$

When $v$ is fixed *a priori*, solving the optimization problem above with respect to $\mu$ is equivalent to directly minimizing the empirical pseudo-Huber loss in (2.2) with $\tau = v\sqrt{n}/z$.

To better understand the loss function $L_n(\mu, v)$, let us first examine its population counterpart:

$$L(\mu, v) = \mathbb{E}L_n(\mu, v) = \frac{nv}{z^2} \mathbb{E}\left( \sqrt{1 + \frac{(y - \mu)^2 z^2}{nv^2}} - 1 \right) + av.$$

We define the population oracle $v_*$ as the value of $v$ that minimizes $L(\mu^*, v)$ when the true mean $\mu^*$ is known *a priori*, that is $v_* = \operatorname{argmin}_\tau L(\mu^*, v)$, or equivalently, ensuring the gradient of $L(\mu^*, v)$ with respect to $v$ at $v = v_*$ is zero:

$$\nabla_v L(\mu^*, v)\big|_{v=v_*} = \left\{ \frac{n}{z^2} \left( \nabla_v \mathbb{E}\sqrt{v^2 + \frac{\varepsilon^2 z^2}{n}} - 1 \right) + a \right\} \bigg|_{v=v^*} = 0.$$

By switching the order of differentiation and expectation, we derive:

$$\mathbb{E}\frac{v_*}{\sqrt{v_*^2 + z^2 \varepsilon^2/n}} = 1 - \frac{az^2}{n}. \tag{2.6}$$

Our first key result leverages the above characterization of $v_*$ to demonstrate how $v_*$ automatically adapts to the unknown standard deviation $\sigma$, thus hinting the effectiveness of our methodology. Let $\sigma_{x^2}^2 := \mathbb{E}\{\varepsilon^2 1(\varepsilon^2 \leq x^2)\}$.

**Theorem 2.3** (Self-tuning property of $v_*$). Suppose $n \geq az^2$. Then, for any $\gamma \in [0, 1)$, we have $v_* > 0$ and

$$\frac{(1-\gamma)\sigma_{\varphi\tau_*^2}^2}{2a} \leq v_*^2 \leq \frac{\sigma^2}{2a},$$

where $\varphi = \gamma/(1-\gamma)$ and $\tau_* = v_*\sqrt{n}/z$. Moreover $\lim_{n\to\infty} v_*^2 = \sigma^2/(2a)$.

The theorem above shows that when $n \geq az^2$, the oracle $v_*^2$ automatically adapts to the unknown variance, which is sandwiched between the scaled truncated variance $\sigma_{\varphi\tau_*^2}^2/(2a)$ and the scaled variance $\sigma^2/(2a)$. By the dominated convergence theorem, $\sigma_{\varphi\tau_*^2}^2$ converges to $\sigma^2$ as $\varphi\tau_*^2$ approaches to $\to \infty$. As the sample size $n$ grows, $\sigma_{\varphi\tau_*^2}^2$ closely approximates $\sigma^2$, thus placing $v_*^2$ between $(1-\gamma)\sigma^2/(2a)$ and $\sigma^2/(2a)$. An asymptotic analysis reveals that $\lim_{n\to\infty} v_*^2 = \sigma^2/(2a)$. Taking $a = 1/2$ yields $\lim_{n\to\infty} v_*^2 = \sigma^2$, indicating that the oracle $v_*^2$ with $a = 1/2$ should approximate the true variance. This observation suggests the optimality of choosing $a = 1/2$, a choice that is assumed throughout the rest of this paper.

Our next result establishes that the proposed empirical loss function is jointly convex in both $\mu$ and $v$. This property enables the application of standard first-order optimization algorithms, facilitating the efficient computation of the global optimum.

**Proposition 2.4** (Joint convexity). The empirical loss function $L_n(\mu, v)$ in (2.5) is jointly convex in both $\mu$ and $v$. Furthermore, if there exist at least two distinct data points, the empirical loss function is strictly convex in both $\mu$ and $v$ provided that $v > 0$.

Lastly, it was brought to our attention that our formulation (2.5) shares similarities with the concomitant estimator by Ronchetti & Huber (2009):

$$\underset{\mu,v}{\operatorname{argmin}} \left\{ \frac{1}{n} \sum_{i=1}^{n} \rho\left( \frac{y_i - \mu}{v} \right) v + av \right\},$$

where $\rho$ represents any loss function, and $a$ is a user-specified constant. A notable gap in the literature is the lack of rigorous guidance on selecting the hyperparameter $a$. Driven by the goal of developing

computationally and statistically efficient robust estimators with improved finite-sample performance for handling potentially heavy-tailed data, we have derived a comparable but distinct formulation, underpinned by rigorously determined hyperparameter $a$ and an additional confidence parameter $z$. In other words, our empirical loss function $L_n$ is a meticulously adapted version of the aforementioned loss function. Specifically, we adopt the smooth pseudo-Huber loss, and set the robustification parameter $\tau$ to $\tau = v\sqrt{n}/z$ to ensure the sub-Gaussian performance of the mean estimator, where $z$ is a carefully chosen confidence parameter. Concurrently, optimal adjustment factor is identified as $a = 1/2$.

## 3 THEORETICAL PROPERTIES

This section presents the self-tuning property for the estimated robustification parameter and then the improved finite-sample property of the self-tuned mean estimator, similar to concentration property in (1.3). We further show that our proposed estimator is asymptotically efficient, thus distinguishing it from the MoM estimator. Recall $a = 1/2$.

### 3.1 THE SELF-TUNING PROPERTY AND THE IMPROVED FINITE-SAMPLE PERFORMANCE

To study the self-tuning property, we need an additional constraint that $v_0 \leq v \leq V_0$, and consider the constrained empirical risk minimization problem

$$\{\widehat{\mu}, \widehat{v}\} = \underset{\mu, \, v_0 \leq v \leq V_0}{\operatorname{argmin}} \left\{ L_n(\mu, v) := \frac{1}{n} \sum_{i=1}^{n} \ell^{\mathrm{p}}(y_i - \mu, v) \right\}. \tag{3.1}$$

It is important to note that when $v$ is either $0$ or $\infty$, the loss function is non-smooth or trivial, respectively. Moreover, the loss function is not strongly convex in $\mu$ in either case, and strong convexity is essential for our theoretical analysis. Recall that $\tau_{v_0} = v_0\sqrt{n}/z$.

**Theorem 3.1** (Self-tuning property). Assume that $n$ is sufficiently large. Suppose $v_0 < c_0 \left( \sigma_{\tau_{v_0}^2/2-1} / \sigma_{\tau_{v_0}^2/2} \wedge 1 \right) \sigma_{\tau_{v_0}^2/2-1} \leq C_0\sigma < V_0$ where $c_0$ and $C_0$ are some universal constants. For any $0 < \delta < 1$, let $z^2 \geq \log(5/\delta)$. Then, with probability at least $1 - \delta$, we have

$$c_0 \left( \sigma_{\tau_{v_0}^2/2-1} / \sigma_{\tau_{v_0}^2/2} \wedge 1 \right) \sigma_{\tau_{v_0}^2/2-1} \leq \widehat{v} \leq C_0\sigma.$$

The theorem above suggests that $\widehat{v}$ automatically adapts to the unknown standard deviation, converging to $\sigma$ if $\sigma_{\tau_{v_0}^2/2-1}$ approximates $\sigma$. This convergence is expected for large sample sizes due to the dominated convergence theorem. However, it is important to recognize that $\sigma_{\tau_{v_0}^2}$ may not approach $\sigma$ at any predictable rate under the minimal assumption of bounded variances for the data. With the above self-tuning property, we can now characterize the finite-sample property of the self-tuned mean estimator $\widehat{\mu}(\widehat{v})$.

**Theorem 3.2** (Self-tuned mean estimators). Assume that $n$ is sufficiently large. Suppose $v_0 < c_0 \left( \sigma_{\tau_{v_0}^2/2-1} / \sigma_{\tau_{v_0}^2/2} \wedge 1 \right) \sigma_{\tau_{v_0}^2/2-1} \leq C_0\sigma < V_0$ where $c_0$ and $C_0$ are some universal constants. For any $0 < \delta < 1$, take $z^2 = \log(n/\delta)$. Then, with probability at least $1 - \delta$, we have

$$|\widehat{\mu}(\widehat{v}) - \mu^*| \leq C \cdot \sigma \sqrt{\frac{\log(n/\delta)}{n}}$$

where $C$ is some constant.

The above result asserts that the self-tuned mean estimator $\widehat{\mu} = \widehat{\mu}(\widehat{v})$ achieves logarithmic deviation from the true mean $\mu^*$ with respect to $1/\delta$, akin to (1.3) (up to logarithmic terms). This is in sharp contrast to the sample mean estimator, which only achieves polynomial dependence on $1/\delta$; see (1.2). For practical applications, we suggest choosing $\delta = 0.05$, which corresponds to a failure probability of 0.05 or, equivalently, a confidence level of 0.95.

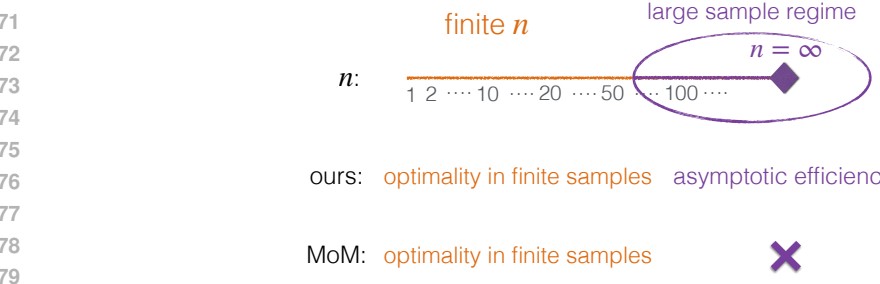

Figure 1: Comparing our self-tuned estimator with the MoM estimator in terms of adaptivity.

## 3.2 ASYMPTOTIC EFFICIENCY

Our next theorem shows that our proposed estimator achieves asymptotic efficiency.

**Theorem 3.3** (Asymptotic efficiency). Fix any $\iota \in (0, 1]$. Assume $\mathbb{E}\varepsilon_i^{2+\iota} < \infty$ and the same assumptions as in Theorem 3.1. Take any $z^2 \geq 2\log(n)$. Then

$$\sqrt{n}\left(\widehat{\mu}(\widehat{v}) - \mu^*\right) \rightsquigarrow \mathcal{N}\left(0, \sigma^2\right).$$

We provide an intuitive explanation for the optimal performance of our self-tuned estimator in the asymptotic regime. Because this estimator is a self-tuned version of the pseudo-Huber estimator in (2.2), we discuss the latter for simplicity. As per Theorem 2.1, taking $\tau = \sigma\sqrt{n/\log(1/\delta)}$ guarantees the sub-Gaussian performance of $\widehat{\mu}(\tau)$ in the finite-sample regime. Meanwhile, as $n$ approaches infinity, $\tau = \sigma\sqrt{n/\log(1/\delta)}$ also grows to infinity, causing the pseudo-Huber loss to approach the least squares loss. This loss corresponds to the negative log-likelihood of Gaussian distributions, and its minimization yields an asymptotically efficient mean estimator.

In summary, our self-tuned estimator can achieve optimal performance in both finite-sample and large-sample regimes. We point out that the large-sample regime is used to approximate the regime when the sample size is relatively large instead of describing the case of $n = \infty$. We will refer to this ability as *adaptivity to both finite-sample and large-sample regimes*, or simply *adaptivity*. As we will see in the next section, the MoM estimator does not naturally possess this adaptivity due to its discontinuous nature. Figure 1 provides a comparison between our self-tuned estimator and the MoM estimator in terms of adaptivity.

## 3.3 COMPARING WITH MoM

Other than the ERM-based approach, the median-of-means technique (Lugosi & Mendelson, 2019a) is another method to construct robust estimators under heavy-tailed distributions. The MoM mean estimator is constructed as follows:

1. Partition $[n] = \{1, \ldots, n\}$ into $k$ blocks $\mathcal{B}_1, \ldots, \mathcal{B}_k$, each with size $|\mathcal{B}_i| \geq \lfloor n/k \rfloor \geq 2$;
2. Compute the sample mean in each block $z_j = \sum_{i \in \mathcal{B}_j} x_i / |\mathcal{B}_j|$;
3. Obtain the MoM mean estimator by taking the median of $z_j$'s:

$$\widehat{\mu}^{\mathrm{MoM}} = \mathrm{med}(z_1, \ldots, z_k)$$

   where $\mathrm{med}(\cdot)$ represents the median operator.

For simplicity and without loss of generality, we assume that $n$ is divisible by $k$ so that each block has exactly $m = n/k$ elements. The following theorem is taken from Lugosi & Mendelson (2019a).

**Theorem 3.4** (Theorem 2 by Lugosi & Mendelson (2019a) ). For any $\delta \in (0, 1)$, if $k = \lceil 8\log(1/\delta) \rceil$, then, with probability at least $1 - \delta$,

$$\left|\widehat{\mu}^{\mathrm{MoM}} - \mu^*\right| \leq \sigma\sqrt{\frac{32\log(1/\delta)}{n}}.$$

The theorem above indicates that, to obtain a sub-Gaussian mean estimator, we only need to choose $k = \lceil 8 \log(1/\delta) \rceil$ when constructing the MoM mean estimator. Thus, the MoM estimator is naturally tuning-free. However, in our numerical experiments, we observed that the MoM estimator often underperforms compared to our proposed estimator. To shed light on this observation, we adopt an asymptotic viewpoint and calculate the relative efficiency of $\widehat{\mu}^{\mathrm{MoM}}$ with respect to our estimator $\widehat{\mu}(\widehat{\tau})$. The following result is a direct consequence of (Minsker, 2019, Theorem 4) and we collect the proof in the appendix for completeness.

**Proposition 3.5** (Asymptotic inefficiency of MoM estimator)**.** Fix any $\iota \in (0, 1]$. Assume $\mathbb{E}|y_i - \mu^*|^{2+\iota} < \infty$. Suppose $k \to \infty$ and $k = o\big(n^{\iota/(1+\iota)}\big)$, then

$$\sqrt{n}\,\big(\widehat{\mu}^{\mathrm{MoM}} - \mu^*\big) \rightsquigarrow \mathcal{N}\left(0, \frac{\pi}{2}\sigma^2\right).$$

We highlight that the MoM mean estimator shares the same asymptotic property as the median estimator (Van der Vaart, 2000) due to taking the median operation in the last step, and thus is asymptotically inefficient. In contrast, our estimator achieves full asymptotic efficiency. The relative efficiency $e_{\mathrm{r}}$ of the MoM estimator with respect to our estimator is

$$e_{\mathrm{r}}\big(\widehat{\mu}^{\mathrm{MoM}}, \widehat{\mu}(\widehat{v})\big) = \frac{2}{\pi} \approx 0.64.$$

This indicates that our proposed estimator outperforms the MoM estimator in terms of asymptotic performance, partially explaining the empirical success of our method; see the numerical results in Section 4 for details.

## 4 Numerical studies

This section examines numerically the finite-sample performance of our proposed robust mean estimator when dealing with heavy-tailed data. Throughout our numerical examples, we take $z = \sqrt{\log(n/\delta)}$ with $\delta = 0.05$ as recommended by Theorem 3.2. This choice guarantees that the result stated in the theorem holds with a probability of at least $0.95$.

We investigate the robustness and efficiency of our proposed estimator under two distinct distribution settings for the random variable $y$:

1. Normal distribution $\mathcal{N}(\mu, \sigma^2)$ with mean $\mu = 0$ and variance $\sigma^2 \geq 1$.
2. Skewed generalized $t$ distribution $\mathsf{sgt}(\mu, \sigma, \lambda, p, q)$, where mean $\mu = 0$, skewness $\lambda = 0.75$, standard deviation $\sigma = \sqrt{q/(q-2)}$, shape parameter $p = 2$, and shape parameter $q > 2$.

For each of the above settings, we generate an independent samples of size $n = 100$ and compute four mean estimators: our proposed estimator (ours), the sample mean estimator (sample mean), the MoM mean estimator (MoM), and the trimmed mean estimator (trimmed mean).

Figure 2 displays the $\alpha$-quantile of the estimation error $\|\widehat{\mu} - \mu\|_2^2$, with $\alpha$ ranging from 0.5 to 0.99, based on 1000 simulations for both distributional settings. For Settings 1 (normal distribution) and 2 (skewed generalized $t$ distribution), we set $\sigma^2 = 1$ and $q = 2.5$, respectively. In the case of normal distributions, our proposed estimator performs almost identically to the sample mean estimator, both of which outperform the MoM and trimmed mean estimator. Since the sample mean estimator is optimal for Gaussian data, this suggests that our estimator does not sacrifice statistical efficiency when applied to Gaussian data. In the case of heavy-tailed skewed generalized $t$ distributions, the estimation error of the sample mean estimator grows rapidly with increasing $\alpha$. This contrasts with the three robust estimators: our estimator, the MoM mean estimator, and the trimmed mean estimator. Our estimator consistently outperforms the others in both settings.

Figure 3 examines the 99%-quantile of the estimation error versus a distribution parameter, based on 1000 simulations. For Gaussian data, the distribution parameter is $\sigma$, and we vary $\sigma$ from 1 to 4 in increments of 0.1. For skewed generalized $t$ distributions, the distribution parameter is $q$, and we vary $q$ from 2.5 to 4 in increments of 0.1. For Gaussian data, our estimator performs identically to the optimal sample mean estimator, with both outperforming the MoM and trimmed mean estimators. In the case of skewed generalized $t$ distributions with $q \leq 3$, all three robust mean estimators either

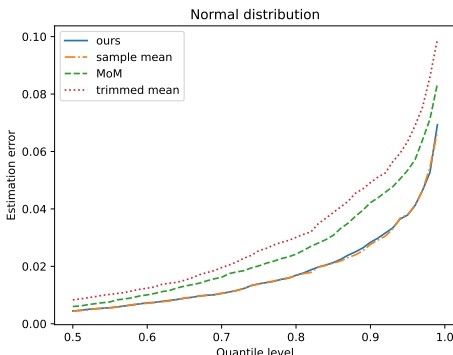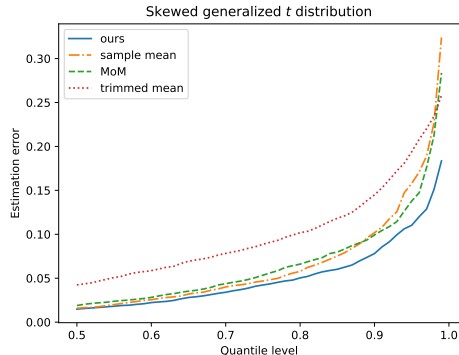

Figure 2: The $\alpha$-quantile of the estimation error (estimation error, $y$-axis) versus $\alpha$ (quantile level, $x$-axis) for our estimator, the sample mean estimator, the MoM estimator, and the trimmed mean estimator.

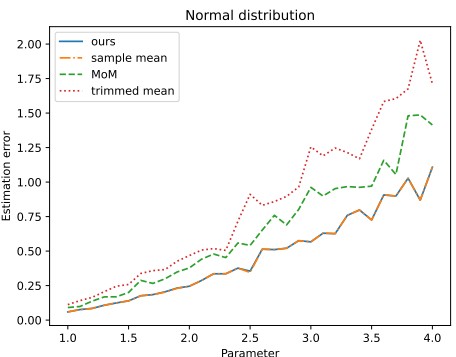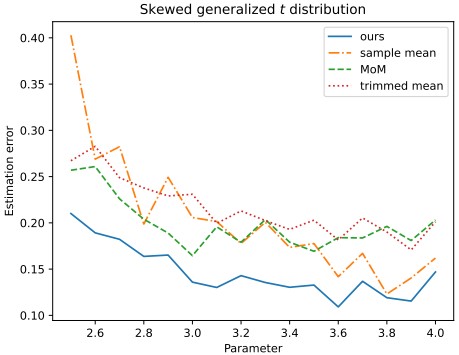

Figure 3: Empirical 99%-quantile of the estimation error (estimation error, $y$-axis) versus a distributution parameter (parameter, $x$-axis) for our estimator, the sample mean estimator, the MoM estimator and the trimmed mean estimator. The distribution parameter is $\sigma$ for normal distribution and $q$ for skewed generalized $t$ distribution.

outperform or are as competitive as the sample mean estimator. However, when $q > 3$, the sample mean estimator starts to outperform both the MoM and trimmed mean estimators. Our proposed estimator, on the other hand, consistently outperforms all other methods across the entire range of parameter values.

We also conduct a computational performance comparison of our self-tuned method with pseudo-Huber loss + cross-validation, and pseudo-Huber loss + Lepski's method. For cross-validation, we pick the best $\tau$ from a list of candidates $\{1, 2, \ldots, 100\}$ using 10-fold cross-validation. In the case of Lepski's method, we follow the appendix and choose $V = 2$, $\rho = 1.2$, and $s = 50$. We run 1000 simulations for the mean estimation problem in Setting 1 with $\sigma^2 = 1$ and a sample size of $n = 100$. All computations are performed on a MacBook Pro with an Apple M1 Max processor and 64 GB of memory. The runtimes for our self-tuning approach, Lepski's method, and cross validation are 1.5, 16.7, and 133.5 seconds, respectively. Our proposed method is approximately $90\times$ faster than cross-validation and about $10\times$ faster than Lepski's method.

Finally, we compare their statistical performance in both settings while varying the distribution parameter in the same manner as in Figure 3. The results are summarized in Figure 4 and Figure 5. In both figures, our method and cross-validation exhibit similar performance, with both outperforming Lepski's method. We suspect this is because Lepski's method depends on additional hyperparameters, and our chosen values may not be optimally tuned. This observation also suggests that, despite its sound theoretical underpinnings, Lepski's method does not uniformly yield strong empirical results.

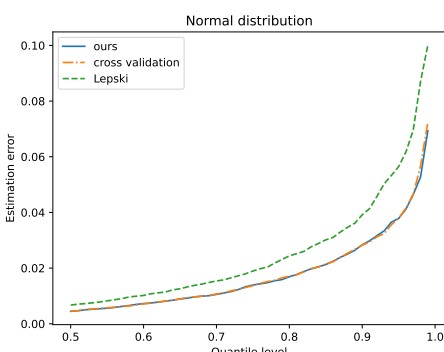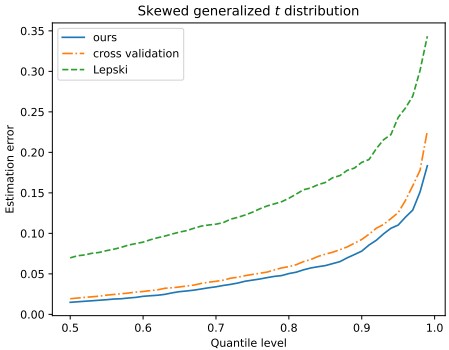

Figure 4: The $\alpha$-quantile of the estimation error (estimation error, $y$-axis) versus $\alpha$ (quantile level, $x$-axis) for our estimator, cross validation and Lepski's method.

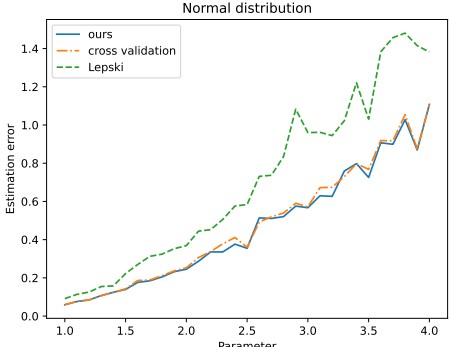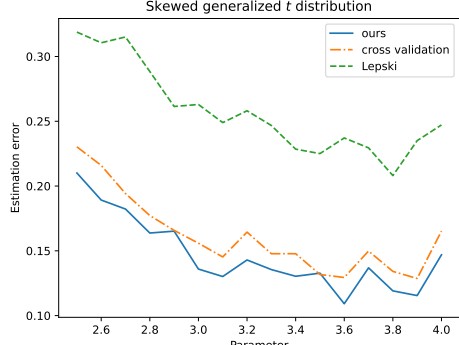

Figure 5: The empirical 99%-quantile of the estimation error (estimation error, $y$-axis) versus a distributution parameter (parameter, $x$-axis) for our estimator, cross validation and Lepski's method.

## 5 CONCLUSIONS AND LIMITATIONS

In summary, the most attractive feature of our method is its self-tuning property, incurring much lower computational cost than cross-validation and Lepski's method. This is particularly important for large-scale inference with a myriad of parameters to be tuned. Statistically, our estimator is as (statistically) efficient as the sample mean estimator for normal distributions and more efficient than popular robust alternatives for asymmetric and/or heavy-tailed distributions.

**Limitation** One limitation of our self-tuned estimator is that its finite-sample performance depends on unknown constants, making it challenging to compute the sample complexity in advance for a fixed confidence level. Another limitation of this study is its scope. We primarily focus on robust mean estimators as they represent the simplest case, and the proofs are already quite complex. Nevertheless, our approach can potentially extend to more general settings, such as regression and matrix estimation problems.

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

# Appendix

## Table of Contents

## A  BASIC FACTS

This section collects some basic facts concerning the loss function. First, as we state in Section 2, the pseudo-Huber loss (2.1) exhibits behavior similar to the Huber loss (Huber, 1964), approximating $x^2/2$ when $x^2 \lesssim \tau^2$ and resembling a straight line with slope $\tau$ when $x^2 \gtrsim \tau^2$. To see this, some algebra yields

$$
\begin{cases}
\frac{\epsilon^2 - 2(1+\epsilon)}{2\epsilon^2} x^2 \le \ell_\tau(x) \le \frac{x^2}{2}, & \text{if } x^2 \le \tau^2 \cdot 4(1+\epsilon)/\epsilon^2, \\
\frac{\tau|x|}{1+\epsilon} \le \ell_\tau(x) \le \tau|x|, & \text{if } x^2 > \tau^2 \cdot 4(1+\epsilon)/\epsilon^2.
\end{cases}
$$

Second, we give the first-order derivatives and the Hessian matrix for the empirical loss function. Let $\tau = v\sqrt{n}/z$ throughout the appendix. Recall that our empirical loss function is

$$
L_n(\mu, v) = \frac{1}{n} \sum_{i=1}^n \ell(y_i - \mu, v) = \frac{1}{n} \sum_{i=1}^n \left\{ \frac{\sqrt{n}}{z} \sqrt{\frac{nv^2}{z^2} + (y_i - \mu)^2} - \left( \frac{n}{z^2} - a \right) v \right\}
$$

$$
= \frac{1}{n} \sum_{i=1}^n \left\{ \frac{\sqrt{n}}{z} \left( \sqrt{\tau^2 + (y_i - \mu)^2} - \tau \right) + a \cdot \frac{\tau}{z\sqrt{n}} \right\}.
$$

---

**Algorithm 1** An alternating gradient descent algorithm.

---
**Input**: $\mu_{\text{init}}, v_{\text{init}}, v_0, V_0, \eta_1, \eta_2, (y_1, \ldots, y_n)$
**for** $k = 0, 1, \ldots$ until convergence **do**
$\quad \mu_{k+1} = \mu_k - \eta_1 \nabla_\mu L_n(\mu_k, v_k)$
$\quad \widetilde{v}_{k+1} = v_k - \eta_2 \nabla_\tau L_n(\mu_{k+1}, v_k)$ and $v_{k+1} = \min\{\max\{\widetilde{v}_{k+1}, v_0\}, V_0\}$
**end for**
**Output**: $\widehat{\mu} = \mu_{k+1}, \widehat{v} = v_{k+1}$

---

The first-order and second-order derivatives of $L_n(\mu, v)$ are

$$\nabla_\mu L_n(\mu, v) = -\frac{1}{n} \sum_{i=1}^n \frac{y_i - \mu}{v\sqrt{1 + z^2(y_i - \mu)^2/(nv^2)}} = -\frac{\sqrt{n}}{z} \cdot \frac{1}{n} \sum_{i=1}^n \frac{y_i - \mu}{\sqrt{\tau^2 + (y_i - \mu)^2}},$$

$$\nabla_v L_n(\mu, v) = \frac{1}{n} \sum_{i=1}^n \frac{n/z^2}{\sqrt{1 + z^2(y_i - \mu)^2/(nv^2)}} - \left(\frac{n}{z^2} - a\right) = \frac{n}{z^2} \cdot \frac{1}{n} \sum_{i=1}^n \left(\frac{\tau}{\sqrt{\tau^2 + (y_i - \mu)^2}} - 1\right) + a$$

where $a = 1/2$. The Hessian matrix is

$$H(\mu, v) = \begin{bmatrix} \frac{\sqrt{n}}{z} \frac{1}{n} \sum_{i=1}^n \frac{\tau^2}{\left(\tau^2 + (y_i - \mu)^2\right)^{3/2}} & \frac{n}{z^2} \frac{1}{n} \sum_{i=1}^n \frac{\tau(y_i - \mu)}{(\tau^2 + (y_i - \mu)^2)^{3/2}} \\ \frac{n}{z^2} \frac{1}{n} \sum_{i=1}^n \frac{\tau(y_i - \mu)}{(\tau^2 + (y_i - \mu)^2)^{3/2}} & \frac{n^{3/2}}{z^3} \frac{1}{n} \sum_{i=1}^n \frac{(y_i - \mu)^2}{(\tau^2 + (y_i - \mu)^2)^{3/2}} \end{bmatrix}.$$

# B AN ALTERNATING GRADIENT DESCENT ALGORITHM

This section presents an alternating gradient descent algorithm to optimize (3.1). The algorithm generates the solution sequence $\{(\mu_k, v_k) : k \geq 0\}$ with the initialization $(\mu_0, v_0) = (\mu_{\text{init}}, v_{\text{init}})$. At the working solution $(\mu_k, v_k)$ for any $k \geq 0$, the $(k + 1)$-th iteration involves the following two steps:

1. $\mu_{k+1} = \mu_k - \eta_1 \nabla_\mu L_n(\mu_k, v_k)$,
2. $\widetilde{v}_{k+1} = v_k - \eta_2 \nabla_\tau L_n(\mu_{k+1}, v_k)$ and $v_{k+1} = \min\{\max\{\widetilde{v}_{k+1}, v_0\}, V_0\}$,

where $\eta_1$ and $\eta_2$ are the learning rates and

$$\nabla_\mu L_n(\mu, v) = -\frac{1}{n} \sum_{i=1}^n \frac{y_i - \mu}{v\sqrt{1 + z^2(y_i - \mu)^2/(nv^2)}},$$

$$\nabla_v L_n(\mu, v) = \frac{1}{n} \sum_{i=1}^n \frac{n/z^2}{\sqrt{1 + z^2(y_i - \mu)^2/(nv^2)}} - \left(\frac{n}{z^2} - a\right).$$

The above two steps are repeated until convergence. The algorithm routine is summarized in Algorithm 1. The learning rates $\eta_1$ and $\eta_2$ can be chosen adaptively in practice. In our experiments, we utilize alternating gradient descent with the Barzilai and Borwein method and backtracking line search.

# C COMPARING WITH LEPSKI'S METHOD

We compare our method with Lepski's method. Specifically, we employ Lepski's method to tune the robustification parameter $v$ and, consequently $\tau = v\sqrt{n}/z$, in the empirical pseudo-Huber loss:

$$L_n^h(\mu, v) := \frac{1}{n} \sum_{i=1}^n \left(\tau\sqrt{\tau^2 + (y_i - \mu)^2} - \tau^2\right).$$

Lepski's method proceeds as follows. Let $v_{\max}$ be an upper bound for $\sigma$, and $\tau_{\max} = v_{\max}\sqrt{n}/z$ with $z = \sqrt{\log(1/\delta)}$. Let $n$ be sufficiently large. Then with probability at least $1 - \delta$, we have

$$|\widetilde{\mu}(v_{\max}) - \mu^*| \leq 6v_{\max}\sqrt{\frac{\log(4/\delta)}{n}} =: \epsilon(v_{\max}, \delta),$$

where $\widetilde{\mu}(v_{\max}) = \mathrm{argmin}_\mu L_n(\mu, v_{\max})$. Let us by convention set $\epsilon(v_{\max}, 0) = +\infty$. Clearly, $\epsilon(v_{\max}, \delta)$ is homogeneous in the sense that

$$\epsilon(v_{\max}, \delta) = B(\delta) v_{\max}, \quad \text{where } B(\delta) = 6\sqrt{\frac{\log(4/\delta)}{n}}.$$

For some parameters $V \in \mathbb{R}$, $\rho > 1$, and $s \in \mathbb{N}$, we choose the following probability measure $\mathcal{V}$ for $v_{\max}$

$$\mathcal{V}(v_{\max}) = \begin{cases} 1/(2s+1), & \text{if } v_{\max} = V\rho^k, \ k \in \mathbb{Z}, \ |k| \le s, \\ 0, & \text{otherwise.} \end{cases}$$

Let us consider for any $v_{\max}$ such that $\epsilon(v_{\max}, \delta \mathcal{V}(v_{\max})) < \infty$ the confidence interval

$$I(v_{\max}) = \widetilde{\mu}(v_{\max}) + \epsilon(v_{\max}, \delta \, \mathcal{V}(v_{\max})) \times [-1, 1],$$

where

$$\epsilon(v_{\max}, \delta \, \mathcal{V}(v_{\max})) = 6 v_{\max} \sqrt{\frac{\log(4/\delta) + \log(2s+1)}{n}}$$

if $v_{\max} = V\rho^k$ for any $k \in \mathbb{Z}$ and $|k| \le s$. We set $I(v_{\max}) = \mathbb{R}$ when $\epsilon(v_{\max}, \delta \mathcal{V}(v_{\max})) = +\infty$.

Let us consider the non-decreasing family of closed intervals

$$J(v_1) = \bigcap \{I(v_{\max}) : v_{\max} \ge v_1\}, \ v_1 \in \mathbb{R}_+.$$

In this definition, we can restrict the intersection to the support of $\mathcal{V}$, since otherwise $I(v_{\max}) = \mathbb{R}$. Lepski's method picks the center point of the intersection

$$\bigcap \{J(v_1) : v_1 \in \mathbb{R}_+, \ J(v_1) \ne \emptyset\}$$

to be the final estimator $\widehat{\mu}_{\mathrm{Lepski}}$. Then the following result is due to Catoni (2012).

**Proposition C.1.** Suppose $|\log(\sigma/V)| \le 2s \log(\rho)$. Then with probability at least $1 - \delta$

$$|\widehat{\mu}_{\mathrm{Lepski}} - \mu^*| \le 12\rho\sigma \sqrt{\frac{\log(4/\delta) + \log(2s+1)}{n}}.$$

If we take the grid fine enough such that $s = n$, then the upper bound above reduces to

$$12\rho\sigma \sqrt{\frac{\log(4/\delta) + \log(2n+1)}{n}},$$

which agrees with deviation bound for our proposed estimator, up to a constant multiplier. Therefore, our proposed estimator is comparable to Lepski's method in terms of the deviation upper ound. Computationally, our estimator is self-tuned and thus computationally more efficient than Lepski's method; detailed numerical results can be found in Section 4.

# D PROOFS FOR SECTION 2

## D.1 PROOFS FOR THEOREM 2.3

*Proof of Theorem 2.3.* We prove first the finite-sample result and then the asymptotic result. Recall that $\tau_* = v_* \sqrt{n}/z$.

**Proving the finite-sample result.** On one side, if $v_* = 0$ and by the definition of $v_*$, $v_*$ satisfies

$$1 - \frac{az^2}{n} = \mathbb{E} \frac{\sqrt{n} v_*}{\sqrt{n v_*^2 + z^2 \varepsilon^2}} = 0,$$

which is a contradiction. Thus $v_* > 0$. Using the convexity of $1/\sqrt{1+x}$ for $x > -1$ and Jensen's inequality acquires

$$1 - \frac{az^2}{n} = \mathbb{E}\frac{\sqrt{n}v_*}{\sqrt{nv_*^2 + z^2\varepsilon^2}} = \mathbb{E}\frac{1}{\sqrt{1 + z^2\varepsilon^2/(nv_*^2)}} \geq \frac{1}{\sqrt{1 + z^2\sigma^2/(nv_*^2)}} \geq 1 - \frac{z^2\sigma^2}{2nv_*^2},$$

where the last inequality uses the inequality $(1+x)^{-1/2} \geq 1 - x/2$, i.e., Lemma H.4 (i) with $r = -1/2$. This implies

$$v_*^2 \leq \frac{\sigma^2}{2a}.$$

On the other side, using the concavity of $\sqrt{x}$, we obtain, for any $\gamma \in [0, 1)$, that

$$
\begin{aligned}
1 - \frac{az^2}{n} &= \mathbb{E}\frac{\sqrt{n}v_*}{\sqrt{nv_*^2 + z^2\varepsilon^2}} = \mathbb{E}\frac{1}{\sqrt{1 + \sigma^2 z^2\varepsilon^2/(nv_*^2)}} \\
&\leq \sqrt{\mathbb{E}\left(\frac{1}{1 + z^2\varepsilon^2/(nv_*^2)}\right)} \\
&\leq \sqrt{\mathbb{E}\left\{\left(1 - (1-\gamma)\frac{z^2\varepsilon^2}{nv_*^2}\right)1\left(\frac{z^2\varepsilon^2}{nv_*^2} \leq \frac{\gamma}{1-\gamma}\right) + \frac{1}{1 + z^2\varepsilon^2/(nv_*^2)}1\left(\frac{z^2\varepsilon^2}{nv_*^2} > \frac{\gamma}{1-\gamma}\right)\right\}} \\
&\leq \sqrt{1 - (1-\gamma)\mathbb{E}\left\{\frac{z^2\varepsilon^2}{nv_*^2}1\left(\frac{z^2\varepsilon^2}{nv_*^2} \leq \frac{\gamma}{1-\gamma}\right)\right\}} \\
&\leq \sqrt{1 - (1-\gamma)\frac{\mathbb{E}\{\varepsilon^2 1(\varepsilon^2 \leq \gamma\tau_*^2/(1-\gamma))\}}{nv_*^2/z^2}}, 
\end{aligned}
\tag{D.1}
$$

where the second inequality uses Lemma D.1, that is,

$$(1+x)^{-1} \leq 1 - (1-\gamma)x, \text{ for any } x \in \left[0, \frac{\gamma}{1-\gamma}\right].$$

Taking square on both sides of inequality (D.1) and using the fact that $n \geq az^2$ together with Lemma H.4 (i) with $r = 2$, aka $(1+x)^2 \geq 1 + 2x$ for $x \geq -1$, we obtain

$$1 - \frac{2az^2}{n} \leq \left(1 - \frac{az^2}{n}\right)^2 \leq 1 - (1-\gamma)\frac{\mathbb{E}\{\varepsilon^2 1(\varepsilon^2 \leq \gamma\tau_*^2/(1-\gamma))\}}{nv_*^2/z^2},$$

or equivalently

$$v_*^2 \geq \frac{\sigma_{\varphi\tau_*^2}^2}{2a},$$

where $\varphi = \gamma/(1-\gamma)$. Combining the upper bound and the lower bound for $v_*^2$ completes the proof for the finite-sample result.

**Proving the asymptotic result.** The above derivation implies that $v_* < \infty$ for any $a > 0$. By the definition of $v_*$, we obtain

$$\frac{az^2}{n} = 1 - \mathbb{E}\frac{1}{\sqrt{1 + z^2\varepsilon^2/(nv_*^2)}}. \tag{D.2}$$

We must have $nv_*^2/z^2 \to \infty$. Otherwise assume

$$\limsup_{n\to\infty} nv_*^2/z^2 \leq M < \infty.$$

Taking $n \to \infty$, the left hand side of the above equality goes to $0$ while the right hand is lower bounded as

$$1 - \mathbb{E}\frac{1}{\sqrt{1 + \varepsilon^2/M}} \geq 1 - \sqrt{\mathbb{E}\left(\frac{1}{1 + \varepsilon^2/M}\right)}$$

$$\geq 1 - \sqrt{1 - \frac{\mathbb{E}\{\varepsilon^2 1(\varepsilon^2 \leq M)\}}{2M}}$$

$$\geq 1 - \sqrt{\frac{1}{2}} > 0,$$

where the first two inequalities follow from the same arguments in deriving (D.1) but with $\gamma = 1/2$, and the third inequality uses the fact that

$$\mathbb{E}\{\varepsilon^2 1(\varepsilon^2 \leq M)\} \leq M.$$

This is a contradiction. Thus $nv_*^2/z^2 \to \infty$. Multiplying both sides of the above equality by $n$, taking $n \to \infty$, and using the dominated convergence theorem, we obtain

$$az^2 = \lim_{n \to \infty} \mathbb{E}\left(n \cdot \frac{\sqrt{1 + z^2\varepsilon^2/(nv_*^2)} - 1}{\sqrt{1 + z^2\varepsilon^2/(nv_*^2)}}\right)$$

$$= \lim_{n \to \infty} \mathbb{E}\left(n \cdot \frac{1}{\sqrt{1 + z^2\varepsilon^2/(nv_*^2)}} \cdot \frac{\sqrt{1 + z^2\varepsilon^2/(nv_*^2)} - 1}{z^2\varepsilon^2/(2nv_*^2)} \cdot \frac{z^2\varepsilon^2}{2nv_*^2}\right)$$

$$= \frac{\mathbb{E}z^2\varepsilon^2}{2\lim_{n \to \infty} v_*^2},$$

and thus $\lim_{n \to \infty} v_*^2 = \sigma^2/(2a)$. This proves the asymptotic result.

$\square$

### D.2 PROOF OF PROPOSITION 2.4

*Proof of Proposition 2.4.* The convexity proof consists of two steps: (1) proving that $L_n(\mu, v)$ is jointly convex in $\mu$ and $v$; (2) proving that $L_n(\mu, v)$ is strictly convex, provided that there are at least two distinct data points.

To show that $L_n(\mu, v) = n^{-1}\sum_{i=1}^n \ell^{\mathrm{P}}(y_i - \mu, v)$ in (2.5) is jointly convex in $\mu$ and $v$, it suffices to show that each $\ell^{\mathrm{P}}(y_i - \mu, v)$ is jointly convex in $\mu$ and $v$. Recall that $\tau = v\sqrt{n}/z$. The Hessian matrix of $\ell^{\mathrm{P}}(y_i - \mu, v)$ is

$$H_i(\mu, v) = \frac{\sqrt{n}}{z} \cdot \frac{1}{\left(\tau^2 + (y_i - \mu)^2\right)^{3/2}} \begin{bmatrix} \tau^2 & (\sqrt{n}/z)\,\tau(y_i - \mu) \\ (\sqrt{n}/z)\,\tau(y_i - \mu) & (\sqrt{n}/z)^2\,(y_i - \mu)^2 \end{bmatrix} \succeq 0,$$

and thus positive semi-definite. Therefore, $L_n(\mu, v)$ is jointly convex in $\mu$ and $v$.

We proceed to show (2). Because the Hessian matrix $H(\mu, v)$ of $L_n(\mu, v)$ satisfies $H(\mu, v) = n^{-1}\sum_{i=1}^n H_i(\mu, v)$ and each $H_i(\mu, v)$ is positive semi-definite, we only need to show that $H(\mu, v)$ is of full rank. Without generality, assume that $y_1 \neq y_2$. Then

$$H_1(\mu, v) + H_2(\mu, v) = \frac{\sqrt{n}}{z} \cdot \sum_{i=1}^2 \frac{1}{\left(\tau^2 + (y_i - \mu)^2\right)^{3/2}} \begin{bmatrix} \tau^2 & (\sqrt{n}/z)\,\tau(y_i - \mu) \\ (\sqrt{n}/z)\,\tau(y_i - \mu) & (\sqrt{n}/z)^2\,(y_i - \mu)^2 \end{bmatrix}.$$

Some algebra yields

$$\det\left(H_1(\mu, v) + H_2(\mu, v)\right) = \frac{n^2\tau^2}{z^4} \cdot \frac{(y_1 - y_2)^2}{(\tau^2 + (y_1 - \mu)^2)^{3/2}(\tau^2 + (y_2 - \mu)^2)^{3/2}} \neq 0$$

for any $\tau > 0$ ($v > 0$), and $\mu \in \mathbb{R}$, provided that $y_1 \neq y_2$. Therefore, $H_1(\mu, v) + H_2(\mu, v)$ is of full rank and thus is $H(\mu, \tau)$, provided $v > 0$, $\mu \in \mathbb{R}$, and $y_1 \neq y_2$.

$\square$

## D.3 Supporting lemmas

**Lemma D.1.** Let $0 \leq \gamma < 1$. For any $0 \leq x \leq \gamma/(1-\gamma)$, we have

$$(1+x)^{-1} \leq 1 - (1-\gamma)x.$$

*Proof of Lemma D.1.* To prove the lemma, it suffices to show, for any $\gamma \in [0,1)$, that

$$1 \leq (1+x) - (1-\gamma)x(1+x), \quad \forall 0 \leq x \leq \frac{\gamma}{1-\gamma},$$

which is equivalently to

$$x\left(x - \frac{\gamma}{1-\gamma}\right) \leq 0, \quad \forall 0 \leq x \leq \frac{\gamma}{1-\gamma}.$$

The above inequality always holds, and this completes the proof.

$\square$

# E Results and proofs for the fixed $v$ case

This section presents the theoretical results concerning the minimizer of the empirical penalized pseudo-Huber loss in (2.5) with $v$ fixed, aka Theorem E.2 and Corollary E.4, and their proofs. Corollary E.4 is a rigorous version of the informal result, aka Theorem 2.1, in Section 2.

## E.1 Results for the fixed $v$ case

With an abuse of notation, we use $\widehat{\mu}(v)$ to denote the minimizer of the empirical penalized pseudo-Huber loss in (2.5) with $v$ fixed. Recall that we have used $\widehat{\mu}(\tau)$ to denote the minimizer of the empirical pseudo-Huber loss in (2.2), and $\widehat{\mu}(v)$ is equivalent to $\widehat{\mu}(\tau)$ with $\tau = v\sqrt{n}/z$. We begin by examining the theoretical properties of $\widehat{\mu}(v)$. We require the following locally strong convexity assumption, which will be verified later in this subsection.

**Assumption E.1** (Locally strong convexity in $\mu$)**.** The empirical Hessian matrix is locally strongly convex with respect to $\mu$ such that, for any $\mu \in \mathbb{B}_r(\mu^*) := \{\mu : |\mu - \mu^*| \leq r\}$,

$$\inf_{\mu \in \mathbb{B}_r(\mu^*)} \frac{\langle \nabla_\mu L_n(\mu, v) - \nabla_\mu L_n(\mu^*, v), \mu - \mu^* \rangle}{|\mu - \mu^*|^2} \geq \kappa_\ell > 0$$

where $r > 0$ is a local radius parameter.

**Theorem E.2.** For any $0 < \delta < 1$, let $v > 0$ be fixed and $z^2 = \log(1/\delta)$. Assume Assumption E.1 holds with any $r \geq r_0(\kappa_\ell) := \kappa_\ell^{-1} \left(\sigma/(\sqrt{2}v) + 1\right)^2 \sqrt{\log(2/\delta)/n}$. Then, with probability at least $1 - \delta$, we have

$$|\widehat{\mu}(v) - \mu^*| < \frac{1}{\kappa_\ell} \left(\frac{\sigma}{\sqrt{2}v} + 1\right)^2 \sqrt{\frac{\log(2/\delta)}{n}} = \frac{C}{\kappa_\ell} \sqrt{\frac{\log(2/\delta)}{n}},$$

where $C = (\sigma/(\sqrt{2}v) + 1)^2$ only depends on $v$ and $\sigma$.

The above theorem states that under the assumption of locally strong convexity, $\widehat{\mu}(v)$ achieves a sub-Gaussian deviation bound when the data have only bounded variances. In particular, if we choose $v = \sigma$ in the theorem, we obtain

$$|\widehat{\mu}(\sigma) - \mu^*| \leq \frac{1}{\kappa_\ell} \left(\frac{\sigma}{\sigma} + 1\right)^2 \sqrt{\frac{\log(2/\delta)}{n}} \leq \frac{4}{\kappa_\ell} \sqrt{\frac{\log(2/\delta)}{n}}.$$

Assumption E.1 essentially requires the loss function to exhibit curvature in a small neighborhood $\mathbb{B}_r(\mu^*)$, while the penalized loss (2.4) transitions from a quadratic function to a linear function roughly at $|x| = \tau \propto \sqrt{n}$. Quadratic functions always have curvature, so intuitively, Assumption E.1 holds as long as

$$\sqrt{n} \gtrsim r \geq r_0(\kappa_\ell) \propto \sqrt{\frac{1}{n}}.$$

The condition above is automatically guaranteed when $n$ is sufficiently large. Choosing $r$ to be the smallest $r_0(\kappa_\ell)$ results in Assumption E.1 being at its weakest. In other words, in this scenario, the empirical loss function only needs to exhibit curvature in a diminishing neighborhood of $\mu^*$, approximately with a radius of $\sqrt{1/n}$. The following lemma rigorously proves this claim.

**Lemma E.3.** Suppose $v \geq v_0$. For any $0 < \delta < 1$, let $n \geq C \max\left\{z^2(\sigma^2 + r^2)/v_0^2, \log(1/\delta)\right\}$ for some absolute constant $C$. Then, with probability at least $1 - \delta$, Assumption E.1 with $\kappa_\ell = 1/(2v)$ and any local radius $r \geq r_0(\kappa_\ell) = r_0(1/(2v))$ holds uniformly over $v \geq v_0 > 0$.

The first sample complexity condition, $n \geq Cz^2(\sigma^2 + r^2)/v_0^2$, arises from the requirement that $\tau_{v_0}^2 := v_0^2 n/z^2 \geq C(\sigma^2 + r^2)$. Because the robustification parameter $\tau_{v_0} = v_0^2 n/z^2$ determines the size of the quadratic region, this requirement is minimal in the sense that Assumption E.1 can hold only when $\tau_v^2$ is larger than $r^2$ plus the noise variance $\sigma^2$. As argued before, Assumption E.1 holds with any $r$ such that $\sqrt{n} \gtrsim r \gtrsim \sqrt{1/n}$. For example, we can take $r \propto \sigma$ to be a constant, and this will not worsen the sample complexity condition. Finally, by combining Lemma E.3 and Theorem E.2, we obtain the following result.

**Corollary E.4.** Suppose $v \geq v_0$. For any $0 < \delta < 1$, let $n \geq C \max\left\{(r^2 + \sigma^2)/v_0^2, 1\right\} \log(1/\delta)$ for some universal constant $C$, where $r \geq 2r_0(1/(2v))$. Take $z^2 = \log(1/\delta)$. Then, for any $v \geq v_0$, with probability at least $1 - \delta$, we have

$$|\widehat{\mu}(v) - \mu^*| \leq 2v \left(\frac{\sigma}{\sqrt{2}v} + 1\right)^2 \sqrt{\frac{\log(4/\delta)}{n}} \lesssim v\sqrt{\frac{1 + \log(1/\delta)}{n}}.$$

This section collects proofs for Theorem E.2, Lemma E.3, and Corollary E.4. Recall that $\tau = v\sqrt{n}/z$, and the gradients with respect to $\mu$ and $v$ are

$$\nabla_\mu L_n(\mu, v) = -\frac{1}{n} \sum_{i=1}^n \frac{y_i - \mu}{v\sqrt{1 + z^2(y_i - \mu)^2/(nv^2)}} = -\frac{\sqrt{n}}{z} \cdot \frac{1}{n} \sum_{i=1}^n \frac{y_i - \mu}{\sqrt{\tau^2 + (y_i - \mu)^2}},$$

$$\nabla_v L_n(\mu, v) = \frac{1}{n} \sum_{i=1}^n \frac{n/z^2}{\sqrt{1 + z^2(y_i - \mu)^2/(nv^2)}} - \left(\frac{n}{z^2} - a\right) = \frac{n}{z^2} \cdot \frac{1}{n} \sum_{i=1}^n \left(\frac{\tau}{\sqrt{\tau^2 + (y_i - \mu)^2}} - 1\right) + a.$$

### E.2 PROOF OF THEOREM E.2

*Proof of Theorem E.2.* Because $\widehat{\mu}(v)$ is the stationary point of $L_n(\mu, v)$, we have

$$\frac{\partial}{\partial \mu} L_n(\widehat{\mu}(v), v) = -\frac{1}{n} \sum_{i=1}^n \frac{y_i - \widehat{\mu}(v)}{v\sqrt{1 + z^2(y_i - \widehat{\mu}(v))^2/(nv^2)}} = -\frac{\sqrt{n}}{z} \cdot \frac{1}{n} \sum_{i=1}^n \frac{y_i - \widehat{\mu}(v)}{\sqrt{\tau^2 + (y_i - \widehat{\mu}(v))^2}} = 0.$$

Let $\Delta = \widehat{\mu}(v) - \mu$. We first assume that $|\Delta| := |\widehat{\mu}(v) - \mu^*| \leq r_0 \leq r$. Using Assumption E.1 obtains

$$\kappa_\ell |\widehat{\mu}(v) - \mu^*|^2 \leq \left\langle \frac{\partial}{\partial \mu} L_n(\widehat{\mu}(v), v) - \frac{\partial}{\partial \mu} L_n(\mu^*, v), \widehat{\mu}(v) - \mu^* \right\rangle$$

$$\leq \left|\frac{1}{\sqrt{n}} \sum_{i=1}^n \frac{\varepsilon_i}{z\sqrt{\tau^2 + \varepsilon_i^2}}\right| |\widehat{\mu}(v) - \mu^*|,$$

or equivalently

$$\kappa_\ell |\widehat{\mu}(v) - \mu^*| \leq \left|\frac{1}{\sqrt{n}} \sum_{i=1}^n \frac{\varepsilon_i}{z\sqrt{\tau^2 + \varepsilon_i^2}}\right|.$$

Applying Lemma E.5 with the fact that $\left|\mathbb{E}\left(\tau \varepsilon_i/(\tau^2 + \varepsilon_i^2)^{1/2}\right)\right| \leq \sigma^2/(2\tau)$, we obtain with probability at least $1 - 2\delta$ that

$$\kappa_\ell |\widehat{\mu}(v) - \mu^*| \leq \left|\frac{\sqrt{n}}{\tau} \frac{1}{n} \sum_{i=1}^n \frac{\tau \varepsilon_i}{z\sqrt{\tau^2 + \varepsilon_i^2}}\right| \leq \frac{\sqrt{n}}{z\tau}\left(\sigma\sqrt{\frac{2\log(1/\delta)}{n}} + \frac{\tau \log(1/\delta)}{3n} + \frac{\sigma^2}{2\tau}\right),$$

or equivalently

$$\kappa_\ell |\widehat{\mu}(v) - \mu^*| \leq \sqrt{\frac{2\log(1/\delta)}{z^2\tau^2/\sigma^2}} + \frac{\log(1/\delta)}{3z\sqrt{n}} + \frac{\sqrt{n}\sigma^2}{2z\tau^2}.$$

Since $\tau = v\sqrt{n}/z$, we have

$$\kappa_\ell |\widehat{\mu}(v) - \mu^*| \leq \left(\frac{\sqrt{2}\sigma}{v} + \frac{\sqrt{\log(1/\delta)}}{3z}\right)\sqrt{\frac{\log(1/\delta)}{n}} + \frac{1}{2} \cdot \frac{\sigma^2}{v^2} \cdot \frac{z}{\sqrt{n}}.$$

Taking $z = \sqrt{\log(1/\delta)}$ then yields

$$\kappa_\ell |\widehat{\mu}(v) - \mu^*| \leq \left(\frac{\sqrt{2}\sigma}{v} + \frac{\sqrt{\log(1/\delta)}}{3\sqrt{\log(1/\delta)}}\right)\sqrt{\frac{\log(1/\delta)}{n}} + \frac{1}{2} \cdot \frac{\sigma^2}{v^2} \cdot \sqrt{\frac{\log(1/\delta)}{n}}$$

$$\leq \left(\frac{\sqrt{2}\sigma}{v} + \frac{1}{3} + \frac{1}{2} \cdot \frac{\sigma^2}{v^2}\right)\sqrt{\frac{\log(1/\delta)}{n}}$$

$$< \left(1 + \frac{\sigma}{\sqrt{2}v}\right)^2 \sqrt{\frac{\log(1/\delta)}{n}}$$

for any $\delta \in (0, 1/2)$. Moving $\kappa_\ell$ to the right hand side and using a change of variable $2\delta \to \delta$, we obtain

$$|\widehat{\mu}(v) - \mu^*| < \frac{1}{\kappa_\ell} \cdot \left(1 + \frac{\sigma}{\sqrt{2}v}\right)^2 \sqrt{\frac{\log(2/\delta)}{n}}$$

$$= r_0 \leq r.$$

This completes the proof, provided that $|\Delta| \leq r_0$.

Lasty, we show that $|\Delta| \leq r_0$ must hold. If not, we shall construct an intermediate solution between $\mu^*$ and $\widehat{\mu}(v)$, denoted by $\mu_\eta = \mu^* + \eta(\widehat{\mu}(v) - \mu^*)$, such that $|\mu_\eta - \mu^*| = r_0$. Specifically, we can choose some $\eta \in (0, 1)$ such that $|\mu_\eta - \mu^*| = r_0$. We then repeat the above calculation and obtain

$$|\widehat{\mu}(v) - \mu^*| \leq \frac{1}{\kappa_\ell} \cdot \left(\frac{\sqrt{2}\sigma}{v} + \frac{1}{3} + \frac{1}{2} \cdot \frac{\sigma^2}{v^2}\right)\sqrt{\frac{\log(2/\delta)}{n}}$$

$$< r_0 = \frac{1}{\kappa_\ell} \cdot \left(1 + \frac{\sigma}{\sqrt{2}v}\right)^2 \sqrt{\frac{\log(2/\delta)}{n}}$$

which is a contradiction. Therefore, it must hold that $|\Delta| \leq r_0$. $\qquad\square$

### E.3 PROOF OF LEMMA E.3

*Proof of Lemma E.3.* We first prove that, with probability at least $1 - \delta$, Assumption E.1 with $\kappa_\ell = 1/(2v)$ and radius $r$ holds for any fixed $v \geq v_0$. Recall that $\tau = v\sqrt{n}/z$. For notational simplicity, let $\Delta = \mu - \mu^*$ and $\tau_{v_0} = v_0\sqrt{n}/z$. It follows that

$$\langle \nabla_\mu L_n(\mu, v) - \nabla_\mu L_n(\mu^*, v), \Delta \rangle = \left\langle \frac{1}{\sqrt{n}}\sum_{i=1}^n \frac{\varepsilon_i}{z\sqrt{\tau^2 + \varepsilon_i^2}} - \frac{1}{\sqrt{n}}\sum_{i=1}^n \frac{y_i - \mu}{z\sqrt{\tau^2 + (y_i - \mu)^2}}, \Delta \right\rangle$$

$$= \frac{1}{\sqrt{n}}\sum_{i=1}^n \frac{\tau^2}{z(\tau^2 + (y_i - \widetilde{\mu})^2)^{3/2}}\Delta^2,$$

where $\widetilde{\mu}$ is some convex combination of $\mu^*$ and $\mu$, that is, $\widetilde{\mu} = (1 - \lambda)\mu^* + \lambda\mu$ for some $\lambda \in [0, 1]$. Obviously, we have $|\widetilde{\mu} - \mu^*| = \lambda|\Delta| \leq |\Delta| \leq r$. Since $(y_i - \widetilde{\mu})^2 \leq 2\varepsilon_i^2 + 2\lambda^2\Delta^2 \leq 2\varepsilon_i^2 + 2\Delta^2 \leq$

$2\varepsilon_i^2 + 2r^2$ the above displayed equality implies that, with probability at least $1 - \delta$,

$$\inf_{\mu \in \mathbb{B}_r(\mu^*)} \frac{\langle \nabla_\mu L_n(\mu, v) - \nabla_\mu L_n(\mu^*, v), \mu - \mu^* \rangle}{|\mu - \mu^*|^2}$$

$$\geq \frac{\sqrt{n}}{z} \cdot \frac{1}{n} \sum_{i=1}^n \frac{\tau^2}{(\tau^2 + 2r^2 + 2\varepsilon_i^2)^{3/2}}$$

$$= \frac{\sqrt{n}}{z} \cdot \frac{\tau^2}{(\tau^2 + 2r^2)^{3/2}} \cdot \frac{1}{n} \sum_{i=1}^n \frac{(\tau^2 + 2r^2)^{3/2}}{(\tau^2 + 2r^2 + 2\varepsilon_i^2)^{3/2}}$$

$$\geq \frac{\sqrt{n}}{z} \cdot \frac{\tau^2}{(\tau^2 + 2r^2)^{3/2}} \cdot \left( \mathbb{E} \frac{(\tau_{v_0}^2 + 2r^2)^{3/2}}{(\tau_{v_0}^2 + 2r^2 + 2\varepsilon_i^2)^{3/2}} - \sqrt{\frac{\log(1/\delta)}{2n}} \right)$$

$$= \frac{\sqrt{n}}{z} \cdot \frac{\tau^2}{(\tau^2 + 2r^2)^{3/2}} \cdot \left( \mathrm{I} - \sqrt{\frac{\log(1/\delta)}{2n}} \right), \tag{E.1}$$

where the last inequality uses Lemma E.6.

It remains to lower bound I. Using the convexity of $1/(1 + x)^{3/2}$ and Jensen's inequality, we obtain

$$\frac{1}{n} \sum_{i=1}^n \mathbb{E} \frac{(\tau_{v_0}^2 + 2r^2)^{3/2}}{(\tau_{v_0}^2 + 2r^2 + 2\varepsilon_i^2)^{3/2}} = \mathbb{E} \frac{(\tau_{v_0}^2 + 2r^2)^{3/2}}{(\tau_{v_0}^2 + 2r^2 + 2\varepsilon_i^2)^{3/2}}$$

$$= \mathbb{E} \frac{1}{(1 + 2\varepsilon_i^2/(\tau_{v_0}^2 + 2r^2))^{3/2}}$$

$$\geq \frac{1}{(1 + 2\sigma^2/(\tau_{v_0}^2 + 2r^2))^{3/2}}$$

$$= \frac{(\tau_{v_0}^2 + 2r^2)^{3/2}}{(\tau_{v_0}^2 + 2r^2 + 2\sigma^2)^{3/2}}.$$

Plugging the above lower bound into (E.1) and using the facts

$$\frac{\tau^3}{(\tau^2 + 2r^2)^{3/2}} \geq \frac{\tau_{v_0}^3}{(\tau_{v_0}^2 + 2r^2)^{3/2}} \ \text{ for } \tau_{v_0} \geq \tau \ \text{ and } \ \frac{\tau^3}{(\tau^2 + 2r^2)^{3/2}} \leq 1,$$

we obtain with probability at least $1 - \delta$

$$\inf_{\mu \in \mathbb{B}_r(\mu^*)} \frac{\langle \nabla_\mu L_n(\mu) - \nabla_\mu L_n(\mu^*), \mu - \mu^* \rangle}{|\mu - \mu^*|^2}$$

$$\geq \frac{\sqrt{n}}{z} \cdot \frac{\tau^2}{(\tau^2 + 2r^2)^{3/2}} \cdot \left( \frac{(\tau_{v_0}^2 + 2r^2)^{3/2}}{(\tau_{v_0}^2 + 2r^2 + 2\sigma^2)^{3/2}} - \sqrt{\frac{\log(1/\delta)}{2n}} \right)$$

$$\geq \frac{\sqrt{n}}{z\tau} \cdot \frac{\tau^3}{(\tau^2 + 2r^2)^{3/2}} \cdot \left( \frac{(\tau_{v_0}^2 + 2r^2)^{3/2}}{(\tau_{v_0}^2 + 2r^2 + 2\sigma^2)^{3/2}} - \sqrt{\frac{\log(1/\delta)}{2n}} \right)$$

$$= \frac{\sqrt{n}}{z\tau} \left( \frac{\tau^3}{(\tau^2 + 2r^2)^{3/2}} \cdot \frac{(\tau_{v_0}^2 + 2r^2)^{3/2}}{(\tau_{v_0}^2 + 2r^2 + 2\sigma^2)^{3/2}} - \frac{\tau^3}{(\tau^2 + 2r^2)^{3/2}} \cdot \sqrt{\frac{\log(1/\delta)}{2n}} \right)$$

$$\geq \frac{\sqrt{n}}{z\tau} \left( \frac{1}{(1 + (2r^2 + 2\sigma^2)/\tau_{v_0}^2)^{3/2}} - \sqrt{\frac{\log(1/\delta)}{2n}} \right)$$

$$= \frac{1}{v} \left( \frac{1}{(1 + (2r^2 + 2\sigma^2)/\tau_{v_0}^2)^{3/2}} - \sqrt{\frac{\log(1/\delta)}{2n}} \right)$$

$$\geq \frac{1}{2v}$$

provided $\tau_{v_0}^2 \geq 4r^2 + 4\sigma^2$ and $n \geq C \log(1/\delta)$ for some large enough absolute constant $C$.

Lastly, the above result holds uniformly over $v \geq v_0$ with probability at least $1 - \delta$ since the probability event does not depend on $v$.

$\square$

## E.4 PROOF OF COROLLARY E.4

*Proof of Corollary E.4.* Recall $z = \sqrt{\log(1/\delta)}$ and

$$r \geq 2v \left( \frac{\sigma}{\sqrt{2v}} + 1 \right)^2 \sqrt{\frac{\log(2/\delta)}{n}}.$$

If $n \geq C \max \left\{ (r^2 + \sigma^2)/v_0^2, 1 \right\} \log(1/\delta)$, which is guaranteed by the conditions of the corollary, then Lemma E.3 implies that, with probability at least $1 - \delta$, Assumption E.1 holds with $\kappa_\ell = 1/(2v)$ and radius $r$ uniformly over $v \geq v_0$. Denote this probability event by $\mathcal{E}$. If Assumption E.1 holds, then by Theorem E.2, we have

$$\mathbb{P} \left( |\widehat{\mu}(v) - \mu^*| \leq 2v \left( \frac{\sigma}{\sqrt{2v}} + 1 \right)^2 \sqrt{\frac{\log(2/\delta)}{n}} \,\middle|\, \mathcal{E} \right) \geq 1 - \delta.$$

Thus

$$\mathbb{P} \left( |\widehat{\mu}(v) - \mu^*| > 2v \left( \frac{\sigma}{\sqrt{2v}} + 1 \right)^2 \sqrt{\frac{\log(2/\delta)}{n}} \right)$$

$$= \mathbb{P} \left( |\widehat{\mu}(v) - \mu^*| > 2v \left( \frac{\sigma}{\sqrt{2v}} + 1 \right)^2 \sqrt{\frac{\log(2/\delta)}{n}}, \, \mathcal{E} \right)$$

$$+ \mathbb{P} \left( |\widehat{\mu}(v) - \mu^*| > 2v \left( \frac{\sigma}{\sqrt{2v}} + 1 \right)^2 \sqrt{\frac{\log(2/\delta)}{n}}, \, \mathcal{E}^c \right)$$

$$\leq \mathbb{P} \left( |\widehat{\mu}(v) - \mu^*| > 2v \left( \frac{\sigma}{\sqrt{2v}} + 1 \right)^2 \sqrt{\frac{\log(2/\delta)}{n}} \,\middle|\, \mathcal{E} \right) + \mathbb{P} \left( \mathcal{E}^c \right)$$

$$\leq 2\delta.$$

Then with probability at least $1 - 2\delta$, we have

$$|\widehat{\mu}(v) - \mu^*| \leq 2v \left( \frac{\sigma}{\sqrt{2v}} + 1 \right)^2 \sqrt{\frac{\log(2/\delta)}{n}}.$$

Using a change of variable $2\delta \to \delta$ finishes the proof. $\square$

## E.5 SUPPORTING LEMMAS

This subsection collects two supporting lemmas that are used earlier in this section.

**Lemma E.5.** Let $\varepsilon_i$ be i.i.d. random variables such that $\mathbb{E}\varepsilon_i = 0$ and $\mathbb{E}\varepsilon_i^2 = 1$. For any $0 < \delta < 1$, with probability at least $1 - 2\delta$, we have

$$\left| \frac{1}{n} \sum_{i=1}^n \frac{\tau \varepsilon_i}{\sqrt{\tau^2 + \varepsilon_i^2}} - \mathbb{E} \frac{\tau \varepsilon_i}{\sqrt{\tau^2 + \varepsilon_i^2}} \right| \leq \sigma \sqrt{\frac{2 \log(1/\delta)}{n}} + \frac{\tau \log(1/\delta)}{3n}.$$

*Proof of Lemma E.5.* The random variables $Z_i := \tau\psi_\tau(\varepsilon_i) = \tau\varepsilon_i/(\tau^2 + \varepsilon_i^2)^{1/2}$ with $\mu_z = \mathbb{E}Z_i$ and $\sigma_z^2 = \text{var}(Z_i)$ are bounded i.i.d. random variables such that

$$|Z_i| = \left|\tau\varepsilon_i/(\tau^2 + \varepsilon_i^2)^{1/2}\right| \leq |\varepsilon_i| \wedge \tau \leq \tau,$$

$$|\mu_z| = |\mathbb{E}Z_i| = \left|\mathbb{E}\left(\tau\varepsilon_i/(\tau^2 + \varepsilon_i^2)^{1/2}\right)\right| \leq \frac{\sigma^2}{2\tau},$$

$$\mathbb{E}Z_i^2 = \mathbb{E}\left(\frac{\tau^2\varepsilon_i^2}{\tau^2 + \varepsilon_i^2}\right) \leq \sigma^2,$$

$$\sigma_z^2 := \text{var}(Z_i) = \mathbb{E}\left(\tau\varepsilon_i/(\tau^2 + \varepsilon_i^2)^{1/2} - \mu_z\right)^2$$

$$= \mathbb{E}\left(\frac{\tau^2\varepsilon_i^2}{\tau^2 + \varepsilon_i^2}\right) - \mu_z^2 \leq \sigma^2.$$

For third and higher order absolute moments, we have

$$\mathbb{E}|Z_i|^k = \mathbb{E}\left|\frac{\tau\varepsilon_i}{\sqrt{\tau^2 + \varepsilon_i^2}}\right|^k \leq \sigma^2\tau^{k-2} \leq \frac{k!}{2}\sigma^2(\tau/3)^{k-2}, \text{ for all integers } k \geq 3.$$

Using Lemma H.2 with $v = n\sigma^2$ and $c = \tau/3$, we have for any $t > 0$

$$\mathbb{P}\left(\left|\sum_{i=1}^n \frac{\tau\varepsilon_i}{\sqrt{\tau^2 + \varepsilon_i^2}} - \sum_{i=1}^n \mathbb{E}\frac{\tau\varepsilon_i}{\sqrt{\tau^2 + \varepsilon_i^2}}\right| \geq \sqrt{2n\sigma^2 t} + \frac{\tau t}{3}\right) \leq 2\exp(-t).$$

Taking $t = \log(1/\delta)$ acquires that for any $0 < \delta < 1$

$$\mathbb{P}\left(\left|\frac{1}{n}\sum_{i=1}^n \frac{\tau\varepsilon_i}{\sqrt{\tau^2 + \varepsilon_i^2}} - \frac{1}{n}\sum_{i=1}^n \mathbb{E}\frac{\tau\varepsilon_i}{\sqrt{\tau^2 + \varepsilon_i^2}}\right| \leq \sigma\sqrt{\frac{2\log(1/\delta)}{n}} + \frac{\tau\log(1/\delta)}{3n}\right) \geq 1 - 2\delta.$$

This completes the proof.

$\square$

**Lemma E.6.** For any $0 < \delta < 1$, with probability at least $1 - \delta$,

$$\frac{1}{n}\sum_{i=1}^n \frac{\tau^3}{(\tau^2 + \varepsilon_i^2)^{3/2}} - \mathbb{E}\frac{\tau^3}{(\tau^2 + \varepsilon_i^2)^{3/2}} \geq -\sqrt{\frac{\log(1/\delta)}{2n}}.$$

Moreover, with probability at least $1 - \delta$, it holds uniformly over $\tau \geq \tau_{v_0} \geq 0$ that

$$\frac{1}{n}\sum_{i=1}^n \frac{\tau^3}{(\tau^2 + \varepsilon_i^2)^{3/2}} \geq \mathbb{E}\frac{\tau_{v_0}^3}{(\tau_{v_0}^2 + \varepsilon_i^2)^{3/2}} - \sqrt{\frac{\log(1/\delta)}{2n}}.$$

*Proof of Lemma E.6.* The random variables $Z_i = Z_i(\tau) := \tau^3/(\tau^2 + \varepsilon_i^2)^{3/2}$ with $\mu_z = \mathbb{E}Z_i$ and $\sigma_z^2 = \text{var}(Z_i)$ are bounded i.i.d. random variables such that

$$0 \leq Z_i = \tau^3/(\tau^2 + \varepsilon_i^2)^{3/2} \leq 1.$$

Therefore, using Lemma H.1 with $v = n$ acquires that for any $t > 0$

$$\mathbb{P}\left(\sum_{i=1}^n \frac{\tau^3}{(\tau^2 + \varepsilon_i^2)^{3/2}} - \sum_{i=1}^n \mathbb{E}\left(\frac{\tau^3}{(\tau^2 + \varepsilon_i^2)^{3/2}}\right) \leq -\sqrt{\frac{nt}{2}}\right) \leq \exp(-t).$$

Taking $t = \log(1/\delta)$ acquires that for any $0 < \delta < 1$

$$\mathbb{P}\left(\frac{1}{n}\sum_{i=1}^n \frac{\tau^3}{(\tau^2 + \varepsilon_i^2)^{3/2}} - \frac{1}{n}\sum_{i=1}^n \mathbb{E}\left(\frac{\tau^3}{(\tau^2 + \varepsilon_i^2)^{3/2}}\right) > -\sqrt{\frac{\log(1/\delta)}{2n}}\right) > 1 - \delta.$$

The second result follows from the fact that $Z_i(\tau)$ is an increasing function of $\tau$. Specifically, we have with probability at least $1 - \delta$

$$\frac{1}{n} \sum_{i=1}^{n} \frac{\tau^3}{(\tau^2 + \varepsilon_i^2)^{3/2}} \geq \frac{1}{n} \sum_{i=1}^{n} \frac{\tau_{v_0}^3}{(\tau_{v_0}^2 + \varepsilon_i^2)^{3/2}}$$

$$\geq \mathbb{E}\left( \frac{\tau_{v_0}^3}{(\tau_{v_0}^2 + \varepsilon_i^2)^{3/2}} \right) + \frac{1}{n} \sum_{i=1}^{n} \frac{\tau_{v_0}^3}{(\tau_{v_0}^2 + \varepsilon_i^2)^{3/2}} - \mathbb{E}\left( \frac{\tau_{v_0}^3}{(\tau_{v_0}^2 + \varepsilon_i^2)^{3/2}} \right)$$

$$\geq \mathbb{E}\left( \frac{\tau_{v_0}^3}{(\tau_{v_0}^2 + \varepsilon_i^2)^{3/2}} \right) - \sqrt{\frac{\log(1/\delta)}{2n}}.$$

This finishes the proof.

$\square$

## F PROOFS FOR THE SELF-TUNED CASE

This section collects the proofs for Theorems 3.1 and 3.2.

### F.1 PROOF OF THEOREM OF 3.1

*Proof of Theorem of 3.1.* Recall that $\tau = v\sqrt{n}/z$. For simplicity, let $\widehat{\tau} = \widehat{v}\sqrt{n}/z$. Define the profile loss $L_n^{\mathrm{pro}}(v)$ as

$$L_n^{\mathrm{pro}}(v) := L_n(\widehat{\mu}(v), v) = \min_{\mu} L_n(\mu, v).$$

Then it is convex and its first-order gradient is

$$\nabla L_n^{\mathrm{pro}}(v) = \nabla L_n(\widehat{\mu}(v), v) = \frac{\partial}{\partial v}\widehat{\mu}(v) \cdot \frac{\partial}{\partial v} L_n(\mu, v)\Big|_{\mu = \widehat{\mu}(v)} + \frac{\partial}{\partial v} L_n(\mu, v)\Big|_{\mu = \widehat{\mu}(v)} = \frac{\partial}{\partial v} L_n(\widehat{\mu}(v), v),$$

(F.1)

where we use the fact that $\partial/\partial\mu\, L_n(\mu, v)|_{\mu = \widehat{\mu}(v)} = 0$, implied by the stationarity of $\widehat{\mu}(v)$.

**Assuming that the constraint is inactive.** We first assume that the constraint is not active for any stationary point $\widehat{v}$, that is, any stationary point $\widehat{v}$ is an interior point of $[v_0, V_0]$, aka $\widehat{v} \in (v_0, V_0)$. By the joint convexity of $L_n(\mu, v)$ and the convexity of $L_n^{\mathrm{pro}}(v)$, $(\widehat{\mu}(\widehat{v}), \widehat{v})$ and $\widehat{v}$ are stationary points of $L_n(\mu, v)$ and $L_n(\widehat{\mu}(v), v)$, respectively. Thus we have

$$\frac{\partial}{\partial\mu} L_n(\mu, v)\Big|_{(\mu,v)=(\widehat{\mu}(\widehat{v}),\widehat{v})} = -\frac{\sqrt{n}}{z} \cdot \frac{1}{n} \sum_{i=1}^{n} \frac{y_i - \widehat{\mu}(\widehat{v})}{\sqrt{\widehat{\tau}^2 + (y_i - \widehat{\mu}(\widehat{v}))^2}} = 0,$$

$$\frac{\partial}{\partial v} L_n(\mu, v)\Big|_{(\mu,v)=(\widehat{\mu}(\widehat{v}),\widehat{v})} = \frac{n}{z^2} \cdot \frac{1}{n} \sum_{i=1}^{n} \frac{\widehat{\tau}}{\sqrt{\widehat{\tau}^2 + (y_i - \widehat{\mu}(\widehat{v}))^2}} - \left( \frac{n}{z^2} - a \right) = 0,$$

$$\nabla L_n^{\mathrm{pro}}(v)\Big|_{v=\widehat{v}} = \nabla L_n(\widehat{\mu}(\widehat{v}), \widehat{v})\Big|_{v=\widehat{v}} = \frac{\partial}{\partial v} L_n(\widehat{\mu}(v), v)\Big|_{v=\widehat{v}} = \frac{\partial}{\partial v} L_n(\mu, v)\Big|_{(\mu,v)=(\widehat{\mu}(\widehat{v}),\widehat{v})} = 0,$$

where the first two equalities are on partial derivatives of $L_n(\mu, v)$ and the last one is on the derivative of the profile loss $L_n^{\mathrm{pro}}(v) \equiv L_n(\widehat{\mu}(v), v)$.

Recall that $\tau = \sqrt{n}v/z$. Let $f(\tau) = z^2 \nabla L_n^{\mathrm{pro}}(v)/n$, that is,

$$f(\tau) = \frac{1}{n} \sum_{i=1}^{n} \frac{\tau}{\sqrt{\tau^2 + (y_i - \widehat{\mu}(v))^2}} - \left( 1 - \frac{az^2}{n} \right).$$

In other words, $\widehat{\tau} = \sqrt{n}\widehat{v}/z$ satisfies $f(\widehat{\tau}) = 0$. Assuming that the conststraint is inactive, we split the proof into two steps.

**Step 1: Proving $\widehat{v} \leq C_0 \sigma$ for some universal constant $C_0$.** We will employ the method of proof by contradiction. Assume there exists some $v$ such that

$$v > (1+\epsilon)\sqrt{r^2 + \sigma^2} \quad \text{and} \quad \nabla L_v^{\mathrm{pro}}(v) = 0;$$

or equivalently, there exists some $\tau$ such that

$$\tau > (1+\epsilon)\sqrt{r^2 + \sigma^2}\sqrt{n}/z =: \bar{\tau} \quad \text{and} \quad f(\tau) = 0, \tag{F.2}$$

where $\epsilon$ and $r$ are to be determined later. Let $\tau_{v_0} = v_0\sqrt{n}/z$. Then, provided $n$ is large enough, Lemma E.3 implies that Assumption E.1 with $\kappa_\ell = 1/(2v)$ and local radius $r \geq r_0(\kappa_\ell)$ holds uniformly over $v \geq v_0$ conditional on the following event

$$\mathcal{E}_1 := \left\{ \frac{1}{n}\sum_{i=1}^n \frac{(\tau_{v_0}^2 + 2r^2)^{3/2}}{(\tau_{v_0}^2 + 2r^2 + 2\varepsilon_i^2)^{3/2}} - \frac{1}{n}\sum_{i=1}^n \mathbb{E}\frac{(\tau_{v_0}^2 + 2r^2)^{3/2}}{(\tau_{v_0}^2 + 2r^2 + 2\varepsilon_i^2)^{3/2}} \geq -\sqrt{\frac{\log(1/\delta)}{2n}} \right\}.$$

Conditional on the intersection of event $\mathcal{E}_1$ and the following event

$$\mathcal{E}_2 := \left\{ \sup_{v \in [v_0, V_0]} \left| \frac{1}{n}\sum_{i=1}^n \frac{\varepsilon_i}{\sqrt{\tau^2 + \varepsilon_i^2}} \right| \leq C \cdot \frac{V_0}{v_0} \cdot \frac{\log(n/\delta)}{n} \right\},$$

where $z \lesssim \sqrt{\log(n/\delta)}$ and $C$ is some constant, and following the proof of Theorem E.2, for any fixed $v$ and thus fixed $\tau = v\sqrt{n}/z$, we have

$$\kappa_\ell |\widehat{\mu}(v) - \mu^*| \leq \left| \frac{1}{\sqrt{n}}\sum_{i=1}^n \frac{\varepsilon_i}{z\sqrt{\tau^2 + \varepsilon_i^2}} \right|.$$

Thus, for any $v$ such that $v_0 \vee \bar{v}_0 := v_0 \vee (1+\epsilon)\sqrt{r^2 + \sigma^2} < v < V_0$, we have on $\mathcal{E}_2$ that

$$\sup_{v_0 \vee \bar{v}_0 < v < V_0} \kappa_\ell(v) |\widehat{\mu}(v) - \mu^*| \leq \sup_{v \in [v_0, V_0]} \kappa_\ell(v) |\widehat{\mu}(v) - \mu^*|$$

$$\leq \sup_{v \in [v_0, V_0]} \left| \frac{1}{\sqrt{n}}\sum_{i=1}^n \frac{\varepsilon_i}{z\sqrt{\tau^2 + \varepsilon_i^2}} \right|$$

$$\leq C \cdot \frac{V_0}{v_0} \cdot \frac{\log(n/\delta)}{z\sqrt{n}},$$

which, by Lemma E.3, yields

$$\sup_{v \in [v_0, V_0]} |\widehat{\mu}(v) - \mu^*| \leq 2C \cdot \frac{V_0^2}{v_0} \cdot \frac{\log(n/\delta)}{z\sqrt{n}} =: r. \tag{F.3}$$

The above $r$ can be further refined by using the finer lower bound $\bar{v}_0$ of $v$ instead of $v_0$, but we use $v_0$ for simplicity. Let $\Delta = \mu^* - \widehat{\mu}(v)$, and we have $|\Delta| \leq r$. Let the event $\mathcal{E}_3$ be

$$\mathcal{E}_3 := \left\{ \frac{1}{n}\sum_{i=1}^n \frac{\sqrt{\bar{\tau}^2 + 2(r^2 + \varepsilon_i^2)} - \bar{\tau}}{\sqrt{\bar{\tau}^2 + 2(r^2 + \varepsilon_i^2)}} - \mathbb{E}\left( \frac{\sqrt{\bar{\tau}^2 + 2(r^2 + \varepsilon_i^2)} - \bar{\tau}}{\sqrt{\bar{\tau}^2 + 2(r^2 + \varepsilon_i^2)}} \right) \leq \sqrt{\frac{\log(1/\delta)2(r^2 + \sigma^2)}{n\bar{\tau}^2}} + \frac{\log(1/\delta)}{3n} \right\}.$$

Thus on the event $\mathcal{E}_1 \cap \mathcal{E}_2 \cap \mathcal{E}_3$ and using the fact that $1 - 1/\sqrt{1+x}$ is an increasing function, we have

$$f(\tau) = \frac{az^2}{n} - \frac{1}{n}\sum_{i=1}^{n}\frac{\sqrt{\tau^2 + (\Delta + \varepsilon_i)^2} - \tau}{\sqrt{\tau^2 + (\Delta + \varepsilon_i)^2}} \geq \frac{az^2}{n} - \frac{1}{n}\sum_{i=1}^{n}\frac{\sqrt{\tau^2 + 2(r^2 + \varepsilon_i^2)} - \tau}{\sqrt{\tau^2 + 2(r^2 + \varepsilon_i^2)}}$$

$$> \frac{az^2}{n} - \frac{1}{n}\sum_{i=1}^{n}\frac{\sqrt{\bar{\tau}^2 + 2(r^2 + \varepsilon_i^2)} - \bar{\tau}}{\sqrt{\bar{\tau}^2 + 2(r^2 + \varepsilon_i^2)}} \qquad\qquad (\tau < \bar{\tau})$$

$$\geq \frac{az^2}{n} - \left\{ \mathbb{E}\left(\frac{\sqrt{\bar{\tau}^2 + 2(r^2 + \varepsilon_i^2)} - \bar{\tau}}{\sqrt{\bar{\tau}^2 + 2(r^2 + \varepsilon_i^2)}}\right) + \frac{1}{n}\sum_{i=1}^{n}\frac{\sqrt{\bar{\tau}^2 + 2(r^2 + \varepsilon_i^2)} - \bar{\tau}}{\sqrt{\bar{\tau}^2 + 2(r^2 + \varepsilon_i^2)}} - \mathbb{E}\left(\frac{\sqrt{\bar{\tau}^2 + 2(r^2 + \varepsilon_i^2)} - \bar{\tau}}{\sqrt{\bar{\tau}^2 + 2(r^2 + \varepsilon_i^2)}}\right) \right\}$$

$$\geq \frac{az^2}{n} - \left(\frac{r^2 + \sigma^2}{\bar{\tau}^2} + \sqrt{\frac{\log(1/\delta)\cdot 2(r^2 + \sigma^2)}{n\bar{\tau}^2}} + \frac{\log(1/\delta)}{3n}\right)$$

$$= \frac{z^2}{n}\left(a - \frac{\log(1/\delta)}{3z^2}\right) - \left(\frac{r^2 + \sigma^2}{r^2 + \sigma^2}\frac{z^2}{(1+\epsilon)^2 n} + \sqrt{\frac{r^2 + \sigma^2}{r^2 + \sigma^2}\frac{2z^2 \log(1/\delta)}{(1+\epsilon)^2 n^2}}\right)$$

$$\text{(Definition of } \bar{\tau})$$

$$\geq \frac{(a - 1/3)z^2}{n} - \left(\frac{r^2 + \sigma^2}{r^2 + \sigma^2}\frac{z^2}{(1+\epsilon)^2 n} + \sqrt{\frac{r^2 + \sigma^2}{r^2 + \sigma^2}\frac{2z^4}{(1+\epsilon)^2 n^2}}\right) \qquad (z^2 \geq \log(1/\delta))$$

$$\geq \frac{(a - 1/3)z^2}{n} - \frac{z^2}{n}\cdot\left(\frac{1}{(1+\epsilon)^2} + \sqrt{\frac{2}{(1+\epsilon)^2}}\right)$$

$$= \frac{z^2}{n}\left(a - \frac{1}{3} - \frac{1}{(1+\epsilon)^2} - \sqrt{\frac{2}{(1+\epsilon)^2}}\right)$$

$$\geq 0,$$

provided that

$$\frac{1}{1+\epsilon} \leq \frac{\sqrt{1 + 2(a - 1/3)} - 1}{\sqrt{2}},$$

or equivalently

$$\epsilon \geq \frac{\sqrt{4a + 2/3} + 2/3 + \sqrt{2} - 2a}{2(a - 1/3)} =: \epsilon(a).$$

In other words, conditional on the event $\mathcal{E}_1 \cap \mathcal{E}_2 \cap \mathcal{E}_3$ and taking $\epsilon \geq \epsilon(a)$, $f(\tau) > 0$ for $\tau > \bar{\tau} := (1 + \epsilon)\sqrt{r^2 + \sigma^2}\sqrt{n}/z$. This contradicts with (F.2), and thus

$$\hat{\tau} \leq (1 + \epsilon)\sqrt{r^2 + \sigma^2}\sqrt{n}/z.$$

If $a = 1/2$ and conditional on the same event, the above holds with

$$\epsilon = 9 \geq \epsilon(1/2).$$

If $n$ is large enough such that $12\sigma \geq 10\sqrt{r^2 + \sigma^2}$, then conditional on the event $\mathcal{E}_1 \cap \mathcal{E}_2 \cap \mathcal{E}_3$, we have

$$v_0 \leq \hat{v} \leq C_0 \sigma,$$

where $C_0 = 12$.

**Step 2: Proving $\hat{v} \geq c_0 \left(\frac{\sigma_{\tau_{v_0}^2/2 - 1}}{\sigma_{\tau_{v_0}^2/2}} \wedge 1\right)\sigma_{\tau_{v_0}^2 - 1}$ for some universal constant $c_0$.** We will again employ the method of proof by contradiction. Let

$$g(\tau) := \left(\frac{1}{n}\sum_{i=1}^{n}\frac{\tau^2}{\sqrt{\tau^2 + (\Delta + \varepsilon_i)^2}}\right)^2 - \left(1 - \frac{az^2}{n}\right)^2.$$

Assume there exists some $v$ such that

$$v < c \quad \text{and} \quad \frac{\partial}{\partial v} L_n(\widehat{\mu}(v), v) = 0;$$

or equivalently, assume there exists some $\tau$ such that

$$\tau < c\sqrt{n}/z =: \underline{\tau} \quad \text{and} \quad g(\tau) = 0. \tag{F.4}$$

It is impossible that $c \leq v_0$ because any stationary point $v$ is in $(v_0, V_0)$. Thus $c > v_0$. Let $\Delta = \widehat{\mu}(v) - \mu^*$. Then on the event $\mathcal{E}_1 \cap \mathcal{E}_2$, using the facts that $\sqrt{x}$ is a concave function and $1/\sqrt{1+y/x}$ is an increasing function of $x$, we have

$$\frac{1}{n} \sum_{i=1}^{n} \frac{\tau^2}{\sqrt{\tau^2 + (\Delta + \varepsilon_i)^2}} = \frac{1}{n} \sum_{i=1}^{n} \frac{1}{\sqrt{1 + (\Delta + \varepsilon_i)^2/\tau^2}}$$

$$\leq \frac{1}{n} \sum_{i=1}^{n} \frac{1}{\sqrt{1 + (\Delta + \varepsilon_i)^2/\underline{\tau}^2}}$$

$$\leq \sqrt{\frac{1}{n} \sum_{i=1}^{n} \frac{1}{1 + (\Delta + \varepsilon_i)^2/\underline{\tau}^2}}$$

$$\leq \sqrt{\frac{1}{n} \sum_{i=1}^{n} \frac{1}{1 + \underline{\tau}^{-2}(\Delta + \varepsilon_i)^2 \cdot 1\left((\Delta + \varepsilon_i)^2 \leq \underline{\tau}^2\right)}}$$

$$\leq \sqrt{1 - \frac{1}{n} \cdot \frac{1}{2\underline{\tau}^2} \sum_{i=1}^{n} (\Delta + \varepsilon_i)^2 \cdot 1\left((\Delta + \varepsilon_i)^2 \leq \underline{\tau}^2\right)}.$$

By the proof from step 1, we have on the event $\mathcal{E}_1 \cap \mathcal{E}_2$ that

$$\sup_{v \in [v_0, V_0]} |\widehat{\mu}(v) - \mu^*| \leq r,$$

where $r$ is defined in (F.3). Then

$$g(\tau) \leq 1 - \frac{1}{n} \cdot \frac{1}{2\underline{\tau}^2} \sum_{i=1}^{n} (\Delta + \varepsilon_i)^2 \cdot 1\left((\Delta + \varepsilon_i)^2 \leq \underline{\tau}^2\right) - \left(1 - \frac{az^2}{n}\right)^2$$

$$< \frac{2az^2}{n} - \frac{1}{n} \cdot \frac{1}{2\underline{\tau}^2} \sum_{i=1}^{n} (\Delta + \varepsilon_i)^2 \cdot 1\left((\Delta + \varepsilon_i)^2 \leq \underline{\tau}^2\right) \qquad \text{(as long as } az^2/n > 0)$$

$$\leq \frac{2az^2}{n} - \frac{1}{n} \cdot \frac{1}{2\underline{\tau}^2} \sum_{i=1}^{n} \left(\varepsilon_i^2 + 2\Delta\varepsilon_i\right) \cdot 1\left(\varepsilon_i^2 \leq \frac{\underline{\tau}^2}{2} - r^2\right)$$

$$\leq \frac{2az^2}{n} - \frac{1}{2\underline{\tau}^2} \left(\frac{1}{n} \sum_{i=1}^{n} \varepsilon_i^2 1\left(\varepsilon_i^2 \leq \frac{\underline{\tau}^2}{2} - r^2\right) - \frac{2}{n} \sum_{i=1}^{n} r|\varepsilon_i| 1\left(\varepsilon_i^2 \leq \frac{\underline{\tau}^2}{2} - r^2\right)\right)$$

$$= \frac{2az^2}{n} - \frac{1}{2\underline{\tau}^2} \left(\text{I} - 2r \cdot \text{II}\right).$$

Define the probability event $\mathcal{E}_4$ as

$$\mathcal{E}_4 := \mathcal{E}_{41} \cap \mathcal{E}_{42},$$

where

$$\mathcal{E}_{41} := \left\{\frac{1}{n} \sum_{i=1}^{n} \varepsilon_i^2 1\left(\varepsilon_i^2 \leq \frac{\underline{\tau}^2}{2} - r^2\right) \geq \mathbb{E}\varepsilon_i^2 1\left(\varepsilon_i^2 \leq \frac{\underline{\tau}^2}{2} - r^2\right) - \sigma_{\frac{\underline{\tau}^2}{2}} \sqrt{\frac{\underline{\tau}^2 \log(1/\delta)}{n}} - \frac{\underline{\tau}^2 \log(1/\delta)}{6n}\right\} \quad \text{and}$$

$$\mathcal{E}_{42} := \left\{\frac{1}{n} \sum_{i=1}^{n} |\varepsilon_i| 1\left(\varepsilon_i^2 \leq \frac{\underline{\tau}^2}{2} - r^2\right) \leq \mathbb{E}|\varepsilon_i| 1\left(\varepsilon_i^2 \leq \frac{\underline{\tau}^2}{2} - r^2\right) + \sqrt{\frac{2\sigma_{\underline{\tau}^2/2}^2 \log(1/\delta)}{n}} + \frac{\underline{\tau} \log(1/\delta)}{3\sqrt{2}n}\right\}.$$

If $n$ is sufficiently large such that

$$r^2 \leq \epsilon_0 \lesssim \left( \frac{\log n + \log(1/\delta)}{z\sqrt{n}} \right)^2 \leq 1 \quad \text{and}$$

$$\frac{r}{\underline{\tau}^2} \left( \sigma_{\underline{\tau}^2/2}^2 + \sqrt{\frac{2\sigma_{\underline{\tau}^2/2}^2 \log(1/\delta)}{n}} + \frac{\tau \log(1/\delta)}{3\sqrt{2}n} \right) \leq \frac{1}{12} \frac{\log(1/\delta)}{n},$$

then conditional on $\mathcal{E}_4$, we have

$$\mathrm{I} \geq \mathbb{E}\varepsilon_i^2 1\left( \varepsilon_i^2 \leq \frac{\tau^2}{2} - r^2 \right) - \sigma_{\frac{\tau^2}{2}} \sqrt{\frac{\tau^2 \log(1/\delta)}{n}} - \frac{\tau^2 \log(1/\delta)}{6n} \quad \text{and}$$

$$\mathrm{II} \leq \mathbb{E}|\varepsilon_i| 1\left( \varepsilon_i^2 \leq \frac{\tau^2}{2} - r^2 \right) + \sqrt{\frac{2\sigma_{\underline{\tau}^2/2}^2 \log(1/\delta)}{n}} + \frac{\tau \log(1/\delta)}{3\sqrt{2}n}.$$

Thus conditional on $\mathcal{E}_4$ we have

$$
\begin{aligned}
g(\tau) &< \frac{2az^2}{n} - \frac{1}{2\underline{\tau}^2} \left( \mathrm{I} - 2r \cdot \mathrm{II} \right) \\
&\leq \frac{2az^2}{n} - \frac{1}{2\underline{\tau}^2} \left( \mathbb{E}\varepsilon_i^2 1\left( \varepsilon_i^2 \leq \frac{\tau^2}{2} - r^2 \right) - \sigma_{\underline{\tau}^2/2} \sqrt{\frac{\tau^2 \log(1/\delta)}{n}} - \frac{\tau^2 \log(1/\delta)}{6n} \right) \\
&\quad + \frac{r}{\underline{\tau}^2} \left( \mathbb{E}|\varepsilon_i| 1\left( \varepsilon_i^2 \leq \frac{\tau^2}{2} - r^2 \right) + \sqrt{\frac{2\sigma_{\underline{\tau}^2/2}^2 \log(1/\delta)}{n}} + \frac{\tau \log(1/\delta)}{3\sqrt{2}n} \right) \\
&\leq \frac{2az^2}{n} - \frac{\sigma_{\underline{\tau}^2/2 - \epsilon_0}^2}{2\underline{\tau}^2} + \frac{\sigma_{\underline{\tau}^2/2} \sqrt{\log(1/\delta)}}{2\underline{\tau}\sqrt{n}} + \frac{\log(1/\delta)}{12n} + \frac{r}{\underline{\tau}^2} \left( \sigma_{\underline{\tau}^2/2}^2 + \sqrt{\frac{2\sigma_{\underline{\tau}^2/2}^2 \log(1/\delta)}{n}} + \frac{\tau \log(1/\delta)}{3\sqrt{2}n} \right) \\
&\leq \frac{z^2}{n} \left( 2a + \frac{\log(1/\delta)}{z^2} \cdot \frac{1}{6} \right) - \frac{\sigma_{\underline{\tau}^2/2 - \epsilon_0}^2}{2\underline{\tau}^2} + \frac{\sigma_{\underline{\tau}^2/2} \sqrt{\log(1/\delta)}}{2\underline{\tau}\sqrt{n}} \\
&= \frac{z^2}{2n} \left( 4a + \frac{\log(1/\delta)}{z^2} \cdot \frac{1}{3} - \frac{\sigma_{\underline{\tau}^2/2 - \epsilon_0}^2}{c^2} + \frac{\sigma_{\underline{\tau}^2/2}}{c} \cdot \frac{\sqrt{\log(1/\delta)}}{z} \right) && (\underline{\tau} = c\sqrt{n}/z) \\
&\leq \frac{z^2}{2n} \left( 4a + \frac{1}{3} - \frac{\sigma_{\underline{\tau}^2/2 - \epsilon_0}^2}{c^2} + \frac{\sigma_{\underline{\tau}^2/2}}{c} \right) && (z^2 \geq \log(1/\delta)) \\
&\leq 0,
\end{aligned}
$$

for any $c$ such that

$$c \leq \frac{\sigma_{\underline{\tau}^2/2}}{2(4a + 1/3)} \left( \sqrt{1 + \frac{4(4a + 1/3)\sigma_{\underline{\tau}^2/2 - \epsilon_0}^2}{\sigma_{\underline{\tau}^2/2}^2}} - 1 \right),$$

In other words, conditional on the event $\mathcal{E}_1 \cap \mathcal{E}_2 \cap \mathcal{E}_4$ and taking any $c$ satisfying the above inequality, we have

$$g(\tau) < 0 \text{ for any } \tau < \underline{\tau} = c\sqrt{n}/z.$$

This is a contradiction. Thus, $\widehat{\tau} \geq \underline{\tau} = c\sqrt{n}/z$, or equivalently $\widehat{v} \geq c > v_0$. Using the inequality

$$\sqrt{1 + x} - 1 \geq 1(x \geq 3) + \frac{x}{3} 1(0 \leq x < 3) \geq \frac{x}{3} \wedge 1 \quad \forall x \geq 0,$$

we obtain

$$\frac{\sigma_{\underline{\tau}^2/2}}{2(4a+1/3)}\left(\sqrt{1+\frac{4(4a+1/3)\sigma_{\underline{\tau}^2/2-\epsilon_0}^2}{\sigma_{\underline{\tau}^2/2}^2}}-1\right)$$

$$=\frac{3\sigma_{\tau_{v_0}^2/2}}{14}\left(\sqrt{1+\frac{28\sigma_{\underline{\tau}^2/2-\epsilon_0}^2}{3\sigma_{\underline{\tau}^2/2}^2}}-1\right) \qquad (a=1/2)$$

$$\geq\frac{3\sigma_{\underline{\tau}^2/2}}{14}\left(\frac{28\sigma_{\underline{\tau}^2/2-\epsilon_0}^2}{9\sigma_{\underline{\tau}^2/2}^2}\wedge 1\right)$$

$$=\frac{2\sigma_{\underline{\tau}^2/2-\epsilon_0}^2}{3\sigma_{\underline{\tau}^2/2}}\wedge\frac{3\sigma_{\underline{\tau}^2/2}}{14}$$

$$\geq\frac{1}{5}\left(\frac{\sigma_{\underline{\tau}^2/2-1}}{\sigma_{\underline{\tau}^2/2}}\wedge 1\right)\sigma_{\underline{\tau}^2/2-1}$$

$$\geq\frac{1}{5}\left(\frac{\sigma_{\tau_{v_0}^2/2-1}}{\sigma_{\tau_{v_0}^2/2}}\wedge 1\right)\sigma_{\tau_{v_0}^2/2-1}.$$

Therefore we can take $c=5^{-1}(\sigma_{\tau_{v_0}^2/2-1}/\sigma_{\tau_{v_0}^2/2}\wedge 1)\sigma_{\tau_{v_0}^2/2-1}$. Thus on the event $\mathcal{E}_1\cap\mathcal{E}_2\cap\mathcal{E}_4$, we have

$$\widehat{v}\geq c:=c_0\left(\frac{\sigma_{\tau_{v_0}^2/2-1}}{\sigma_{\tau_{v_0}^2/2}}\wedge 1\right)\sigma_{\tau_{v_0}^2/2-1},$$

where $c_0=1/5$ is a universal constant. This finishes the proof of step 2.

**Proving that the constraint is inactive.** If $\widehat{v}\notin(v_0,V_0)$, then $\widehat{v}\in\{v_0,V_0\}$. Suppose $\widehat{v}=v_0$, then $\widehat{v}=v_0<c$. Recall that $\tau_{v_0}=v_0\sqrt{n}/z$. Then we must have $f(\tau_{v_0})\geq 0$, and thus $g(\tau_{v_0})\geq 0$. However, conditional on the probability event $\mathcal{E}_1\cap\mathcal{E}_2\cap\mathcal{E}_4$, repeating the above analysis in step 2 obtains $g(\tau_{v_0})<0$. This is a contradiction. Therefore $\widehat{v}\neq v_0$. Similarly, conditional on probability event $\mathcal{E}_1\cap\mathcal{E}_2\cap\mathcal{E}_3$, we can obtain $\widehat{v}\neq V_0$. Therefore, conditional on the probability event $\mathcal{E}_1\cap\mathcal{E}_2\cap\mathcal{E}_3\cap\mathcal{E}_4$, the constraint must be inactive, aka $\widehat{v}\in(v_0,V_0)$.

Using the first result of Lemma E.6 with $\tau^2$ and $\varepsilon_i^2$ replaced by $\tau_{v_0}^2+2r^2$ and $2\varepsilon_i^2$ respectively, Lemma F.1, Lemma F.2 with $\tau^2$ and $w_i^2$ replaced by $\bar{\tau}^2$ and $2(r^2+\varepsilon_i^2)$ respectively, and Lemma F.3, we obtain

$$\mathbb{P}(\mathcal{E}_1)\geq 1-\delta,\ \mathbb{P}(\mathcal{E}_2)\geq 1-\delta,\ \mathbb{P}(\mathcal{E}_3)\geq 1-\delta,\ \mathbb{P}(\mathcal{E}_4)\geq 1-2\delta,$$

and thus

$$\mathbb{P}(\mathcal{E}_1\cap\mathcal{E}_2\cap\mathcal{E}_3\cap\mathcal{E}_4)\geq 1-5\delta.$$

Putting the above results together, and using Lemmas F.1 and F.3, we obtain with probability at least $1-5\delta$ that

$$c_0(\sigma_{\tau_{v_0}^2/2-1}/\sigma_{\tau_{v_0}^2/2}\wedge 1)\sigma_{\tau_{v_0}^2/2-1}\leq\widehat{v}\leq C_0\sigma.$$

Using a change of variable $5\delta\to\delta$ completes the proof. $\qquad\square$

### F.2  PROOF OF THEOREM 3.2

*Proof of Theorem 3.2.* On the probability event $\mathcal{E}_1\cap\mathcal{E}_2\cap\mathcal{E}_3\cap\mathcal{E}_4$ where $\mathcal{E}_k$'s are defined the same as in the proof of Theorem 3.1, we have

$$c_0(\sigma_{\tau_{v_0}^2/2-1}/\sigma_{\tau_{v_0}^2/2}\wedge 1)\sigma_{\tau_{v_0}^2/2-1}\leq\widehat{v}\leq C_0\sigma.$$

Following the proof of Theorem E.2, for any fixed $v$ and thus $\tau$, we have

$$\kappa_\ell|\widehat{\mu}(v)-\mu^*|\leq\left|\frac{1}{\sqrt{n}}\sum_{i=1}^{n}\frac{\varepsilon_i}{z\sqrt{\tau^2+\varepsilon_i^2}}\right|.$$

For any $v$ such that $c_0'\sigma_{\tau_{v_0}^2/2-1} \le v \le C_0\sigma$ where $c_0' = c_0(\sigma_{\tau_{v_0}^2/2-1}/\sigma_{\tau_{v_0}^2/2} \wedge 1)$ and any $z > 0$, using Lemma F.1 but with $v_0$ and $V_0$ replaced by $c_0'\sigma_{\tau_{v_0}^2/2-1}$ and $C_0\sigma$ respectively, we obtain with probability at least $1 - \delta$

$$\sup_{v\in[c_0'\sigma_{\tau_{v_0}^2/2-1},\, C_0\sigma]} \kappa_\ell(v)\,|\widehat{\mu}(v) - \mu^*| \le \sup_{v\in[c_0'\sigma_{\tau_{v_0}^2/2-1},\, C_0\sigma]} \kappa_\ell(v)\,|\widehat{\mu}(v) - \mu^*|$$

$$\le \sup_{v\in[c_0'\sigma_{\tau_{v_0}^2/2-1},\, C_0\sigma]} \left| \frac{1}{\sqrt{n}}\sum_{i=1}^{n} \frac{\varepsilon_i}{z\sqrt{\tau^2 + \varepsilon_i^2}} \right|$$

$$\le \frac{\sigma}{c_0'\sigma_{\tau_{v_0}^2/2-1}} \sqrt{\frac{2\log(n/\delta)}{n}} + \frac{1}{z}\frac{\log(n/\delta)}{\sqrt{n}}$$

$$+ \frac{\sigma^2}{2c_0'^2\sigma_{\tau_{v_0}^2/2-1}^2}\frac{z}{\sqrt{n}} + \frac{3(C_0\sigma - c_0'\sigma_{\tau_{v_0}^2/2-1})}{\sigma_{\tau_{v_0}^2/2-1}}\frac{1}{z\sqrt{n}},$$

which yields

$$\sup_{v\in[c_0'\sigma_{\tau_{v_0}^2/2-1},\, C_0\sigma]} |\widehat{\mu}(v) - \mu^*| \le C\sigma\,\frac{\log(n/\delta) \vee z^2 \vee 1}{z\sqrt{n}},$$

where $C$ is some constant only depending on $\sigma/\sigma_{\tau_{v_0}^2/2-1}$, $c_0'$, and $C_0$. Putting the above pieces together and if $\log(1/\delta) \le z^2 \le \log(n/\delta)$, we obtain with probability at least $1 - 6\delta$ that

$$|\widehat{\mu}(\widehat{v}) - \mu^*| \le \sup_{v\in[c_0'\sigma_{\tau_{v_0}^2/2-1},\, C_0\sigma]} |\widehat{\mu}(v) - \mu^*| \le C\cdot\sigma\,\frac{\log(n/\delta) \vee 1}{z\sqrt{n}}.$$

Using a change of variable $6\delta \to \delta$ and then setting $z = \log(n/\delta)$ gives

$$|\widehat{\mu}(\widehat{v}) - \mu^*| \le \sup_{v\in[c_0'\sigma_{\tau_{v_0}^2/2-1},\, C_0\sigma]} |\widehat{\mu}(v) - \mu^*| \le C\cdot\sigma\,\sqrt{\frac{\log(n/\delta)}{n}}$$

with a lightly different constant $C$, provided that $\log(n/\delta) \ge 1$, aka $n \ge e\delta$. This completes the proof. $\qquad\square$

## F.3 SUPPORTING LEMMAS

We collect supporting lemmas, aka Lemmas F.1, F.2, and F.3, in this subsection.

**Lemma F.1.** Let $0 < \delta < 1$. Suppose $\sigma \lesssim V_0$ and $z \lesssim \sqrt{\log(n/\delta)}$. Then, with probability at least $1 - \delta$, we have

$$\sup_{v\in[v_0,V_0]} \left| \frac{1}{n}\sum_{i=1}^{n} \frac{\varepsilon_i}{\sqrt{\tau^2 + \varepsilon_i^2}} \right| \le C\cdot\frac{V_0}{v_0}\cdot\frac{\log(n/\delta)}{n}$$

where $C$ is some constant.

*Proof of Lemma F.1.* To prove the uniform bound over $[v_0, V_0]$, we adopt a covering argument. For any $0 < \epsilon \le 1$, there exists an $\epsilon$-cover $\mathcal{N}$ of $[v_0, V_0]$ such that $|\mathcal{N}| \le 3(V_0 - v_0)/\epsilon$. Let $\tau_w = w\sqrt{n}/z$.

Then for every $v \in [v_0, V_0]$, there exists a $w \in \mathcal{N} \subset [v_0, V_0]$ such that $|w - \tau| \le \epsilon$ and

$$
\left| \frac{1}{\sqrt{n}} \sum_{i=1}^{n} \frac{\varepsilon_i}{z\sqrt{\tau^2 + \varepsilon_i^2}} \right| \le \left| \frac{1}{\sqrt{n}} \sum_{i=1}^{n} \frac{\varepsilon_i}{z\sqrt{\tau_w^2 + \varepsilon_i^2}} \right|
$$

$$
+ \left| \frac{1}{\sqrt{n}} \sum_{i=1}^{n} \frac{\varepsilon_i}{z\sqrt{\tau_w^2 + \varepsilon_i^2}} - \frac{1}{\sqrt{n}} \sum_{i=1}^{n} \frac{\varepsilon_i}{z\sqrt{\tau^2 + \varepsilon_i^2}} \right|
$$

$$
\le \left| \frac{1}{\sqrt{n}} \sum_{i=1}^{n} \frac{\varepsilon_i}{z\sqrt{\tau_w^2 + \varepsilon_i^2}} - \mathbb{E}\left[ \frac{1}{\sqrt{n}} \sum_{i=1}^{n} \frac{\varepsilon_i}{z\sqrt{\tau_w^2 + \varepsilon_i^2}} \right] \right|
$$

$$
+ \left| \mathbb{E}\left[ \frac{1}{\sqrt{n}} \sum_{i=1}^{n} \frac{\varepsilon_i}{z\sqrt{\tau_w^2 + \varepsilon_i^2}} \right] \right|
$$

$$
+ \left| \frac{1}{\sqrt{n}} \sum_{i=1}^{n} \frac{\varepsilon_i}{z\sqrt{\tau_w^2 + \varepsilon_i^2}} - \frac{1}{\sqrt{n}} \sum_{i=1}^{n} \frac{\varepsilon_i}{z\sqrt{\tau^2 + \varepsilon_i^2}} \right|
$$

$$
= \mathrm{I} + \mathrm{II} + \mathrm{III}.
$$

For II, we have

$$
\mathrm{II} \le \frac{\sqrt{n}}{z} \cdot \frac{\sigma^2}{2\tau_w^2} \le \frac{z\sigma^2}{2v_0^2 \sqrt{n}}.
$$

For III, using the inequality

$$
\left| \frac{x}{\sqrt{\tau_w^2 + x^2}} - \frac{x}{\sqrt{\tau^2 + x^2}} \right| \le \frac{|\tau_w - \tau|}{2\,|\tau_w| \wedge |\tau|},
$$

we obtain

$$
\mathrm{III} \le \frac{\sqrt{n}}{z} \cdot \frac{\epsilon}{2(w \wedge v)} \le \frac{\sqrt{n}}{z} \cdot \frac{\epsilon}{2v_0}.
$$

We then bound I. For any fixed $\tau_w$, applying Lemma E.5 with the fact that $\left| \mathbb{E}\left( \tau_w \varepsilon_i / (\tau_w^2 + \varepsilon_i^2)^{1/2} \right) \right| \le \sigma^2/(2\tau_w)$, we obtain with probability at least $1 - 2\delta$

$$
\left| \frac{1}{\sqrt{n}} \sum_{i=1}^{n} \frac{\varepsilon_i}{z\sqrt{\tau_w^2 + \varepsilon_i^2}} - \mathbb{E}\left[ \frac{1}{\sqrt{n}} \sum_{i=1}^{n} \frac{\varepsilon_i}{z\sqrt{\tau_w^2 + \varepsilon_i^2}} \right] \right| \le \frac{\sqrt{n}}{z\tau_w} \left( \sigma \sqrt{\frac{2\log(1/\delta)}{n}} + \frac{\tau_w \log(1/\delta)}{n} \right)
$$

$$
\le \frac{\sigma}{z\tau_{v_0}} \sqrt{2\log(1/\delta)} + \frac{1}{z} \frac{\log(1/\delta)}{\sqrt{n}}
$$

where $\tau_{v_0} = v_0\sqrt{n}/z$. Therefore, putting above pieces together and using the union bound, we obtain with probability at least $1 - 6\epsilon^{-1}(V_0 - v_0)\delta$

$$
\sup_{v \in [v_0, V_0]} \left| \frac{1}{\sqrt{n}} \sum_{i=1}^{n} \frac{\varepsilon_i}{z\sqrt{\tau^2 + \varepsilon_i^2}} \right| \le \sup_{w \in \mathcal{N}} \left| \frac{1}{\sqrt{n}} \sum_{i=1}^{n} \frac{\varepsilon_i}{z\sqrt{\tau_w^2 + \varepsilon_i^2}} - \mathbb{E}\left[ \frac{1}{\sqrt{n}} \sum_{i=1}^{n} \frac{\varepsilon_i}{z\sqrt{\tau_w^2 + \varepsilon_i^2}} \right] \right|
$$

$$
+ \frac{z\sigma^2}{2v_0^2 \sqrt{n}} + \frac{\sqrt{n}}{z} \cdot \frac{\epsilon}{2v_0}
$$

$$
\le \frac{\sigma}{v_0} \sqrt{\frac{2\log(1/\delta)}{n}} + \frac{1}{z} \frac{\log(1/\delta)}{\sqrt{n}} + \frac{\sigma^2}{2v_0^2} \frac{z}{\sqrt{n}} + \frac{\sqrt{n}}{z} \cdot \frac{\epsilon}{2v_0}.
$$

Taking $\epsilon = 6(V_0 - v_0)/n$, we obtain with probability at least $1 - n\delta$

$$
\sup_{v \in [v_0, V_0]} \left| \frac{1}{\sqrt{n}} \sum_{i=1}^{n} \frac{\varepsilon_i}{z\sqrt{\tau^2 + \varepsilon_i^2}} \right| \le \frac{\sigma}{v_0} \sqrt{\frac{2\log(1/\delta)}{n}} + \frac{1}{z} \frac{\log(1/\delta)}{\sqrt{n}} + \frac{\sigma^2}{2v_0^2} \frac{z}{\sqrt{n}} + \frac{3(V_0 - v_0)}{v_0} \frac{1}{z\sqrt{n}}.
$$

Thus with probability at least $1 - \delta$, we have

$$\sup_{v \in [v_0, V_0]} \left| \frac{1}{\sqrt{n}} \sum_{i=1}^{n} \frac{\varepsilon_i}{z\sqrt{\tau^2 + \varepsilon_i^2}} \right| \leq \frac{\sigma}{v_0} \sqrt{\frac{2\log(n/\delta)}{n}} + \frac{1}{z} \frac{\log(n/\delta)}{\sqrt{n}} + \frac{\sigma^2}{2v_0^2} \frac{z}{\sqrt{n}} + \frac{3(V_0 - v_0)}{v_0} \frac{1}{z\sqrt{n}}$$

$$\leq C \cdot \frac{V_0}{v_0} \cdot \frac{\log(n/\delta)}{z\sqrt{n}}$$

provided $z \lesssim \sqrt{\log(n/\delta)}$, where $C$ is a constant only depending on $\sigma^2/(v_0 V_0)$. When $v_0$ and $V_0$ are taken symmetrically around 1, $v_0 V_0$ is close to 1. Multiplying both sides by $z/\sqrt{n}$ finishes the proof. $\qquad \square$

**Lemma F.2.** Let $w_i$ be i.i.d. copies of $w$. For any $0 < \delta < 1$, with probability at least $1 - \delta$

$$\frac{1}{n} \sum_{i=1}^{n} \frac{\sqrt{\tau^2 + w_i^2} - \tau}{\sqrt{\tau^2 + w_i^2}} - \mathbb{E}\left( \frac{\sqrt{\tau^2 + w_i^2} - \tau}{\sqrt{\tau^2 + w_i^2}} \right) \leq \sqrt{\frac{\log(1/\delta) \mathbb{E} w_i^2}{n\tau^2}} + \frac{\log(1/\delta)}{3n}.$$

*Proof of Lemma F.2.* The random variables

$$Z_i = Z_i(\tau) := \frac{\sqrt{\tau^2 + w_i^2} - \tau}{\sqrt{\tau^2 + w_i^2}} = \frac{\sqrt{1 + w_i^2/\tau^2} - 1}{\sqrt{1 + w_i^2/\tau^2}}$$

with $\mu_z = \mathbb{E} Z_i$ and $\sigma_z^2 = \mathrm{var}(Z_i)$ are bounded i.i.d. random variables such that

$$0 \leq Z_i \leq 1 \wedge \frac{w_i^2}{2\tau^2}.$$

Moreover we have

$$\mathbb{E} Z_i^2 \leq \frac{\mathbb{E} w_i^2}{2\tau^2}, \ \sigma_z^2 := \mathrm{var}(Z_i) \leq \frac{\mathbb{E} w_i^2}{2\tau^2}.$$

For third and higher order absolute moments, we have

$$\mathbb{E} |Z_i|^k \leq \frac{\mathbb{E} w_i^2}{2\tau^2} \leq \frac{k!}{2} \cdot \frac{\mathbb{E} w_i^2}{2\tau^2} \cdot \left( \frac{1}{3} \right)^{k-2}, \ \text{for all integers } k \geq 3.$$

Therefore, using Lemma H.2 with $v = n \mathbb{E} w_i^2/(2\tau^2)$ and $c = 1/3$ acquires that for any $t > 0$

$$\mathbb{P}\left( \sum_{i=1}^{n} \frac{(1 + w_i^2/\tau^2)^{1/2} - 1}{(1 + w_i^2/\tau^2)^{1/2}} - \sum_{i=1}^{n} \mathbb{E}\left( \frac{(1 + w_i^2/\tau^2)^{1/2} - 1}{(1 + w_i^2/\tau^2)^{1/2}} \right) \geq -\sqrt{\frac{tn \mathbb{E} w_i^2}{\tau^2}} - \frac{t}{3} \right) \leq \exp(-t).$$

Taking $t = \log(1/\delta)$ acquires that for any $0 < \delta < 1$

$$\mathbb{P}\left( \frac{1}{n} \sum_{i=1}^{n} \frac{(1 + w_i^2/\tau^2)^{1/2} - 1}{(1 + w_i^2/\tau^2)^{1/2}} - \mathbb{E}\left( \frac{(1 + w_i^2/\tau^2)^{1/2} - 1}{(1 + w_i^2/\tau^2)^{1/2}} \right) > -\sqrt{\frac{\log(1/\delta) \mathbb{E} w_i^2}{n\tau^2}} - \frac{\log(1/\delta)}{3n} \right) > 1 - \delta.$$

This finishes the proof.

$\qquad \square$

**Lemma F.3.** For any $0 < \delta < 1$, we have with probability at least $1 - \delta$ that

$$\frac{1}{n} \sum_{i=1}^{n} \varepsilon_i^2 1\left( \varepsilon_i^2 \leq \frac{\tau^2}{2} - r^2 \right) \geq \frac{1}{n} \sum_{i=1}^{n} \mathbb{E} \varepsilon_i^2 1\left( \varepsilon_i^2 \leq \frac{\tau^2}{2} - r^2 \right) - \sigma_{\tau^2/2} \sqrt{\frac{\tau^2 \log(1/\delta)}{n}} - \frac{\tau^2 \log(1/\delta)}{6n}.$$

For any $0 < \delta < 1$, we have with probability at least $1 - \delta$ that

$$\frac{1}{n} \sum_{i=1}^{n} |\varepsilon_i| 1\left( \varepsilon_i^2 \leq \frac{\tau^2}{2} - r^2 \right) \leq \frac{1}{n} \sum_{i=1}^{n} \mathbb{E} |\varepsilon_i| 1\left( \varepsilon_i^2 \leq \frac{\tau^2}{2} - r^2 \right) + \sqrt{\frac{2\sigma_{\tau^2/2}^2 \log(1/\delta)}{n}} + \frac{\tau \log(1/\delta)}{3\sqrt{2}n}.$$

Consequently, we have, with probability at least $1 - 2\delta$, the above two inequalities hold simultaneously.

*Proof of Lemma F.3.* We prove the first two results and the last result directly follows from first two.

**First result.** Let $Z_i = \varepsilon_i^2 1\left(\varepsilon_i^2 \leq \underline{\tau}^2/2 - r^2\right)$. The random variables $Z_i$ with $\mu_z = \mathbb{E}Z_i$ and $\sigma_z^2 = \mathrm{var}(Z_i)$ are bounded i.i.d. random variables such that

$$
|Z_i| = \left|\varepsilon_i^2 1\left(\varepsilon_i^2 \leq \underline{\tau}^2/2 - r^2\right)\right| \leq \underline{\tau}^2/2,
$$
$$
|\mu_z| = |\mathbb{E}Z_i| = \left|\mathbb{E}\left(\varepsilon_i^2 1\left(\varepsilon_i^2 \leq \underline{\tau}^2/2 - r^2\right)\right)\right| \leq \sigma_{\underline{\tau}^2/2}^2,
$$
$$
\mathbb{E}Z_i^2 = \mathbb{E}\left(\varepsilon_i^4 1\left(\varepsilon_i^2 \leq \underline{\tau}^2/2 - r^2\right)\right) \leq \underline{\tau}^2 \sigma_{\underline{\tau}^2/2}^2/2,
$$
$$
\sigma_z^2 := \mathrm{var}(Z_i) = \mathbb{E}\left(Z_i - \mu_z\right)^2 \leq \underline{\tau}^2 \sigma_{\underline{\tau}^2/2}^2/2.
$$

For third and higher order absolute moments, we have

$$
\mathbb{E}|Z_i|^k = \mathbb{E}\left|\varepsilon_i^2 1\left(\varepsilon_i^2 \leq \underline{\tau}^2/2 - r^2\right)\right|^k \leq \frac{\underline{\tau}^2 \sigma_{\underline{\tau}^2/2}^2}{2}\left(\frac{\underline{\tau}^2}{2}\right)^{k-2} \leq \frac{k!}{2}\frac{\underline{\tau}^2 \sigma_{\underline{\tau}^2/2}^2}{2}\left(\frac{\underline{\tau}^2}{6}\right)^{k-2}, \text{ for all integers } k \geq 3.
$$

Using Lemma H.2 with $v = n\underline{\tau}^2 \sigma_{\underline{\tau}^2/2}^2/2$ and $c = \underline{\tau}^2/6$, we have for any $t > 0$

$$
\mathbb{P}\left(\sum_{i=1}^{n}\varepsilon_i^2 1\left(\varepsilon_i^2 \leq \frac{\underline{\tau}^2}{2} - r^2\right) - \sum_{i=1}^{n}\mathbb{E}\varepsilon_i^2 1\left(\varepsilon_i^2 \leq \frac{\underline{\tau}^2}{2} - r^2\right) \leq -\sqrt{n\underline{\tau}^2 \sigma_{\underline{\tau}^2/2}^2 t} - \frac{\underline{\tau}^2 t}{6}\right) \leq \exp\left(-t\right).
$$

Taking $t = \log(1/\delta)$ acquires the desired result.

**Second result.** With an abuse of notation, let $Z_i = |\varepsilon_i|1\left(\varepsilon_i^2 \leq \underline{\tau}^2/2 - r^2\right)$. The random variables $Z_i$ with $\mu_z = \mathbb{E}Z_i$ and $\sigma_z^2 = \mathrm{var}(Z_i)$ are bounded i.i.d. random variables such that

$$
|Z_i| = \left|\varepsilon_i 1\left(\varepsilon_i^2 \leq \underline{\tau}^2/2 - r^2\right)\right| \leq \underline{\tau}/\sqrt{2},
$$
$$
|\mu_z| = |\mathbb{E}Z_i| = \left|\mathbb{E}\left(|\varepsilon_i|1\left(\varepsilon_i^2 \leq \underline{\tau}^2/2 - r^2\right)\right)\right| \leq \sqrt{2}\sigma_{\underline{\tau}^2/2}^2/\underline{\tau},
$$
$$
\mathbb{E}Z_i^2 = \mathbb{E}\left(\varepsilon_i^2 1\left(\varepsilon_i^2 \leq \underline{\tau}^2/2 - r^2\right)\right) \leq \sigma_{\underline{\tau}^2/2}^2,
$$
$$
\sigma_z^2 := \mathrm{var}(Z_i) = \mathbb{E}\left(Z_i - \mu_z\right)^2 \leq \sigma_{\underline{\tau}^2/2}^2.
$$

For third and higher order absolute moments, we have

$$
\mathbb{E}|Z_i|^k = \mathbb{E}\left||\varepsilon_i|1\left(\varepsilon_i^2 \leq \underline{\tau}^2/2 - r^2\right)\right|^k \leq \sigma_{\underline{\tau}^2/2}^2\left(\frac{\underline{\tau}}{\sqrt{2}}\right)^{k-2} \leq \frac{k!}{2}\sigma_{\underline{\tau}^2/2}^2\left(\frac{\underline{\tau}}{3\sqrt{2}}\right)^{k-2}, \text{ for all integers } k \geq 3.
$$

Using Lemma H.2 with $v = n\sigma_{\underline{\tau}^2/2}^2$ and $c = \underline{\tau}/(3\sqrt{2})$, we have for any $t > 0$

$$
\mathbb{P}\left(\sum_{i=1}^{n}|\varepsilon_i|1\left(\varepsilon_i^2 \leq \frac{\underline{\tau}^2}{2} - r^2\right) - \sum_{i=1}^{n}\mathbb{E}|\varepsilon_i|1\left(\varepsilon_i^2 \leq \frac{\underline{\tau}^2}{2} - r^2\right) \geq \sqrt{2n\sigma_{\underline{\tau}^2/2}^2 t} + \frac{\underline{\tau} t}{3\sqrt{2}}\right) \leq \exp\left(-t\right).
$$

Taking $t = \log(1/\delta)$ acquires the desired result. $\qquad\square$

# G PROOFS FOR SECTION 3.2

This section collects proofs for results in Section 3.2.

## G.1 PROOF OF THEOREM 3.5

*Proof of Theorem 3.5.* First, the MoM estimator $\widehat{\mu}^{\mathrm{MoM}} = M(z_1, \ldots, z_k)$ is equivalent to

$$
\mathrm{argmin}\sum_{j=1}^{k}|z_j - \mu|.
$$

For any $x \in \mathbb{R}$, let $\ell(x) = |x|$ and define $L(x) = \mathbb{E}\ell'(x + Z)$ where $Z \sim \mathcal{N}(0, 1)$ and

$$
\ell'(x) = \begin{cases} 1, & \text{if } x > 0, \\ 0, & \text{if } x = 0, \\ -1, & \text{otherwise.} \end{cases}
$$

If the assumptions of Theorem 4 of Minsker (2019) are satisfied, we obtain, after some algebra, that

$$\sqrt{n}\left(\widehat{\mu}^{\text{MoM}} - \mu^*\right) \rightsquigarrow \mathcal{N}\left(0, \frac{\mathbb{E}(\ell'(Z))^2}{(L'(0))^2}\right).$$

Some algebra derives that

$$\frac{\mathbb{E}(\ell'(Z))^2}{(L'(0))^2} = \frac{\pi\sigma^2}{2}.$$

It remains to check the assumptions there. Assumptions (1), (4), and (5) trivially hold. Assumption (2) can be verified by using the following Berry-Esseen bound.

**Fact G.1.** Let $y_1, \ldots, y_m$ be i.i.d. random copies of $y$ with mean $\mu$, variance $\sigma^2$ and $\mathbb{E}|y-\mu|^{2+\iota} < \infty$ for some $\iota \in (0, 1]$. Then there exists an absolute constant $C$ such that

$$\sup_{t \in \mathbb{R}}\left|\mathbb{P}\left(\sqrt{m}\frac{\bar{y}-\mu}{\sigma} \leq t\right) - \Phi(t)\right| \leq C\frac{\mathbb{E}|y-\mu|^{2+\iota}}{\sigma^{2+\iota}m^{\iota/2}}.$$

It remains to check Assumption (3). Because $g(m) \lesssim m^{-\iota/2}$, $\sqrt{k}g(m) \lesssim \sqrt{k}m^{-\iota/2} \to 0$ if $k = o(n^{\iota/(1+\iota)})$ as $n \to \infty$. Thus Assumption (3) holds if $k = o(n^{\iota/(1+\iota)})$ and $k \to \infty$. This completes the proof. $\qquad\square$

### G.2 PROOF OF THEOREM 3.3

In this subsection, we state and prove a stronger result of Theorem 3.3, aka Theorem G.2. Theorem 3.3 can then be proved following the same proof under the assumption that $\mathbb{E}|\varepsilon_i|^{2+\iota} < \infty$ for any prefixed $0 < \iota \leq 1$.

**Theorem G.2.** Assume the same assumptions as in Theorem 3.1. Take $z^2 \geq 2\log(n)$. If $\mathbb{E}\varepsilon_i^4 < \infty$, then

$$\sqrt{n}\begin{bmatrix}\widehat{\mu} - \mu^* \\ \widehat{v} - v_*\end{bmatrix} \rightsquigarrow \mathcal{N}\left(0, \Sigma\right), \text{ where } \Sigma = \begin{bmatrix}\sigma^2 & \sigma\,\mathbb{E}\varepsilon_i^3/2 \\ \sigma\,\mathbb{E}\varepsilon_i^3/2 & (\sigma^2\mathbb{E}\varepsilon_i^4 - \sigma^6)/4\end{bmatrix}.$$

*Proof of Theorem G.2.* Now we are ready to analyze the self-tuned mean estimator $\widehat{\mu} = \widehat{\mu}(\widehat{v})$. For any $\delta \in (0, 1)$, following the proof of Theorem 3.1, we obtain with probability at least $1 - \delta$ that

$$|\widehat{\mu}(\widehat{v}) - \mu^*| \leq \sup_{v \in [v_0, V_0]}|\widehat{\mu}(v) - \mu^*| \leq 2C \cdot \frac{V_0^2}{v_0} \cdot \frac{\log(n/\delta)}{z\sqrt{n}}.$$

Taking $z^2 \geq \log(n/\delta)$ with $\delta = 1/n$ in the above inequality, we obtain $\widehat{\mu} \to \mu^*$ in probability. Theorem G.3 implies that $\widehat{v} \to \sigma$ in probability. Thus we have $\|\widehat{\theta} - \theta^*\|_2 \to 0$ in probability, where

$$\widehat{\theta} = (\widehat{\mu}, \widehat{v})^{\mathrm{T}}, \text{ and } \theta^* = (\mu^*, \sigma)^{\mathrm{T}}.$$

Using the Taylor's theorem for vector-valued functions, we obtain

$$\nabla L_n(\widehat{\theta}) = 0 = \nabla L_n(\theta^*) + H_n(\theta^*)(\widehat{\theta} - \theta^*) + \frac{R_2(\theta)}{2}(\widehat{\theta} - \theta^*)^{\otimes 2},$$

where $\otimes$ indicates the tensor product. Let $\tau_\sigma = \sigma\sqrt{n}/z$. We say that $X_n$ and $Y_n$ are asymptotically equivalent, denoted as $X_n \simeq Y_n$, if both $X_n$ and $Y_n$ converge in distribution to some same random

variable/vector $Z$. Rearranging, we obtain

$$\sqrt{n}\left(\widehat{\theta}-\theta^*\right) \simeq \left[H_n(\theta^*)\right]^{-1}\left(-\sqrt{n}\,\nabla L_n(\theta^*)\right)$$

$$= \begin{bmatrix} \frac{\sqrt{n}}{z}\cdot\frac{1}{n}\sum_{i=1}^n\frac{\tau_\sigma^2}{(\tau_\sigma^2+\varepsilon_i^2)^{3/2}} & \frac{n}{z^2}\cdot\frac{1}{n}\sum_{i=1}^n\frac{\tau_\sigma\varepsilon_i}{(\tau_\sigma^2+\varepsilon_i^2)^{3/2}} \\ \frac{n}{z^2}\cdot\frac{1}{n}\sum_{i=1}^n\frac{\tau_\sigma\varepsilon_i}{(\tau_\sigma^2+\varepsilon_i^2)^{3/2}} & \frac{n^{3/2}}{z^3}\cdot\frac{1}{n}\sum_{i=1}^n\frac{\varepsilon_i^3}{(\tau_\sigma^2+\varepsilon_i^2)^{3/2}} \end{bmatrix}^{-1}$$

$$\begin{bmatrix} \sqrt{n}\cdot\frac{1}{n}\sum_{i=1}^n\frac{\tau_\sigma\varepsilon_i}{\sigma\sqrt{\tau_\sigma^2+\varepsilon_i^2}} \\ \sqrt{n}\cdot\frac{n}{z^2}\frac{1}{n}\sum_{i=1}^n\frac{\sqrt{1+\varepsilon_i^2/\tau_\sigma^2}-1}{\sqrt{1+\varepsilon_i^2/\tau_\sigma^2}}-\sqrt{n}\cdot a \end{bmatrix}$$

$$\simeq \begin{bmatrix} \sigma & 0 \\ 0 & \sigma^3 \end{bmatrix}\begin{bmatrix} \sqrt{n}\cdot\frac{1}{n}\sum_{i=1}^n\frac{\tau_\sigma\varepsilon_i}{\sigma\sqrt{\tau_\sigma^2+\varepsilon_i^2}} \\ \sqrt{n}\cdot\frac{n}{z^2}\frac{1}{n}\sum_{i=1}^n\frac{\sqrt{1+\varepsilon_i^2/\tau_\sigma^2}-1}{\sqrt{1+\varepsilon_i^2/\tau_\sigma^2}}-\sqrt{n}\cdot a \end{bmatrix}$$

$$= \begin{bmatrix} \sigma & 0 \\ 0 & \sigma^3 \end{bmatrix}\begin{bmatrix} \mathrm{I} \\ \mathrm{II} \end{bmatrix},$$

where the second $\simeq$ uses the fact that

$$H_n(\theta^*)\xrightarrow{\text{a.s.}}\begin{bmatrix} \frac{1}{\sigma} & 0 \\ 0 & \frac{1}{\sigma^3} \end{bmatrix}.$$

We proceed to derive the asymptotic property of $(\mathrm{I},\mathrm{II})^{\mathsf{T}}$. For I, we have

$$\mathrm{I} = \sqrt{n}\cdot\left(\frac{1}{n}\sum_{i=1}^n\frac{\tau_\sigma\varepsilon_i}{\sigma\sqrt{\tau_\sigma^2+\varepsilon_i^2}}-\mathbb{E}\left[\frac{\tau_\sigma\varepsilon_i}{\sigma\sqrt{\tau_\sigma^2+\varepsilon_i^2}}\right]\right)+\sqrt{n}\cdot\mathbb{E}\left[\frac{\tau_\sigma\varepsilon_i}{\sigma\sqrt{\tau_\sigma^2+\varepsilon_i^2}}\right]$$

$$\rightsquigarrow\mathcal{N}\left(0,\lim_{n\to\infty}\mathrm{var}\left[\frac{\tau_\sigma\varepsilon_i}{\sigma\sqrt{\tau_\sigma^2+\varepsilon_i^2}}\right]\right)+\lim_{n\to\infty}\sqrt{n}\cdot\mathbb{E}\left[\frac{\tau_\sigma\varepsilon_i}{\sigma\sqrt{\tau_\sigma^2+\varepsilon_i^2}}\right].$$

It remains to calculate

$$\lim_{n\to\infty}\mathbb{E}\left(\frac{\sqrt{n}\tau_\sigma\varepsilon_i}{\sqrt{\tau_\sigma^2+\varepsilon_i^2}}\right)\quad\text{and}\quad\lim_{n\to\infty}\mathrm{var}\left[\frac{\tau\varepsilon_i}{\sqrt{\tau_\sigma^2+\varepsilon_i^2}}\right].$$

For the former term, if there exists some $0<\iota\le 1$ such that $\mathbb{E}|\varepsilon_i|^{2+\iota}<\infty$, using the fact that $\mathbb{E}\varepsilon_i=0$, we have

$$\left|\mathbb{E}\left(\frac{\sqrt{n}\tau_\sigma\varepsilon_i}{\sqrt{\tau_\sigma^2+\varepsilon_i^2}}\right)\right| = \sqrt{n}\tau_\sigma\cdot\left|\mathbb{E}\left\{\frac{-\varepsilon_i/\tau_\sigma}{\sqrt{1+\varepsilon_i^2/\tau_\sigma^2}}\right\}\right| = \sqrt{n}\tau_\sigma\cdot\left|\mathbb{E}\left\{\frac{\tau_\sigma^{-1}\varepsilon_i\left(\sqrt{1+\varepsilon_i^2/\tau_\sigma^2}-1\right)}{\sqrt{1+\varepsilon_i^2/\tau_\sigma^2}}\right\}\right|$$

$$\le \frac{\sqrt{n}\tau_\sigma}{2}\cdot\mathbb{E}\left|\frac{\varepsilon_i^3/\tau_\sigma^3}{\sqrt{1+\varepsilon_i^2/\tau_\sigma^2}}\right| \le \frac{\sqrt{n}\tau_\sigma}{2}\cdot\frac{\mathbb{E}|\varepsilon_i|^{2+\iota}}{\tau_\sigma^{2+\iota}}$$

$$\le \frac{\sqrt{n}\,\mathbb{E}|\varepsilon_i|^{2+\iota}}{2\tau_\sigma^{1+\iota}}\to 0, \tag{G.1}$$

where the first inequality uses Lemma H.4 (ii) with $r=1/2$, that is, $\sqrt{1+x}\le 1+x/2$ for $x\ge -1$. For the second term, we have

$$\lim_{n\to\infty}\mathrm{var}\left[\frac{\tau_\sigma\varepsilon_i}{\sqrt{\tau_\sigma^2+\varepsilon_i^2}}\right] = \lim_{n\to\infty}\mathbb{E}\left[\frac{\tau_\sigma^2\varepsilon_i^2}{\tau_\sigma^2+\varepsilon_i^2}\right] = \sigma^2,$$

by the dominated convergence theorem. Thus

$$\mathrm{I}\rightsquigarrow\mathcal{N}(0,1).$$

For II, recall $a=1/2$ and using the facts that

$$\lim_{n\to\infty}\frac{n}{z^2}\cdot\mathbb{E}\left(\frac{\sqrt{1+\varepsilon_i^2/\tau_\sigma^2}-1}{\sqrt{1+\varepsilon_i^2/\tau_\sigma^2}}\right) = \lim_{n\to\infty}\frac{n}{2\tau_\sigma^2z^2}\cdot\mathbb{E}\left(\frac{1}{\sqrt{1+\varepsilon_i^2/\tau_\sigma^2}}\cdot\frac{\sqrt{1+\varepsilon_i^2/\tau_\sigma^2}-1}{1/(2\tau_\sigma^2)}\right) = \frac{1}{2},$$

$$\lim_{n\to\infty}\sqrt{n}\cdot\left(\frac{n}{z^2}\cdot\mathbb{E}\left(\frac{\sqrt{1+\varepsilon_i^2/\tau_\sigma^2}-1}{\sqrt{1+\varepsilon_i^2/\tau_\sigma^2}}\right)-\frac{1}{2}\right) = 0,$$

we have

$$\text{II} = \sqrt{n} \cdot \frac{n}{z^2} \cdot \frac{1}{n} \sum_{i=1}^{n} \frac{\sqrt{1 + \varepsilon_i^2/\tau_\sigma^2} - 1}{\sqrt{1 + \varepsilon_i^2/\tau_\sigma^2}} - \sqrt{n} \cdot \frac{1}{2}$$

$$\simeq \sqrt{n} \cdot \frac{1}{n} \sum_{i=1}^{n} \left( \frac{n}{z^2} \cdot \frac{\sqrt{1 + \varepsilon_i^2/\tau_\sigma^2} - 1}{\sqrt{1 + \varepsilon_i^2/\tau_\sigma^2}} - \mathbb{E} \left( \frac{n}{z^2} \cdot \frac{\sqrt{1 + \varepsilon_i^2/\tau_\sigma^2} - 1}{\sqrt{1 + \varepsilon_i^2/\tau_\sigma^2}} \right) \right)$$

$$\simeq \mathcal{N} \left( 0, \lim_{n \to \infty} \text{var} \left( \frac{n}{z^2} \cdot \frac{\sqrt{1 + \varepsilon_i^2/\tau_\sigma^2} - 1}{\sqrt{1 + \varepsilon_i^2/\tau_\sigma^2}} \right) \right).$$

If $\mathbb{E}\varepsilon_i^4 < \infty$, then

$$\lim_{n \to \infty} \text{var} \left( \frac{n}{z^2} \cdot \frac{\sqrt{1 + \varepsilon_i^2/\tau_\sigma^2} - 1}{\sqrt{1 + \varepsilon_i^2/\tau_\sigma^2}} \right) = \frac{\mathbb{E}\varepsilon_i^4}{4\sigma^4} - \frac{1}{4},$$

and thus $\text{II} \simeq \mathcal{N} \left( 0, (\mathbb{E}\varepsilon_i^4/\sigma^4 - 1)/4 \right)$. For the cross covariance, we have

$$\lim_{n \to \infty} \text{cov} \left( \frac{\tau_\sigma \varepsilon_i}{\sigma \sqrt{\tau_\sigma^2 + \varepsilon_i^2}}, \frac{n}{z^2} \cdot \frac{\sqrt{1 + \varepsilon_i^2/\tau_\sigma^2} - 1}{\sqrt{1 + \varepsilon_i^2/\tau_\sigma^2}} \right)$$

$$= \lim_{n \to \infty} \mathbb{E} \left( \frac{\tau_\sigma \varepsilon_i}{\sigma \sqrt{\tau_\sigma^2 + \varepsilon_i^2}} \cdot \frac{n}{z^2} \cdot \frac{\sqrt{1 + \varepsilon_i^2/\tau_\sigma^2} - 1}{\sqrt{1 + \varepsilon_i^2/\tau_\sigma^2}} \right)$$

$$= \frac{\mathbb{E}\varepsilon_i^3}{2\sigma^3}.$$

Thus

$$\sqrt{n} \left( \widehat{\theta} - \theta^* \right) \rightsquigarrow \mathcal{N}(0, \Sigma),$$

where

$$\Sigma = \begin{bmatrix} \sigma & 0 \\ 0 & \sigma^3 \end{bmatrix} \begin{bmatrix} 1 & \mathbb{E}\varepsilon_i^3/(2\sigma^3) \\ \mathbb{E}\varepsilon_i^3/(2\sigma^3) & (\mathbb{E}\varepsilon_i^4/\sigma^4 - 1)/4 \end{bmatrix} \begin{bmatrix} \sigma & 0 \\ 0 & \sigma^3 \end{bmatrix} = \begin{bmatrix} \sigma^2 & \sigma\mathbb{E}\varepsilon_i^3/2 \\ \sigma\mathbb{E}\varepsilon_i^3/2 & (\sigma^2\mathbb{E}\varepsilon_i^4 - \sigma^6)/4 \end{bmatrix}.$$

Therefore, for $\widehat{\mu}$ only, we have

$$\sqrt{n} \left( \widehat{\mu} - \mu^* \right) \rightsquigarrow \mathcal{N}(0, \sigma^2).$$

$\square$

## G.3 Consistency of $\widehat{v}$

This subsection proves that $\widehat{v}$ is a consistent estimator of $\sigma$. Recall that

$$\nabla_v L_n(\mu, v) = \frac{n}{z^2} \cdot \frac{1}{n} \sum_{i=1}^{n} \left( \frac{\tau}{\sqrt{\tau^2 + (y_i - \mu)^2}} - 1 \right) + a$$

where $a = 1/2$. We emphasize that the following proof only needs the second moment assumption $\sigma^2 = \mathbb{E}\varepsilon_i^2 < \infty$.

**Theorem G.3** (Consistency of $\widehat{v}$). Assume the same assumptions as in Theorem 3.1. Take $z^2 \geq \log(n)$. Then

$$\widehat{v} \longrightarrow \sigma \quad \text{in probability.}$$

*Proof of Theorem G.3.* By the proof of Theorem 3.1, we obtain with probability at least $1 - \delta$ that the following two results hold simultaneously:

$$\sup_{v \in [v_0, V_0]} |\widehat{\mu}(v) - \mu^*| \leq 2C \cdot \frac{V_0^2}{v_0} \cdot \frac{\log(n/\delta)}{z\sqrt{n}} =: r, \tag{G.2}$$

$$v_0 < c_0 \sigma_{\tau_{v_0}^2 - 1} \leq \widehat{v} \leq C_0 \sigma < V_0, \tag{G.3}$$

provided that $z^2 \geq \log(5/\delta)$ and $n$ is large enough. Therefore, the constraint in the optimization problem (3.1) is not active, and thus

$$\nabla_v L_n(\widehat{\mu}, \widehat{v}) = 0.$$

Using Lemma G.4 together with the equality above, we obtain with probability at least $1 - \delta$ that

$$\frac{c_0}{V_0^3}|\widehat{v} - \sigma|^2 \leq \frac{c_0}{\widehat{v}^3 \vee \sigma^3}|\widehat{v} - \sigma|^2 \leq \rho_\ell |\widehat{v} - \sigma|^2$$

$$\leq \langle \nabla_v L_n(\widehat{\mu}, \widehat{v}) - \nabla_v L_n(\widehat{\mu}, \sigma), \widehat{v} - \sigma \rangle$$

$$\leq |\nabla_v L_n(\widehat{\mu}, \sigma)| \, |\widehat{v} - \sigma|$$

$$\leq \left| \frac{n}{z^2} \cdot \frac{1}{n} \sum_{i=1}^n \left( \frac{\tau_\sigma}{\sqrt{\tau_\sigma^2 + (y_i - \widehat{\mu})^2}} - 1 \right) + a \right| |\widehat{v} - \sigma|.$$

Plugging (G.2) into the above inequality and canceling $|\widehat{v} - \sigma|$ on both sides, we obtain with probability at least $1 - 2\delta$ that

$$\frac{c_0}{V_0^3}|\widehat{v} - \sigma| \leq \left| \frac{n}{z^2} \cdot \frac{1}{n} \sum_{i=1}^n \left( \frac{\tau_\sigma}{\sqrt{\tau_\sigma^2 + (y_i - \widehat{\mu})^2}} - 1 \right) + a \right|$$

$$\leq \sup_{\mu \in \mathbb{B}_r(\mu^*)} \left| \frac{n}{z^2} \cdot \frac{1}{n} \sum_{i=1}^n \left( \frac{\tau_\sigma}{\sqrt{\tau_\sigma^2 + (y_i - \mu)^2}} - 1 \right) + a \right|$$

$$= \frac{n}{z^2} \cdot \sup_{\mu \in \mathbb{B}_r(\mu^*)} \left| \frac{1}{n} \sum_{i=1}^n \left( \frac{\tau_\sigma}{\sqrt{\tau_\sigma^2 + (y_i - \mu)^2}} - 1 \right) + \frac{az^2}{n} \right|$$

$$\leq \frac{n}{z^2} \cdot \sup_{\mu \in \mathbb{B}_r(\mu^*)} \left| \frac{1}{n} \sum_{i=1}^n \left( 1 - \frac{\tau_\sigma}{\sqrt{\tau_\sigma^2 + (y_i - \mu)^2}} \right) - \mathbb{E}\left( 1 - \frac{\tau_\sigma}{\sqrt{\tau_\sigma^2 + (y_i - \mu)^2}} \right) \right|$$

$$+ \frac{n}{z^2} \cdot \sup_{\mu \in \mathbb{B}_r(\mu^*)} \left| \mathbb{E}\left( 1 - \frac{\tau_\sigma}{\sqrt{\tau_\sigma^2 + (y_i - \mu)^2}} \right) - \frac{az^2}{n} \right|$$

$$=: \mathrm{I} + \mathrm{II}.$$

It remains to bound terms I and II. We start with term II. Let $r_i^2 = (y_i - \mu)^2$. We have

$$\mathrm{II} = \frac{n}{z^2} \cdot \sup_{\mu \in \mathbb{B}_r(\mu^*)} \left| \mathbb{E}\left( 1 - \frac{\tau_\sigma}{\sqrt{\tau_\sigma^2 + (y_i - \mu)^2}} \right) - \frac{az^2}{n} \right|$$

$$= \max \left\{ \sup_{\mu \in \mathbb{B}_r(\mu^*)} \left( \frac{n}{z^2} \cdot \mathbb{E}\frac{\sqrt{1 + r_i^2/\tau_\sigma^2} - 1}{\sqrt{1 + r_i^2/\tau_\sigma^2}} - a \right), \sup_{\mu \in \mathbb{B}_r(\mu^*)} \left( a - \frac{n}{z^2} + \mathbb{E}\frac{1}{\sqrt{1 + r_i^2/\tau_\sigma^2}} \right) \right\}$$

$$=: \mathrm{II}_1 \vee \mathrm{II}_2.$$

In order to bound II, we bound $\mathrm{II}_1$ and $\mathrm{II}_2$ respectively. For term $\mathrm{II}_1$, using Lemma H.4 (ii), aka $(1 + x)^r \leq 1 + rx$ for $x \geq -1$ and $r \in (0, 1)$, and $a = 1/2$, we have

$$\mathrm{II}_1 = \sup_{\mu \in \mathbb{B}_r(\mu^*)} \left( \frac{n}{z^2} \cdot \mathbb{E}\frac{\sqrt{1 + r_i^2/\tau_\sigma^2} - 1}{\sqrt{1 + r_i^2/\tau_\sigma^2}} - a \right)$$

$$\leq \sup_{\mu \in \mathbb{B}_r(\mu^*)} \left\{ \frac{n}{z^2} \cdot \left( 1 + \mathbb{E}\frac{r_i^2}{2\tau_\sigma^2} - 1 \right) - a \right\}$$

$$\leq \frac{n}{z^2} \cdot \mathbb{E}\frac{\varepsilon_i^2 + 2r|\varepsilon_i| + r^2|}{2\tau_\sigma^2} - \frac{1}{2} \qquad (a = 1/2)$$

$$\leq \frac{r}{\sigma}\left( 1 + \frac{r}{2\sigma} \right)$$

$$\leq \frac{2r}{\sigma}$$

if $n$ is large enough such that $r \leq 2\sigma$. To bound $\mathrm{II}_2$, we need Lemma D.1. Specifically, for any $0 \leq \gamma < 1$, we have

$$(1+x)^{-1} \leq 1 - (1-\gamma)x, \text{ for any } 0 \leq x \leq \frac{\gamma}{1-\gamma}.$$

Using this result, we obtain

$$
\begin{aligned}
\mathbb{E}\frac{1}{\sqrt{1+r_i^2/\tau_\sigma^2}} &\leq \sqrt{\mathbb{E}\frac{1}{1+r_i^2/\tau_\sigma^2}} \qquad\qquad\qquad\qquad\qquad\qquad\quad \text{(concavity of } \sqrt{x}) \\
&\leq \sqrt{\mathbb{E}\left\{\left(1 - \frac{(1-\gamma)r_i^2}{\tau_\sigma^2}\right)1\left(\frac{r_i^2}{\tau_\sigma^2} \leq \frac{\gamma}{1-\gamma}\right) + \frac{1}{1+r_i^2/\tau_\sigma^2}1\left(\frac{r_i^2}{\tau_\sigma^2} > \frac{\gamma}{1-\gamma}\right)\right\}} \\
&\leq \sqrt{1 - (1-\gamma)\,\mathbb{E}\left(\frac{r_i^2}{\tau_\sigma^2}1\left(\frac{r_i^2}{\tau_\sigma^2} \leq \frac{\gamma}{1-\gamma}\right)\right)} \qquad\qquad\qquad \text{(Lemma D.1)} \\
&\leq \sqrt{1 - (1-\gamma)\,\mathbb{E}\left(\frac{r_i^2}{\tau_\sigma^2}1\left(\frac{r_i^2}{\tau_\sigma^2} \leq \frac{\gamma}{1-\gamma}\right)\right)} \\
&\leq \sqrt{1 - (1-\gamma)\,\mathbb{E}\left(\frac{\varepsilon_i^2 - 2r|\varepsilon_i| + r^2}{\tau_\sigma^2}1\left(\frac{2(\varepsilon_i^2 + r^2)}{\tau_\sigma^2} \leq \frac{\gamma}{1-\gamma}\right)\right)} \\
&\qquad\qquad\qquad\qquad\qquad\qquad\qquad\qquad\qquad\qquad\qquad\qquad (\forall\, \mu \in \mathbb{B}_r(\mu^*)) \\
&\leq 1 - \frac{1-\gamma}{2}\,\mathbb{E}\left(\frac{\varepsilon_i^2 - 2r|\varepsilon_i| + r^2}{\tau_\sigma^2}1\left(\frac{2(\varepsilon_i^2 + r^2)}{\tau_\sigma^2} \leq \frac{\gamma}{1-\gamma}\right)\right),
\end{aligned}
$$

where the first inequality uses the concavity of $\sqrt{x}$, the third inequality uses Lemma D.1, and the last inequality uses the inequality that $(1+x)^{-1} \leq 1 - x/2$ for $x \in [0,1]$, aka Lemma H.4 (iii) with $r = -1$, provided that

$$(1-\gamma)\,\mathbb{E}\left(\frac{\varepsilon_i^2 - 2r|\varepsilon_i| - r^2}{\tau_\sigma^2}1\left(\frac{2(\varepsilon_i^2 + r^2)}{\tau_\sigma^2} \leq \frac{\gamma}{1-\gamma}\right)\right) \leq (1-\gamma)\frac{\sigma^2 - 2r\sigma - r^2}{\tau_\sigma^2} \leq 1.$$

Thus term $\mathrm{II}_2$ can be bounded as

$$
\begin{aligned}
\mathrm{II}_2 &= \sup_{\mu \in \mathbb{B}_r(\mu^*)}\left(a - \frac{n}{z^2} + \frac{n}{z^2}\cdot\mathbb{E}\frac{1}{\sqrt{1+r_i^2/\tau_\sigma^2}}\right) \\
&\leq a - \frac{n}{z^2} + \frac{n}{z^2}\cdot\left\{1 - \frac{1-\gamma}{2}\,\mathbb{E}\left(\frac{\varepsilon_i^2 - 2r|\varepsilon_i| + r^2}{\tau_\sigma^2}1\left(\frac{2(\varepsilon_i^2 + r^2)}{\tau_\sigma^2} \leq \frac{\gamma}{1-\gamma}\right)\right)\right\} \\
&\leq a - \frac{1-\gamma}{2\sigma^2}\cdot\mathbb{E}\varepsilon_i^2 + \frac{1-\gamma}{2\sigma^2}\cdot 2r\cdot\mathbb{E}(|\varepsilon_i|) \\
&\leq a - \frac{1-\gamma}{2} + \frac{r(1-\gamma)}{\sigma} \\
&= \frac{\gamma}{2} + \frac{r(1-\gamma)}{\sigma}. \qquad\qquad\qquad\qquad\qquad\qquad\qquad\qquad\qquad (a = 1/2)
\end{aligned}
$$

Combining the upper bound for $\mathrm{II}_1$ and $\mathrm{II}_2$ and using the fact that, we obtain

$$\mathrm{II} \leq \max\{\mathrm{II}_1, \mathrm{II}_2\} \leq \frac{\gamma}{2} + \frac{2r}{\sigma} \to 0,$$

if $\gamma = \gamma(n) \to 0$.

We proceed to bound I. Recall that

$$\mathrm{I} = \frac{n}{z^2}\cdot\sup_{\mu \in \mathbb{B}_r(\mu^*)}\left|\frac{1}{n}\sum_{i=1}^{n}\left(1 - \frac{\tau_\sigma}{\sqrt{\tau_\sigma^2 + (y_i - \mu)^2}}\right) - \mathbb{E}\left(1 - \frac{\tau_\sigma}{\sqrt{\tau_\sigma^2 + (y_i - \mu)^2}}\right)\right|.$$

For any $0 < \epsilon \leq 2r$, there exists an $\epsilon$-cover $\mathcal{N} \subseteq \mathbb{B}_r(\mu^*)$ of $\mathbb{B}_r(\mu^*)$ such that $|\mathcal{N}| \leq 6r/\epsilon$. Then for any $\mu \in \mathbb{B}_r(\mu^*)$ there exists a $\omega \in \mathcal{N}$ such that $|\omega - \mu| \leq \gamma$, and

$$
\left| \frac{1}{n} \sum_{i=1}^{n} \left( 1 - \frac{\tau_\sigma}{\sqrt{\tau_\sigma^2 + (y_i - \mu)^2}} \right) - \mathbb{E} \left( 1 - \frac{\tau_\sigma}{\sqrt{\tau_\sigma^2 + (y_i - \mu)^2}} \right) \right|
$$

$$
= \left| \frac{1}{n} \sum_{i=1}^{n} \frac{\sqrt{1 + (y_i - \mu)^2/\tau_\sigma^2} - 1}{\sqrt{1 + (y_i - \mu)^2/\tau_\sigma^2}} - \mathbb{E} \frac{\sqrt{1 + (y_i - \mu)^2/\tau_\sigma^2} - 1}{\sqrt{1 + (y_i - \mu)^2/\tau_\sigma^2}} \right|
$$

$$
\leq \left| \frac{1}{n} \sum_{i=1}^{n} \frac{\sqrt{1 + (y_i - \omega)^2/\tau_\sigma^2} - 1}{\sqrt{1 + (y_i - \omega)^2/\tau_\sigma^2}} - \mathbb{E} \frac{\sqrt{1 + (y_i - \omega)^2/\tau_\sigma^2} - 1}{\sqrt{1 + (y_i - \omega)^2/\tau_\sigma^2}} \right|
$$

$$
+ \left| \frac{1}{n} \sum_{i=1}^{n} \frac{\sqrt{1 + (y_i - \mu)^2/\tau_\sigma^2} - 1}{\sqrt{1 + (y_i - \mu)^2/\tau_\sigma^2}} - \frac{1}{n} \sum_{i=1}^{n} \frac{\sqrt{1 + (y_i - \omega)^2/\tau_\sigma^2} - 1}{\sqrt{1 + (y_i - \omega)^2/\tau_\sigma^2}} \right|
$$

$$
+ \left| \mathbb{E} \frac{\sqrt{1 + (y_i - \mu)^2/\tau_\sigma^2} - 1}{\sqrt{1 + (y_i - \mu)^2/\tau_\sigma^2}} - \mathbb{E} \frac{\sqrt{1 + (y_i - \omega)^2/\tau_\sigma^2} - 1}{\sqrt{1 + (y_i - \omega)^2/\tau_\sigma^2}} \right|
$$

$$
= \mathrm{I}_1 + \mathrm{I}_2 + \mathrm{I}_3.
$$

For $\mathrm{I}_1$, using Lemma F.2 acquires with probability at least $1 - 2\delta$ that

$$
\mathrm{I}_1 \leq \sqrt{\frac{\mathbb{E}(y_i - \omega)^2 \, \log(1/\delta)}{n\tau_\sigma^2}} + \frac{\log(1/\delta)}{3n}
$$

$$
\leq \sqrt{\frac{2(\sigma^2 + r^2) \log(1/\delta)}{n\tau_\sigma^2}} + \frac{\log(1/\delta)}{3n}
$$

$$
\leq \frac{2z\sqrt{\log(1/\delta)}}{n} + \frac{\log(1/\delta)}{3n}
$$

provided $r^2 \leq \sigma^2$. Let

$$
g(x) = -\frac{1}{n} \sum_{i=1}^{n} \frac{\tau}{\sqrt{\tau^2 + (x + \varepsilon_i)^2}}.
$$

Using the mean value theorem and the inequality that $|x/(1 + x^2)^{3/2}| \leq 1/2$, we obtain

$$
|g(x) - g(y)| = \left| \frac{1}{n} \sum_{i=1}^{n} \frac{(\widetilde{x} + \varepsilon_i)/\tau_\sigma}{(1 + (\widetilde{x} + \varepsilon_i)^2/\tau_\sigma^2)^{3/2}} \cdot \frac{x - y}{\tau_\sigma} \right| \leq \frac{|x - y|}{2\tau_\sigma},
$$

where $\widetilde{x}$ is some convex combination of $x$ and $y$. Then we have

$$
\mathrm{I}_2 = \left| \frac{1}{n} \sum_{i=1}^{n} \frac{(\widetilde{\Delta} + \varepsilon_i)/\tau_\sigma}{(1 + (\widetilde{\Delta} + \varepsilon_i)^2/\tau_\sigma^2)^{3/2}} \cdot \frac{\Delta_\mu - \Delta_\omega}{\tau_\sigma} \right| \leq \frac{\epsilon}{2\tau_\sigma}
$$

where $\widetilde{\Delta}$ is some convex combination of $\Delta_w = \mu^* - w$ and $\Delta_\mu = \mu^* - \mu$. For $\mathrm{II}_3$, a similar argument for bounding $\mathrm{II}_2$ yields

$$
\mathrm{I}_3 = \left| \mathbb{E} \left( \frac{(\widetilde{\Delta} + \varepsilon_i)/\tau_\sigma}{(1 + (\widetilde{\Delta} + \varepsilon_i)^2/\tau_\sigma^2)^{3/2}} \right) \cdot \frac{\Delta_\mu - \Delta_\omega}{\tau_\sigma} \right|
$$

$$
\leq \mathbb{E}|\widetilde{\Delta} + \varepsilon_i| \cdot \frac{\epsilon}{\tau_\sigma^2}
$$

$$
\leq \frac{\epsilon\sqrt{2(r^2 + \sigma^2)}}{\tau_\sigma^2},
$$

where the last inequality uses Jensen's inequality, i.e. $\mathbb{E}|\widetilde{\Delta} + \varepsilon_i| \leq \sqrt{\mathbb{E}(\widetilde{\Delta} + \varepsilon_i^2)} \leq \sqrt{2(r^2 + \sigma^2)}$. Putting the above pieces together and using the union bound, we obtain with probability at least

$$1 - 12\epsilon^{-1}r\delta$$

$$\mathrm{I} \leq \frac{n}{z^2} \cdot \sup_{\omega \in \mathcal{N}} \left| \frac{1}{n} \sum_{i=1}^{n} \frac{\sqrt{1 + (y_i - \omega)^2/\tau_\sigma^2} - 1}{\sqrt{1 + (y_i - \omega)^2/\tau_\sigma^2}} - \mathbb{E} \frac{\sqrt{1 + (y_i - \omega)^2/\tau_\sigma^2} - 1}{\sqrt{1 + (y_i - \omega)^2/\tau_\sigma^2}} \right|$$

$$+ \frac{n}{z^2} \cdot \frac{\epsilon}{2\tau_\sigma} \left( 1 + \frac{2\sqrt{2(r^2 + \sigma^2)}}{\tau_\sigma} \right)$$

$$\leq \frac{2\sqrt{\log(1/\delta)}}{z} + \frac{\log(1/\delta)}{3z^2} + \frac{\epsilon\sqrt{n}}{\sigma z},$$

provided that

$$2\sqrt{2(r^2 + \sigma^2)} \leq \tau_\sigma.$$

Putting above results together, we obtain with probability at least $1 - (12r/\epsilon + 2)\delta$ that

$$|\widehat{v} - \sigma| \lesssim \mathrm{I} + \mathrm{II}$$

$$\leq \frac{2\sqrt{\log(1/\delta)}}{z} + \frac{\log(1/\delta)}{3z^2} + \frac{\epsilon\sqrt{n}}{\sigma z} + \frac{\gamma}{2} + \frac{2r}{\sigma}.$$

Let $C' = 24CV_0^2/v_0$. Therefore, taking $\epsilon = 1/\sqrt{n}$, $\delta = 1/\log n$, and $z^2 \geq \log(n)$, we obtain with probability at least

$$1 - \frac{C'\left(\sqrt{\log n} + \log\log n/\sqrt{\log n}\right) + 2}{\log n}$$

that

$$|\widehat{v} - \sigma| \lesssim \sqrt{\frac{\log\log n}{\log n}} + \frac{\log\log n}{\log n} + \frac{1}{\sqrt{\log n}} + \gamma + r \to 0.$$

Therefore $\widehat{v} \to \sigma$ in probability. This finishes the proof.

$$\square$$

### G.4 LOCAL STRONG CONVEXITY IN $v$

In this section, we first present the local strong convexity of the empirical loss function with respect to $v$ uniformly over a neighborhood of $\mu^*$.

**Lemma G.4** (Local strong convexity in $v$). Let $\mathbb{B}_r(\mu^*) = \{\mu : |\mu - \mu^*| \leq r\}$. Assume $r = r(n) = o(1)$. Let $0 < \delta < 1$ and $n$ is sufficiently large. Take $\varpi$ such that $\max\{\varpi r\sqrt{n}, \varpi\} \to 0$ and $\varpi\sqrt{n} \to \infty$. Then, with probability at least $1 - \delta$, we have

$$\inf_{\mu \in \mathbb{B}_r(\mu^*)} \frac{\langle \nabla_v L_n(\mu, v) - \nabla_v L_n(\mu, v_*), v - \sigma \rangle}{|v - \sigma|^2} \geq \rho_\ell = \frac{\sigma_{c\varpi^2 n/(4z^2)}^2}{2(v^3 \vee \sigma^3)} \geq \frac{c_0}{v^3 \vee \sigma^3},$$

where $c$ and $c_0$ are some constants.

*Proof of Lemma G.4.* Recall $\tau = v\sqrt{n}/z$. For notational simplicity, write $\tau_\sigma = \sigma\sqrt{n}/z$, $\tau_{v_0} = v_0\sqrt{n}/z$, $\tau_\varpi = \varpi\sqrt{n}/z$, and $\Delta = \mu^* - \mu$. It follows that

$$\langle \nabla_v L_n(\mu, v) - \nabla_v L_n(\mu, \sigma), v - \sigma \rangle = \frac{n}{z^2} \left\langle \frac{1}{n} \sum_{i=1}^{n} \frac{\tau}{\sqrt{\tau^2 + (y_i - \mu)^2}} - \frac{1}{n} \sum_{i=1}^{n} \frac{\tau_\sigma}{\sqrt{\tau_\sigma^2 + (y_i - \mu)^2}}, v - \sigma \right\rangle$$

$$= \frac{n^{3/2}}{z^3} \cdot \frac{1}{n} \sum_{i=1}^{n} \frac{(y_i - \mu)^2}{(\widetilde{\tau}^2 + (y_i - \mu)^2)^{3/2}} |v - \sigma|^2$$

$$\geq \frac{n^{3/2}}{z^3} \cdot \frac{1}{n} \sum_{i=1}^{n} \frac{(y_i - \mu)^2}{((\tau \vee \tau_\sigma)^2 + (y_i - \mu)^2)^{3/2}} |v - \sigma|^2$$

where $\widetilde{\tau}$ is some convex combination of $\tau$ and $\tau_\sigma$, that is $\widetilde{\tau} = (1-\lambda)\tau_\sigma + \lambda\tau$ for some $\lambda \in [0,1]$. Because $\tau^3 x^2/(\tau^2 + x^2)^{3/2}$ is an increasing function of $\tau$, if $\tau_\varpi \leq \tau \vee \tau_\sigma$, we have

$$\frac{\langle \nabla_v L_n(\mu, v) - \nabla_v L_n(\mu, \sigma), v - v_* \rangle}{|v - \sigma|^2} \geq \frac{n^{3/2}}{z^3(\tau \vee \tau_\sigma)^3} \cdot \frac{1}{n} \sum_{i=1}^n \frac{(\tau \vee \tau_\sigma)^3 (y_i - \mu)^2}{(\tau^2 \vee \tau_\sigma^2 + (y_i - \mu)^2)^{3/2}}$$

$$\geq \frac{n^{3/2}}{z^3(\tau \vee \tau_\sigma)^3} \cdot \frac{1}{n} \sum_{i=1}^n \frac{\tau_\varpi^3 (y_i - \mu)^2}{(\tau_\varpi^2 + (y_i - \mu)^2)^{3/2}}.$$

Thus

$$\inf_{\mu \in \mathbb{B}_r(\mu^*)} \frac{\langle \nabla_v L_n(\mu, v) - \nabla_v L_n(\mu, \sigma), v - v_* \rangle}{|v - \sigma|^2}$$

$$\geq \frac{n^{3/2}}{z^3(\tau \vee \tau_*)^3} \cdot \inf_{\mu \in \mathbb{B}_r(\mu^*)} \frac{1}{n} \sum_{i=1}^n \frac{\tau_\varpi^3 (y_i - \mu)^2}{(\tau_\varpi^2 + (y_i - \mu)^2)^{3/2}}$$

$$= \frac{n^{3/2}}{z^3(\tau \vee \tau_\sigma)^3} \cdot \left( \inf_{\mu \in \mathbb{B}_r(\mu^*)} \left( \mathbb{E} \frac{\tau_\varpi^3 (y_i - \mu)^2}{(\tau_\varpi^2 + (y_i - \mu)^2)^{3/2}} \right) \right.$$

$$\left. - \sup_{\mu \in \mathbb{B}_r(\mu^*)} \left| \frac{1}{n} \sum_{i=1}^n \frac{\tau_\varpi^3 (y_i - \mu)^2}{(\tau_\varpi^2 + (y_i - \mu)^2)^{3/2}} - \mathbb{E} \frac{\tau_\varpi^3 (y_i - \mu)^2}{(\tau_\varpi^2 + (y_i - \mu)^2)^{3/2}} \right| \right)$$

$$= \frac{n^{3/2}}{z^3(\tau \vee \tau_\sigma)^3} \cdot (\mathrm{I} - \mathrm{II}).$$

It remains to lower bound I and upper bound II. We start with I. Let $f(x) = x/(1+x)^{3/2}$ which satisfies

$$f(x) \geq \begin{cases} \epsilon x & x \leq c_\epsilon \\ 0 & x > c_\epsilon, \end{cases}$$

and $Z = (y - \mu)^2/\tau_\varpi^2$ in which $y \sim y_i$. Suppose $r^2 \leq c_\epsilon \tau_\varpi^2/4$, then we have

$$\inf_{\mu \in \mathbb{B}_r(\mu^*)} \left( \mathbb{E} \frac{\tau_\varpi^3 (y_i - \mu)^2}{(\tau_\varpi^2 + (y_i - \mu)^2)^{3/2}} \right) = \inf_{\mu \in \mathbb{B}_r(\mu^*)} \mathbb{E} \left( \frac{\tau_\varpi^2 Z}{(1 + Z)^{3/2}} \right)$$

$$\geq \epsilon \cdot \inf_{\mu \in \mathbb{B}_r(\mu^*)} \mathbb{E} \left[ (y - \mu)^2 1((y - \mu)^2 \leq c_\epsilon \tau_\varpi^2) \right]$$

$$\geq \epsilon \cdot \inf_{\mu \in \mathbb{B}_r(\mu^*)} \mathbb{E} \left[ (y - \mu)^2 1(\varepsilon^2 \leq c_\epsilon \tau_\varpi^2/2 - r^2) \right]$$

$$\geq \epsilon \cdot \inf_{\mu \in \mathbb{B}_r(\mu^*)} \left( \mathbb{E} \left[ (\Delta^2 + \varepsilon^2) 1 \left( \varepsilon^2 \leq \frac{c_\epsilon \tau_\varpi^2}{4} \right) \right] - \frac{8\Delta\sigma^2}{c_\epsilon \tau_\varpi^2} \right)$$

$$\geq \epsilon \cdot \left( \mathbb{E} \left[ \varepsilon^2 1 \left( \varepsilon^2 \leq \frac{c_\epsilon \tau_\varpi^2}{4} \right) \right] - \frac{8r\sigma^2}{c_\epsilon \tau_\varpi^2} \right).$$

We then proceed with II. For any $0 < \gamma \leq 2r$, there exists an $\gamma$-cover $\mathcal{N}$ of $\mathbb{B}_r(\mu^*)$ such that $|\mathcal{N}| \leq 6r/\gamma$. Then for any $\mu \in \mathbb{B}_r(\mu^*)$ there exists an $\omega \in \mathcal{N}$ such that $|\omega - \mu| \leq \gamma$, and thus by Lemma G.5 we have

$$\left| \frac{1}{n} \sum_{i=1}^n \frac{\tau_\varpi^3 (y_i - \mu)^2}{(\tau_\varpi^2 + (y_i - \mu)^2)^{3/2}} - \mathbb{E} \frac{\tau_\varpi^3 (y_i - \mu)^2}{(\tau_\varpi^2 + (y_i - \mu)^2)^{3/2}} \right|$$

$$\leq \left| \frac{1}{n} \sum_{i=1}^n \frac{\tau_\varpi^3 (y_i - \omega)^2}{(\tau_\varpi^2 + (y_i - \omega)^2)^{3/2}} - \mathbb{E} \frac{\tau_\varpi^3 (y_i - \omega)^2}{(\tau_\varpi^2 + (y_i - \omega)^2)^{3/2}} \right|$$

$$+ \left| \frac{1}{n} \sum_{i=1}^n \frac{\tau_\varpi^3 (y_i - \omega)^2}{(\tau_\varpi^2 + (y_i - \omega)^2)^{3/2}} - \frac{1}{n} \sum_{i=1}^n \frac{\tau_\varpi^3 (y_i - \mu)^2}{(\tau_\varpi^2 + (y_i - \mu)^2)^{3/2}} \right|$$

$$+ \left| \mathbb{E} \frac{\tau_\varpi^3 (y_i - \omega)^2}{(\tau_\varpi^2 + (y_i - \omega)^2)^{3/2}} - \mathbb{E} \frac{\tau_\varpi^3 (y_i - \mu)^2}{(\tau_\varpi^2 + (y_i - \mu)^2)^{3/2}} \right|$$

$$= \mathrm{II}_1 + \mathrm{II}_2 + \mathrm{II}_3.$$

For $\text{II}_1$, Lemma G.5 implies with probability at least $1 - 2\delta$

$$\text{II}_1 \leq \sqrt{\frac{2\tau_\varpi^2 \mathbb{E}(y_i - \omega)^2 \log(1/\delta)}{3n}} + \frac{\tau_\varpi^2 \log(1/\delta)}{3\sqrt{3}n} \leq \sqrt{\frac{2\tau_\varpi^2 (\sigma^2 + r^2) \log(1/\delta)}{3n}} + \frac{\tau_\varpi^2 \log(1/\delta)}{3\sqrt{3}n}.$$

Let

$$g(x) = \frac{1}{n} \sum_{i=1}^{n} \frac{\tau^3 (x + \varepsilon_i)^2}{(\tau^2 + (x + \varepsilon_i)^2)^{3/2}}.$$

Using the mean value theorem and the inequality that $|\tau^2 x / (\tau^2 + x^2)^{3/2}| \leq 1/\sqrt{3}$, we obtain

$$|g(x) - g(y)| = \left| \frac{1}{n} \sum_{i=1}^{n} \frac{\tau^3 (\widetilde{x} + \varepsilon_i) \left( \tau^2 - (\widetilde{x} + \varepsilon_i)^2 \right)}{(\tau^2 + (\widetilde{x} + \varepsilon_i)^2)^{5/2}} (x - y) \right| \leq \frac{\tau}{\sqrt{3}} |x - y|.$$

Then we have

$$\text{II}_2 = \left| \frac{1}{n} \sum_{i=1}^{n} \frac{\tau_\varpi^3 (\widetilde{\Delta} + \varepsilon_i) \left( \tau_\varpi^2 - (\widetilde{\Delta} + \varepsilon_i)^2 \right)}{(\tau_\varpi^2 + (\widetilde{\Delta} + \varepsilon_i)^2)^{5/2}} (\Delta_w - \Delta_\mu) \right| \leq \frac{\tau_\varpi \gamma}{\sqrt{3}}$$

where $\widetilde{\Delta}$ is some convex combination of $\Delta_w = \mu^* - w$ and $\Delta_\mu = \mu^* - \mu$. For $\text{II}_3$, we have

$$\text{II}_3 = \left| \mathbb{E} \left( \frac{\tau_\varpi^3 (\widetilde{\Delta} + \varepsilon_i) \left( \tau_\varpi^2 - (\widetilde{\Delta} + \varepsilon_i)^2 \right)}{(\tau_\varpi^2 + (\widetilde{\Delta} + \varepsilon_i)^2)^{5/2}} \right) (\Delta_w - \Delta_\mu) \right| \leq \gamma \mathbb{E} |\widetilde{\Delta} + \varepsilon_i| \leq \gamma \sqrt{\mathbb{E} \left( \widetilde{\Delta} + \varepsilon_i \right)^2},$$

where the last inequality uses Jensen's inequality. Putting the above pieces together and using the union bound, we obtain with probability at least $1 - 12\gamma^{-1} r\delta$

$$\text{II} \leq \sup_{\omega \in \mathcal{N}} \left| \frac{1}{n} \sum_{i=1}^{n} \frac{\tau_\varpi^3 (y_i - \omega)^2}{(\tau_\varpi^2 + (y_i - \omega)^2)^{3/2}} - \mathbb{E} \frac{\tau_\varpi^3 (y_i - \omega)^2}{(\tau_\varpi^2 + (y_i - \omega)^2)^{3/2}} \right| + \frac{\tau_\varpi \gamma}{\sqrt{3}} + \gamma \sqrt{r^2 + \sigma^2}$$

$$\leq \sqrt{\frac{2\tau_\varpi^2 (r^2 + \sigma^2) \log(1/\delta)}{3n}} + \frac{\tau_\varpi^2 \log(1/\delta)}{3\sqrt{3}n} + \frac{\tau_\varpi \gamma}{\sqrt{3}} + \gamma \sqrt{r^2 + \sigma^2}$$

$$= \sqrt{r^2 + \sigma^2} \left( \sqrt{\frac{2\varpi^2 \log(1/\delta)}{3z^2}} + \gamma \right) + \frac{\varpi^2 \log(1/\delta)}{3\sqrt{3}z^2} + \frac{\varpi \gamma \sqrt{n}}{\sqrt{3}}.$$

Combining the bounds for I and II yields with probability at least $1 - \delta$

$$\inf_{\mu \in \mathbb{B}_r(\mu^*)} \frac{\langle \nabla_v L_n(\mu, v) - \nabla_v L_n(\mu, \sigma), v - \sigma \rangle}{|v - \sigma|^2}$$

$$\geq \frac{n^{3/2}}{z^3 (\tau \vee \tau_\sigma)^3} \left\{ \epsilon \left( \mathbb{E} \left[ \varepsilon^2 1 \left( \varepsilon^2 \leq \frac{c_\epsilon \tau_\varpi^2}{4} \right) \right] - \frac{8r\sigma^2}{c_\epsilon \tau_\varpi^2} \right) \right.$$

$$\left. - \sqrt{r^2 + \sigma^2} \left( \sqrt{\frac{2\varpi^2 \log(1/\delta)}{3z^2}} + \gamma \right) - \frac{\varpi^2 \log(1/\delta)}{3\sqrt{3}z^2} - \frac{\varpi \gamma \sqrt{n}}{\sqrt{3}} \right\}$$

$$\geq \frac{1}{2(v \vee \sigma)^3} \mathbb{E} \left[ \varepsilon^2 1 \left( \varepsilon^2 \leq \frac{c_\epsilon \tau_\varpi^2}{4} \right) \right]$$

where $\epsilon, \varpi, \gamma, n$ are picked such that $\epsilon = 3/4$, $\gamma = 12r$, and

$$\epsilon \left( \mathbb{E} \left[ \varepsilon^2 1 \left( \varepsilon^2 \leq \frac{c_\epsilon \tau_\varpi^2}{4} \right) \right] - \frac{8r\sigma^2 z^2}{c_\epsilon \varpi^2 n} \right) - \sqrt{r^2 + \sigma^2} \left( \sqrt{\frac{2\varpi^2 \log(1/\delta)}{3z^2}} + \gamma \right) - \frac{\varpi^2 \log(1/\delta)}{3\sqrt{3}z^2} - \frac{\varpi \gamma \sqrt{n}}{\sqrt{3}}$$

$$\geq \frac{1}{2} \mathbb{E} \left[ \varepsilon^2 1 \left( \varepsilon^2 \leq \frac{c_\epsilon \tau_\varpi^2}{4} \right) \right] \geq \frac{1}{4} \sigma.$$

For example, we can pick $\varpi$ such that

$$\max\{\varpi r \sqrt{n}, \varpi\} \to 0 \text{ and } \varpi \sqrt{n} \to \infty$$

as $n \to \infty$. This completes the proof.

$\square$

## G.5 SUPPORTING LEMMAS

This subsection proves a supporting lemma that is used prove Lemma G.4.

**Lemma G.5.** Let $w_i$ be i.i.d. copies of $w$. For any $0 < \delta < 1$, we have

$$\frac{1}{n}\sum_{i=1}^n \frac{\tau^3 w_i^2}{(\tau^2+w_i^2)^{3/2}} - \mathbb{E}\frac{\tau^3 w_i^2}{(\tau^2+w_i^2)^{3/2}} \geq -\sqrt{\frac{2\tau^2\mathbb{E}w_i^2\log(1/\delta)}{3n}} - \frac{\tau^2\log(1/\delta)}{3\sqrt{3}n}, \text{ with prob. } 1-\delta,$$

$$\left|\frac{1}{n}\sum_{i=1}^n \frac{\tau^3 w_i^2}{(\tau^2+w_i^2)^{3/2}} - \mathbb{E}\frac{\tau^3 w_i^2}{(\tau^2+w_i^2)^{3/2}}\right| \leq \sqrt{\frac{2\tau^2\mathbb{E}w_i^2\log(1/\delta)}{3n}} + \frac{\tau^2\log(1/\delta)}{3\sqrt{3}n}, \text{ with prob. } 1-2\delta.$$

*Proof of Lemma G.5.* We only prove the first result and the second result follows similarly. The random variables $Z_i = Z_i(\tau) := \tau^3 w_i^2/(\tau^2+w_i^2)^{3/2}$ with $\mu_z = \mathbb{E}Z_i$ and $\sigma_z^2 = \text{var}(Z_i)$ are bounded i.i.d. random variables such that

$$0 \leq Z_i = \tau^3 w_i^2/(\tau^2+w_i^2)^{3/2} \leq w_i^2 \wedge \frac{\tau^2}{\sqrt{3}} \wedge \frac{\tau|w_i|}{\sqrt{3}}.$$

Moreover we have

$$\mathbb{E}Z_i^2 = \mathbb{E}\left(\frac{\tau^6 w_i^4}{(\tau^2+\varepsilon_i^2)^3}\right) \leq \frac{\tau^2\mathbb{E}w_i^2}{3}, \ \sigma_z^2 := \text{var}(Z_i) \leq \frac{\tau^2\mathbb{E}w_i^2}{3}.$$

For third and higher order absolute moments, we have

$$\mathbb{E}|Z_i|^k = \mathbb{E}\left|\frac{\tau^3 w_i^2}{(\tau^2+\varepsilon_i^2)^{3/2}}\right|^k \leq \frac{\tau^2\mathbb{E}w_i^2}{3}\cdot\left(\frac{\tau^2}{\sqrt{3}}\right)^{k-2} \leq \frac{k!}{2}\cdot\frac{\tau^2\mathbb{E}w_i^2}{3}\cdot\left(\frac{\tau^2}{3\sqrt{3}}\right)^{k-2}, \text{ for all integers } k \geq 3.$$

Therefore, using Lemma H.2 with $v = n\tau^2\,\mathbb{E}w_i^2/3$ and $c = \tau^2/(3\sqrt{3})$ acquires that for any $t \geq 0$

$$\mathbb{P}\left(\sum_{i=1}^n \frac{\tau^3 w_i^2}{(\tau^2+\varepsilon_i^2)^{3/2}} - \sum_{i=1}^n \mathbb{E}\left(\frac{\tau^3 w_i^2}{(\tau^2+\varepsilon_i^2)^{3/2}}\right) \geq -\sqrt{\frac{2n\tau^2\mathbb{E}w_i^2 t}{3}} - \frac{\tau^2 t}{3\sqrt{3}}\right) \leq \exp(-t).$$

Taking $t = \log(1/\delta)$ acquires that for any $0 < \delta < 1$

$$\mathbb{P}\left(\frac{1}{n}\sum_{i=1}^n \frac{\tau^3 w_i^2}{(\tau^2+w_i^2)^{3/2}} - \frac{1}{n}\sum_{i=1}^n \mathbb{E}\left(\frac{\tau^3 w_i^2}{(\tau^2+\varepsilon_i^2)^{3/2}}\right) > -\sqrt{\frac{2\tau^2\mathbb{E}w_i^2\log(1/\delta)}{3n}} - \frac{\tau^2\log(1/\delta)}{3\sqrt{3}n}\right) > 1-\delta.$$

This finishes the proof. $\qquad\square$

# H PRELIMINARY LEMMAS

This section collects preliminary lemmas that are frequently used in the proofs for the main results and supporting lemmas. We first collect the Hoeffding's inequality and then present a form of Bernstein's inequality. We omit their proofs and refer interested readers to Boucheron et al. (2013).

**Lemma H.1** (Hoeffding's inequality)**.** Let $Z_1, \ldots, Z_n$ be independent real-valued random variables such that $a \leq Z_i \leq b$ almost surely. Let $S_n = \sum_{i=1}^n (Z_i - \mathbb{E}Z_i)$ and $v = n(b-a)^2$. Then for all $t \geq 0$,

$$\mathbb{P}\left(S_n \geq \sqrt{vt/2}\right) \leq e^{-t}, \ \mathbb{P}\left(S_n \leq -\sqrt{vt/2}\right) \leq e^{-t}, \ \mathbb{P}\left(|S_n| \geq \sqrt{vt/2}\right) \leq 2e^{-t}.$$

**Lemma H.2** (Bernstein's inequality)**.** Let $Z_1, \ldots, Z_n$ be independent real-valued random variables such that

$$\sum_{i=1}^n \mathbb{E}Z_i^2 \leq v, \ \sum_{i=1}^n \mathbb{E}|Z_i|^k \leq \frac{k!}{2}vc^{k-2} \text{ for all } k \geq 3.$$

If $S_n = \sum_{i=1}^n (Z_i - \mathbb{E}Z_i)$, then for all $t \geq 0$,

$$\mathbb{P}\left(S_n \geq \sqrt{2vt}+ct\right) \leq e^{-t}, \ \mathbb{P}\left(S_n \leq -(\sqrt{2vt}+ct)\right) \leq e^{-t}, \ \mathbb{P}\left(|S_n| \geq \sqrt{2vt}+ct\right) \leq 2e^{-t}.$$

*Proof of Lemma H.2.* This lemma involves a two-sided extension of Theorem 2.10 by Boucheron et al. (2013). The proof follows from a similar argument used in the proof of Theorem 2.10, and thus is omitted. $\qquad\square$

Our third lemma concerns the localized Bregman divergence for convex functions. It was first established in Fan et al. (2018). For any loss function $L$, define the Bregman divergence and the symmetric Bregman divergence as

$$D_L(\beta_1, \beta_2) = L(\beta_1) - L(\beta_2) - \langle \nabla L(\beta_2), \beta_1 - \beta_2 \rangle,$$
$$D_L^s(\beta_1, \beta_2) = D_L(\beta_1, \beta_2) + D_L(\beta_2, \beta_1).$$

**Lemma H.3.** For any $\beta_\eta = \beta^* + \eta(\beta - \beta^*)$ with $\eta \in (0, 1]$ and any convex loss function $L$, we have

$$D_L^s(\beta_\eta, \beta^*) \leq \eta D_L^s(\beta, \beta^*).$$

Our forth lemma in this section concerns three basic inequalities that are frequently used in the proofs.

**Lemma H.4.** The following inequalities hold:

(i) $(1 + x)^r \geq 1 + rx$ for $x \geq -1$ and $r \in \mathbb{R} \setminus (0, 1)$;

(ii) $(1 + x)^r \leq 1 + rx$ for $x \geq -1$ and $r \in (0, 1)$;

(iii) $(1 + x)^r \leq 1 + (2^r - 1)x$ for $x \in [0, 1]$ and $r \in \mathbb{R} \setminus (0, 1)$.