# OpenReview forum: "Adapting to both finite-sample and asymptotic regimes"
_ICLR.cc/2025/Conference — ICLR 2025 Conference Withdrawn Submission_

### Official Review · Reviewer_xy4j · 2024-10-21

**Soundness:** 2
**Presentation:** 2
**Contribution:** 2
**Rating:** 3
**Confidence:** 3

**Summary:**

The paper considers empirical risk minimization (ERM) using the pseudo-Huber loss so that it jointly optimizes the robustification parameter of the loss function during the ERM process for the task of estimating the mean of an scalar random variable from i.i.d. samples in the general case when the unknown distribution of the scalar random variable is assumed only to have finite variance.
The papers goal is to provide a self-tuning estimator which can achieve a sharp upper bound on the generalization error.

This is a resubmission with small changes, the paper has been reviewed for TMLR under the title "Do we need to estimate the variance in robust mean estimation?": https://openreview.net/forum?id=CIv8NfvxsX. Unfortunately, the paper does not address any of the main concerns which were raised by the rejection for its previous version.

**Strengths:**

As the authors pointed out, mean estimation using the pseudo-Huber loss with adaptive robustification parameter has been studied before (Ronchetti & Huber, 2009), and understanding its theoretical properties would be a significant contribution to the field.

**Weaknesses:**

I agree to the main concerns raised for the rejection of the previous version of this paper: https://openreview.net/forum?id=CIv8NfvxsX.

The generalization bound of Theorem 3.2 is far from optimal, and there is an other algorithm already out there which reaches the optimal performance and not even mentioned in this paper by Lee and Valiant (FOCS, 2021): https://doi.org/10.1109/FOCS52979.2021.00071. I believe this result is too relevant to be ignored here.

The algorithm of the paper is not completely self-tuning, it has two tuning parameters $v_0$ and $V_0$ which need to bracket the unknown variance. This issue should be discussed because it challenges the adaptivity of the presented algorithm.

The bound of Theorem 3.2 is scaled by an extra $\sqrt{\log(n)}$ factor compared to the optimal performance. Worse, the "constant" C is scaled by the standard deviation $\sigma$ as mentioned in line 1480 and the derivation above, and it seems to me that so it can even grow arbitrarily large. So it is not clear then what is the real dependence of the bound of Theorem 3.2 on $\sigma$.

**Questions:**

Maybe the $\log(n)$ factor could be removed in Lemma F.1 and so from the final bound by using the chaining argument (e.g., Section 3 in the book of Pollard 1990 on Empirical Processes).

---

### Official Review · Reviewer_tYVe · 2024-11-01

**Soundness:** 1
**Presentation:** 2
**Contribution:** 1
**Rating:** 3
**Confidence:** 5

**Summary:**

The paper discusses a novel mean estimator such that it has the optimal finite-sample and asymptotical guarantees in some sense. Their method requires no parameter to tune and is computational efficient. They also validate their result with some other classical estimators in some simulation studies.

**Strengths:**

The paper aims to tackle a fundamental problem raised from Catoni’s seminar result that attains optimal finite-sample and asymptotic performance guarantee.

**Weaknesses:**

-	**Missing Comparisons**: It is concerning that the authors have repeatedly avoided comparing their work to [Lee and Valiant 2021], which addresses optimal sub-Gaussian mean estimation without knowledge of the variance. Previous reviewers have highlighted this omission, but it remains unaddressed. The estimator in [Lee and Valiant 2021] achieves optimal finite-sample guarantees and asymptotic variance (“as accurate as the sample mean for the Gaussian of matching variance”) without requiring prior knowledge of the exact variance. The omission is significant, as [Lee and Valiant 2021] presents a concentration result with an optimal dependence factor of $\sigma (1 + o(1))$ before the $\sqrt{2\log(1/\delta)}$, while the current submission’s concentration result (e.g., Theorem 3.2) shows a gap and depends on some numerical constant $C$.
-	**Self-tuning property without knowing the variance**:  Contrary to the paper’s claims, the proposed estimator does not fully exhibit the self-tuning property for unknown variance. It requires upper and lower bounds on the variance (as outlined in Section 3), yet the authors provide no practical guidance on selecting these bounds in the numerical experiments. By comparison, the results in [Lee and Valiant 2021] do not rely on such assumptions.
-	**Paper Scope**: The current title is too wide and could be made more precise. Adding specific terms like “mean estimator” would make the title better reflect the paper scope and contributions.
-	**Inconsistent Notation**: Notable inconsistencies appear throughout the main body, where $\hat\mu (\hat v) $ and $\hat\mu(\tau) $ are used interchangeably. Additionally, it would improve clarify $\sigma_u$ is defined before $\sigma_{x^2}$, as the notation for sigma_{x^2}$ is too specific.


[Lee and Valiant 2021] Optimal Sub-Gaussian Mean Estimation in R. FOCS 2021.

**Questions:**

-	Can the authors clarify the conditions in Theorem 3.1 regarding the inequalities involving $v_0, V_0, \sigma$? Providing concrete examples with classical distributions and details when these conditions are met under relevant parameter settings would be helpful.

---

### Official Review · Reviewer_y2pi · 2024-11-01

**Soundness:** 4
**Presentation:** 4
**Contribution:** 3
**Rating:** 6
**Confidence:** 3

**Summary:**

This paper presents an empirical risk minimization (ERM) approach incorporating concomitant scaling, designed to eliminate the need for tuning a robustification parameter when handling heavy-tailed data. The method introduces a novel loss function that simultaneously optimizes both the mean and robustification parameters. By jointly optimizing these parameters, the robustification parameter adapts automatically to the unknown variance in the data, making the method self-tuning. The authors highlight improvements over previous approaches in both computational and estimation efficiency. The method circumvents the need for cross-validation or Lepski’s method for tuning, while the estimator's variance meets the Cramer-Rao lower bound, signifying optimal asymptotic efficiency. This approach is described as algorithmically adaptive to both finite-sample and large-sample contexts, demonstrating consistent performance across these regimes. Numerical experiments further support the efficacy of the proposed methodology.

**Strengths:**

The paper tackles a fundamental and ubiquitous problem in statistical learning, i.e., robust mean estimation. It does so by providing an optimization method that is shown to be efficient from an estimation point of view (both in finite samples and asymptotically), as well as from a computational point of view (the joint optimization of the mean and robustification parameter is more efficient than, for instance, cross-validation methods).

The paper is also extremely clear in its presentation of the results, their motivation, and the intuition behind them.

**Weaknesses:**

Here are a few minor weaknesses:

1. Subsection 3.2 sounds a bit off compared to the rest of the paper (the language is sloppy and the presentation not as clear), I think it could be better rewritten

2. There's a typo in the captions of Figures 3 and Figure 5 ("distributution")

**Questions:**

Reading the paper, I was curious about two points that I think the authors could clarify:

1. How similar is their loss to the Ronchetti & Huber one? Given that they bring this point up, I think they should be clearer about the connections;

2. Though I see that the loss function is tailored specifically to the robustification of the quadratic loss (it is quadratic near the origin, linear away from it), do the authors think that their self-tuning approach could be extended to more general loss functions?

---

### Official Review · Reviewer_RqbM · 2024-11-02

**Soundness:** 2
**Presentation:** 1
**Contribution:** 1
**Rating:** 1
**Confidence:** 4

**Summary:**

The authors study the empirical risk minimisation problem on a variant of the Huber loss for estimating a scalar signal corrupted by a noise with mean zero and finite variance. In practice, this consists in optimising jointly for the signal we wish to recover and a "robustification" parameter.

**Strengths:**

This paper is a nice exercise in basic statistics and probability. The proofs seem to be correct.

**Weaknesses:**

I believe this paper requires some fundamental changes in order to be published in any venue.

The title is really not informative and should be changed. What is being adapted "to both finite sample and asymptotic regimes"? That is not clear until all the paper has been read.

The paper is nearly impossible to read, as it is inconsistent in stating its results and the precise model under consideration.

The introduction is confusing. One one hand it contains irrelevant details (like the definition of sub-gaussian variables) and on the other hand is really generic: it states that in practice you have noisy data, but this doesn't automatically imply a heavy tailed distribution. As a matter of fact, the paper simply studies sub-gaussian noise with mean zero and finite variance.

The specific loss that you are using is redefined a number of times, until finally defining 2.5. Everything before that in section 2 is superfluous. Additionally, you fix $a=0.5$ by looking at the population loss. This to me is a fundamentally unjustified choice.

The significance of the contribution is also not clear to me. If I were to look at the sklearn implementation of Huber regression I would also find a modification of the Huber loss where one optimise for the estimator and a "robustness" parameter [1], which is in essence the same idea of the paper.


Minor points:
There are a number of inaccuracies and inconsistencies. I list some of the most glaring ones.

1. Line 34: what assumed Gaussian shape?
2. Line 36: I am inferring you mean that the mean is not finite. If so it would be better to write it.
3. Line 59: it's not clear the role of the parameter $\tau$
4. Line 74: $d$ was not defined before.
5. Line 140: when referring to sub-Gaussian performance, it would be good to point to the relevant equation before
6. Line 141: adding a new parameter $v$ serves no purpose as $v = \sigma$


[1] A. Owen 2006, "A robust hybrid of lasso and ridge regression"

**Questions:**

1. Can you comment (at the very least numerically) on the case where the noise has Pareto or Cauchy distribution?
2. Would you be able to offer some numerical evidence that picking $a=0.5$ is indeed a good choice?
3. Could you comment on the differences between your approach and [1]?



[1] A. Owen 2006, "A robust hybrid of lasso and ridge regression"

---

### Note · Authors · 2024-11-13

I have read and agree with the venue's withdrawal policy on behalf of myself and my co-authors.